# PUMILIO proteins promote colorectal cancer growth via suppressing p21

Yuanyuan Gong 1,2,3,8, Zukai Liu1,8, Yihang Yuan4,8, Zhenzhen Yang1,8, Jiawei Zhang 1, Qin Lu4, Wei Wang1, Chao Fang 4,5✉, Haifan Lin 6✉ & Sanhong Liu 1,7✉

PUMILIO (PUM) proteins belong to the highly conserved PUF family post-transcriptional regulators involved in diverse biological processes. However, their function in carcinogenesis remains under-explored. Here, we report that *Pum1* and *Pum2* display increased expression in human colorectal cancer (CRC). Intestine-specific knockout of *Pum1* and *Pum2* in mice significantly inhibits the progression of colitis-associated cancer in the AOM/DSS model. Knockout or knockdown of *Pum1* and/or *Pum2* in human CRC cells result in a significant decrease in the tumorigenicity and delayed G1/S transition. We identify p21/Cdkn1a as a direct target of PUM1. Abrogation of the PUM1 binding site in the *p21* mRNA also results in decreased cancer cell growth and delayed G1/S transition. Furthermore, intravenous injection of nanoparticle-encapsulated anti-*Pum1* and *Pum2* siRNAs reduces colorectal tumor growth in murine orthotopic colon cancer models. These findings reveal the requirement of PUM proteins for CRC progression and their potential as therapeutic targets.

[1] Shanghai Institute for Advanced Immunochemical Studies and School of Life Science and Technology, ShanghaiTech University, Shanghai, China. [2] Shanghai Institute of Nutrition and Health, Chinese Academy of Sciences, Shanghai, China. [3] University of Chinese Academy of Sciences, Beijing, China. [4] Hongqiao International Institute of Medicine, Tongren Hospital and State Key Laboratory of Oncogenes and Related Genes, Department of Pharmacology and Chemical Biology, Shanghai Jiao Tong University School of Medicine (SJTU-SM), Shanghai, China. [5] Key Laboratory of Basic Pharmacology of Ministry of Education & Joint International Research Laboratory of Ethnomedicine of Ministry of Education, Zunyi Medical University, Zunyi, China. [6] The Yale Stem Cell Center and Department of Cell Biology, Yale University School of Medicine, New Haven, CT, USA. [7] Shanghai Frontiers Science Center of TCM Chemical Biology, Institute of Interdisciplinary Integrative Medicine Research, Shanghai University of Traditional Chinese Medicine, Shanghai, China. [8]These authors contributed equally: Yuanyuan Gong, Zukai Liu, Yihang Yuan, Zhenzhen Yang. ✉email: fangchao32@sjtu.edu.cn; haifan.lin@yale.edu; liush@shutcm.edu.cn

According to a global epidemiological analysis in 2018, colorectal cancer (CRC) accounts for 10% of all cancers, ranking third, with its mortality rate ranked second[1]. Although the roles of genetic alterations such as *APC*, *TP53*, *KRAS*, and key signaling pathways such as Wnt, RAS-MAPK, and PI3K in driving CRC have been extensively studied[2,3], the incidence and mortality of CRC are still at the forefront of all types of cancer and have been on the rise. There is a pressing need to investigate new molecular mechanisms underlying the pathogenesis of CRC for early detection, diagnosis, and targeted therapy. Recent studies have shown that RNA binding proteins appear to be important for the initiation and development of CRC[4–6]. However, the exact roles of these proteins in CRC and how they regulate RNA targets in CRC remain largely unknown.

PUMILIO (PUM) proteins belong to the highly conserved PUF (Pumilio-Fem3-binding Factor) family of RNA-binding proteins[7]. There are two mammalians PUM homologs, PUM1 and PUM2. They bind to target transcripts through the PUMILIO Response Element (PRE) that contains an eight-nucleotide sequence motif (UGUANAUA). In most cases, PUM proteins bind to the 3′UTR of target mRNAs and recruit partner proteins to reduce the stability of the mRNAs and/or inhibit their translation[8–10]. Many PUF target genes have been identified for their important roles in embryonic development, body axis formation, body size control, neurogenesis, spermatogenesis, and self-renewal of stem cells[7,11–21]. In addition, PUM proteins have recently emerged as promising regulators in cancer. Several studies reported that PUM proteins are required for cancer cell growth[22–26]. However, an investigation suggested that PUM represses oncogenes in multiple cancer types including CRC[27,28]. These contradictory proposals indicate that the function and action mechanism of PUM proteins in colorectal oncogenesis remain largely elusive.

Here, we report the important function of human PUM1 and PUM2 in CRC tumorigenesis. In addition, we demonstrate that this function is partly achieved by repressing the expression of negative cell cycle regulators p21. Furthermore, we show that intravenous injection of nanoparticles encapsulated with anti-*Pum1* and *Pum2* siRNAs significantly inhibits the growth of human colorectal tumors in mice. These findings define the biological function of PUM proteins in CRC and the underlying molecular mechanism. Moreover, it provides a new approach for the treatment of CRC.

## Results

### PUM1 and PUM2 display increased expression in human CRC.

To investigate the functions of PUM proteins in cancer, we analyzed two online databases containing patients' clinical features and gene expression data. We first examined the expression of PUM proteins in different tumors in the Human Protein Atlas (www.proteinatlas.org), and found that PUM1 had the highest protein level in CRC compared to other cancer types and PUM2 was also highly expressed in CRC (Supplementary Fig. 1a–b). We then examined Cancer Genome Atlas (TCGA) Datasets (http://software.broadinstitute.org/software/igv/tcga), which revealed that transcripts of *Pum1* but not *Pum2* was expressed at higher levels in Hong colorectal[29] and Skrzypczak colorectal[30] as compared to normal colon tissues (Supplementary Fig. 1c–d).

To further correlate *Pum1* and *Pum2* to CRC, we performed a pairwise comparison of mRNA expression in human CRC samples from 22 patients with their adjacent tissues. We found that *Pum1* and *Pum2* were highly expressed in CRC clinical specimens (Fig. 1a, c). Particularly, the mRNA level of *Pum1* in 15 patients and *Pum2* in 13 patients were at least 10-fold higher than their paired adjacent normal tissues. Furthermore, TCGA datasets showed that *Pum1* but not *Pum2* expression was positively correlated with CRC stage (Fig. 1b, d). We then plotted receiver operating characteristic (ROC) curves to further analyze the diagnostic values of *Pum1* and *Pum2* with Hong and Skrzypczak colorectal datasets. Our analysis showed that *Pum1* has a larger area under the curve (AUC) values than *Pum2* in CRC (Supplementary Fig. 1e–f). In addition, there may be a negative correlation between PUM1 expression and the survival rate of CRC patients (Supplementary Fig. 1g–h). Together, all these data implicate that PUM1 and PUM2 were involved in the progression of human CRC.

### Intestine-specific deletion of *Pum1* and *Pum2* inhibit AOM/DSS-induced colon carcinogenesis in vivo.

To examine the involvement of PUM proteins in CRC, we employed a well-established azoxymethane/dextran sodium sulfate (AOM/DSS) mouse model to evaluate the role of PUM proteins by conditional knockout of *Pum1* and *Pum2* genes in the intestinal epithelium (Fig. 1e). *Lgr5^cre^::Pum1^flox/flox^::Pum2^flox/flox^* mice were generated by crossing *Lgr5-cre* mice with *Pum1^flox/flox^::Pum2^flox/flox^* mice. We treated *Lgr5^cre^::Pum1^flox/flox^::Pum2^flox/flox^* mice with tamoxifen to obtain mice with *Pum1* and *Pum2* double knockout in colon epithelial cells (*Pum1/2^CKO^*). Tamoxifen-treated *Pum1^flox/flox^::Pum2^flox/flox^* mice were used as controls. Both experimental and control mice were injected with AOM, a procarcinogen that induces G to A transitions in DNA[31], followed by three cycles of treatment with DSS, a reagent that causes epithelial injury and subsequent colonic inflammation[32]. Immunohistochemical analysis revealed no significant presence of PUM1 and PUM2 proteins in the colon of *Pum1/2^CKO^* mice (Fig. 1j), confirming the effectiveness of the knockout. After AOM/DSS treatment, the number as well as the size of tumors were markedly reduced in the gut of *Pum1/2^CKO^* mice, as compared with control mice (Fig. 1f). Quantitatively, the number of large tumors (>4 mm in diameter) in *Pum1/2^CKO^* mice was drastically decreased by 91.5%, and the number of mid-sized (2–4 mm in diameter) and small (<2 mm) tumors were also reduced, by 59.4% and 25.9%, respectively (Fig. 1g). The morphology of the small intestine of knockout mice did not change as compared with the control mice before and after AOM/DSS treatment, indicating the difference in tumor burden between the *Pum1/2^CKO^* and control mice were not due to any difference in intestinal anatomy (Supplementary Fig. 2). Intriguingly, *Pum1/2^CKO^* mice showed longer colons and heavier body weight than control mice (Fig. 1h, i), despite that the *Pum1/2^KO^* mice are smaller than the wildtype controls[33]. This suggests that control mice were more prone to tumors, since shortening of colon length in DSS-induced mice is an indicator of inflammation severity, and inflammation is a potential cause of carcinogenesis in the colon[34]. In addition, there was a profound decrease in adenomas that display a high grade of dysplasia in *Pum1/2^CKO^* mice (Fig. 1k). This indicates that the malignant progression was inhibited in *Pum1/2^CKO^* intestine. Furthermore, Ki-67 and TUNEL staining revealed that *Pum1/2^CKO^* dramatically reduced tumor cell proliferation but had no significant effect on apoptosis (Fig. 1l, m). Taken together, these results demonstrate that *Pum1/2* deficiency significantly inhibits the initiation and development of CRC.

### PUM proteins contribute to CRC cell growth in vitro and their tumorigenicity in vivo.

To further characterize the function of PUM1 and PUM2 in CRC, we first examined the expression of *Pum1* and *Pum2* in six different CRC cell lines, with a normal human colon mucosal epithelial cell line, NCM460, as a control (Fig. 2a, b). *Pum1* was expressed in all six CRC cell lines at significantly higher levels than the normal cell line, with HCT116

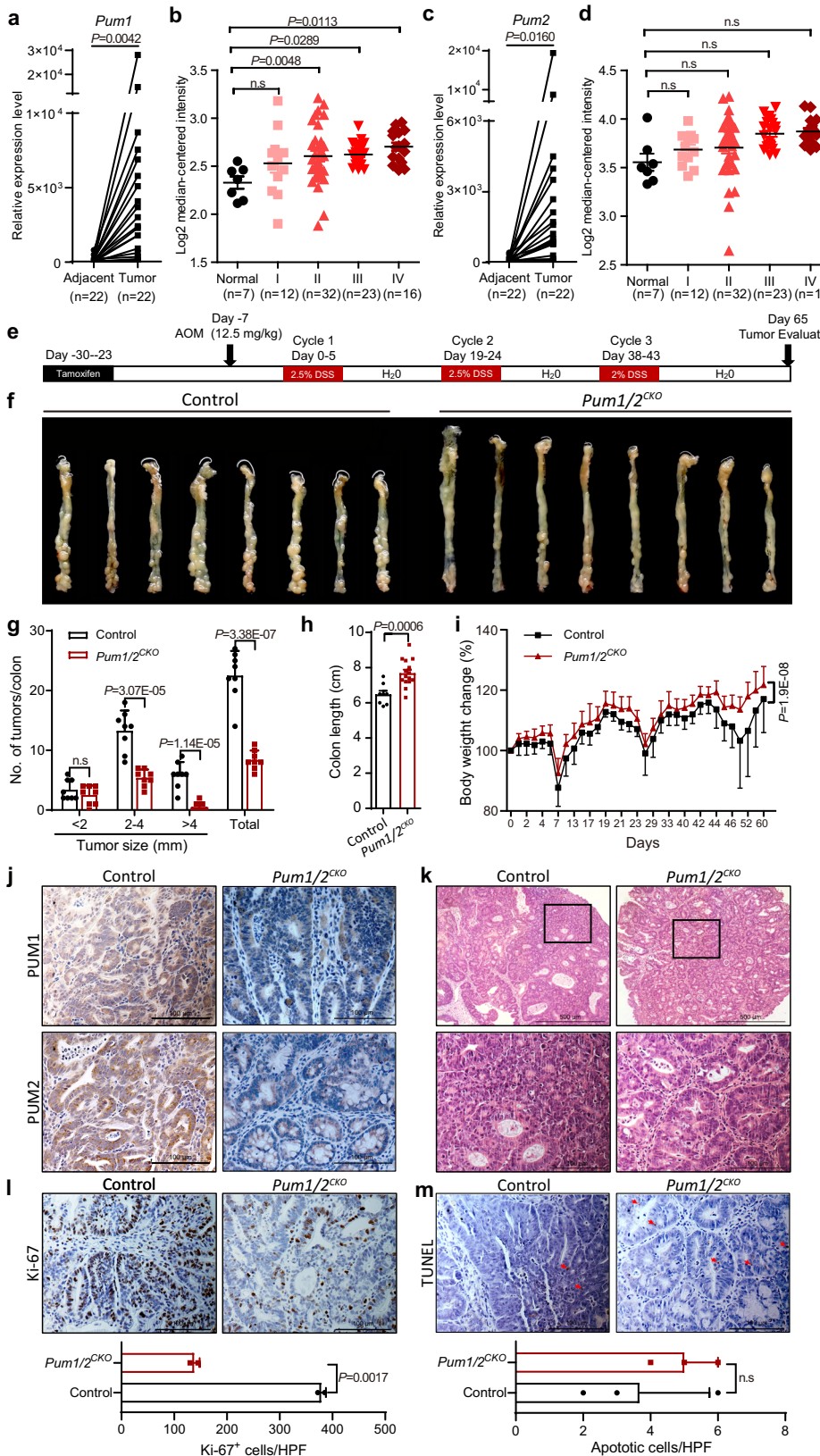

cells showed the highest levels of protein expression. *Pum2* also showed abnormally high levels of expression in five out of six cancer cell lines, even though the extent of over-expression was generally not as large as *Pum1*.

In order to study the function of *Pum1* and *Pum2* in CRC, we chose HCT116 as our cell model since it had the highest levels of

Pum1 and Pum2 expression among detected CRC cell lines. We knocked down *Pum1* and/or *Pum2* in HCT116 cells using siRNAs (Supplementary Fig. 3a). These knockdowns reduced the colony formation ability and increased cell-doubling time (Supplementary Fig. 3b, e), indicating that *Pum1* and/or *Pum2* knockdown inhibited the growth of CRC cells in vitro. In addition, we found

**Fig. 1 Pum1/2$^{CKO}$ suppresses the development of colorectal cancer. a, c** RT-qPCR examination of the relative *Pum1* (**a**) or *Pum2* (**c**) mRNA levels in human CRC specimens from 22 patients as compared to adjacent normal tissues. Actin was used as an internal control. **b, d** mRNA expression of *Pum1* (**b**) or *Pum2* (**d**) in human colorectal cancer clinical specimens at different stages using the TCGA mRNA HiSeq expression array data. Error bars represent SD. **e** Schematic overview of the CRC induction model. C57BL/6J mice ($n = 8$ for each group) were injected with tamoxifen every other day for 7 days to knock out *Pum1/2* in intestinal epithelial cells. 16 days later, the mice were injected with azoxymethane (AOM), followed by three cycles of treatment with dextran sodium sulfate (DSS), as described in "Methods" section. **f** Representative images of colon and rectum in control (*Pum1$^{flox/flox}$::Pum2$^{flox/flox}$*) and *Pum1/2$^{CKO}$* (*Lgr5$^{cre}$::Pum1$^{flox/flox}$::Pum2$^{flox/flox}$*) mice after AOM/DSS treatment, depicting the extent of tumor burden. **g** The number and diameter of tumors in the entire colon and rectum were measured at the end of the study ($n = 8$). Error bars represent SD. **h** The length of control and *Pum1/2$^{CKO}$* colon after AOM/DSS treatment ($n = 8$). Error bars represent SD. **i** Body weight of *Pum1/2$^{CKO}$* and control mice in AOM/DSS model from day 0 to 60 normalized to body weight at day 0 was normalized to 100 ($n = 8$). Error bars represent SD. **j** Representative micrographs of immunohistochemistry staining for PUM1 and PUM2 in colon tissues containing tumors from control and *Pum1/2$^{CKO}$* mice after AOM/DSS treatment. **k** Representative micrographs of H&E staining of colon tumors in control and *Pum1/2$^{CKO}$* mice after AOM/DSS treatment. **l, m** Representative micrographs of immunochemistry staining for Ki-67 (**l**) and TUNEL (**m**) in tumors from control and *Pum1/2$^{CKO}$* mice after AOM/DSS treatment. Bar graph showing the numbers of Ki-67 (**l**, $n = 2$) and TUNEL (**m**, $n = 3$) positive tumor cells per high power fields (HPF) from control and *Pum1/2$^{CKO}$* mice after AOM/DSS treatment. Error bars represent SD. Source data are provided as a Source Data file.

that *Pum1* and/or *Pum2* knockdown reduced the number of cells in S phase (Supplementary Fig. 3f, g). Moreover, knockdown of *Pum1/2* did not have a significant impact on the survival and migration of HCT116 cells, as indicated by the transwell assay and flow-cytometry using apoptotic markers (Supplementary Fig. 4). This indicates that the growth deficiency of CRC cells caused by loss of *Pum1* and *Pum2* was contributed by delayed G1/S transition. Notably, *Pum1* and *Pum2* knockdown showed similar defects in colorectal growth, and double knockdown of *Pum1* and *Pum2* showed additive defects, indicating that they play similar roles.

To assess the entire spectrum of *Pum* function in CRC, we knocked out *Pum1* or *Pum2* in both HCT116 and RKO cells, using the CRISPR/Cas9 nickase method[35]. Our designed sgRNAs were able to target all isoforms of *Pum1* and *Pum2*, respectively (Supplementary Fig. 5). Immunofluorescence staining indicated that PUM1 and PUM2 are diffusely distributed in the cytoplasm of HCT116 and RKO cells and that both proteins were no longer detectable in the knockout cells, validating the effectiveness of the deletions (Supplementary Fig. 6a, b). The cytoplasmic localization and depletion of PUM1 and PUM2 were further confirmed by cytoplasm-nuclear fractionation (Supplementary Fig. 6c, d) and western blotting (Fig. 2c), respectively. In addition, we tried to generate *Pum1/2* double knockout HCT116 cells by screening more than 400 single cell clones but failed to recover any double knockout cell. This result indicates that *Pum1/2* double knockout may seriously impair HCT116 cells survival.

The deletion of *Pum1* and *Pum2* significantly inhibited the colony formation and proliferation of HCT116 and RKO cells, as indicated by two independent *Pum1* (*Pum1$^{-/-}$*) and *Pum2* (*Pum2$^{-/-}$*) knockout clones (Fig. 2d–i and Supplementary Fig. 7). Consistently, *Pum1*-deficiency caused a delay in G1/S transition in both HCT116 and RKO cells (Fig. 2j, k). In the xenograft assay, knocking out *Pum1* significantly inhibited the tumorigenicity of HCT116 and RKO cells (Fig. 2l–q). *Pum2$^{-/-}$* clones displayed similar phenotype, though less severe in HCT116 cells (Fig. 2d–q). Together, these results indicate that both *Pum1* and *Pum2* are essential to CRC tumorigenicity in vivo and cell growth in vitro.

Because long G1 (i.e., slow cycling) and quiescence are hallmarks of cancer stem cells (CSCs), we wondered if *Pum1*-deficiency affected CSC-related properties of HCT116 and RKO cells. For this reason, we conducted RT-qPCR analysis to detect the expression of putative CSC markers, such as CD133[36–39], CD166[38–40], CD26[41], CD44[38,40,42], EpCAM[40], GLI-1[43], Msi1[44,45], ALDH1A1[46,47] and Lgr5[38,48–51] in WT, *Pum1$^{-/-}$*-1, *Pum1$^{-/-}$*-2, *Pum2$^{-/-}$*-1, and *Pum2$^{-/-}$*-2 HCT116 cells (Supplementary Fig. 6e). Among them, ALDH1A1 and Lgr5 expression was too low to detect. CD133, CD166, CD44, GLI-1, and Msi1 showed

various degrees of reduction in the *Pum1/2*-deficient cells. The expression of CD26 was significantly downregulated in both *Pum1* and *Pum2* knockout cells, while EpCAM expression was decreased in *Pum2* but not *Pum1* knockout cells. CD133 is most drastically downregulated in *Pum1$^{-/-}$*-1, *Pum1$^{-/-}$*-2, and *Pum2$^{-/-}$*-1, except for one *Pum2$^{-/-}$* replicate (*Pum2$^{-/-}$*-2). Thus, *Pum1/2* knockout may have an effect on the CSC population.

To further characterize the effect of *Pum1/2* on CSC properties, we conducted flow cytometry using CD44 and CD133, two common colorectal CSC markers, and Lgr5$^+$, an intestinal stem cell marker. In *Pum1$^{-/-}$* and *Pum2$^{-/-}$* cells, the number of CD133$^+$ cells was remarkably reduced, except for one sample (*Pum2$^{-/-}$*-2) but the number of CD44$^+$ cells was not reduced (Supplementary Fig. 6f, h). Unexpectedly, the number of Lgr5$^+$ cells was increased (Supplementary Fig. 6g, i). These changes reflect that PUM1 and PUM2 have a function in regulating stem cell fate, with their deficiency leads to increased CD133$^-$ Lgr5$^+$ cells.

**PUM1 functions in CRC cells mainly by directly regulating cancer-related and cell cycle-related genes.** To explore the molecular mechanism underlying the PUM function in CRC, we analyzed the transcriptome of *Pum1$^{-/-}$* and *Pum2$^{-/-}$* HCT116 cells by deep sequencing (Fig. 3a, b). We identified 1132 and 1226 differentially expressed genes in *Pum1* and *Pum2* knockout cells, respectively (FDR < 0.1, fold change ≥ 1.5; Supplementary Data 1 and 2). To assess the reproducibility of the replicates, principal component analysis (PCA) was used to visualize the variation among different genotypes. Based on the random clustering of replicates in the PCA plot, we observed a high degree of uniformity among the replicates (Supplementary Fig. 8a). In *Pum1$^{-/-}$* cells, 740 genes showed increased mRNA levels and 392 genes showed reduced mRNA levels. Among them, several cell cycle inhibitor genes were upregulated, including *Cdkn1a/p21*, *Cdkn2d*, and *Cdkn2c* (Fig. 3a). The tumor suppressor function of these genes has been well-established[52–55]. Notably, a key oncogene, *Myc*, was downregulated. Similarly, In *Pum2$^{-/-}$* HCT116 cells, 749 and 477 genes were significantly upregulated and downregulated, respectively. The upregulated genes contain a slightly different set of cell cycle genes. For instance, *Ccnd3*, *Tgfb1* were upregulated in *Pum2$^{-/-}$* cells but not *Pum1$^{-/-}$* cells (Fig. 3b).

As PUM proteins regulate their targets largely at the post-transcriptional level, we expected a change of their targets at the protein level as well. Therefore, proteomics assays using TMT labeling were performed to identify protein changes between wild type, *Pum1$^{-/-}$*, and *Pum2$^{-/-}$* HCT116 cells (using fold change ≥ 1.2, a standard cut-off for TMT mass spectrometry because its data are nonlinearly compressed; Fig. 3c, d and Supplementary

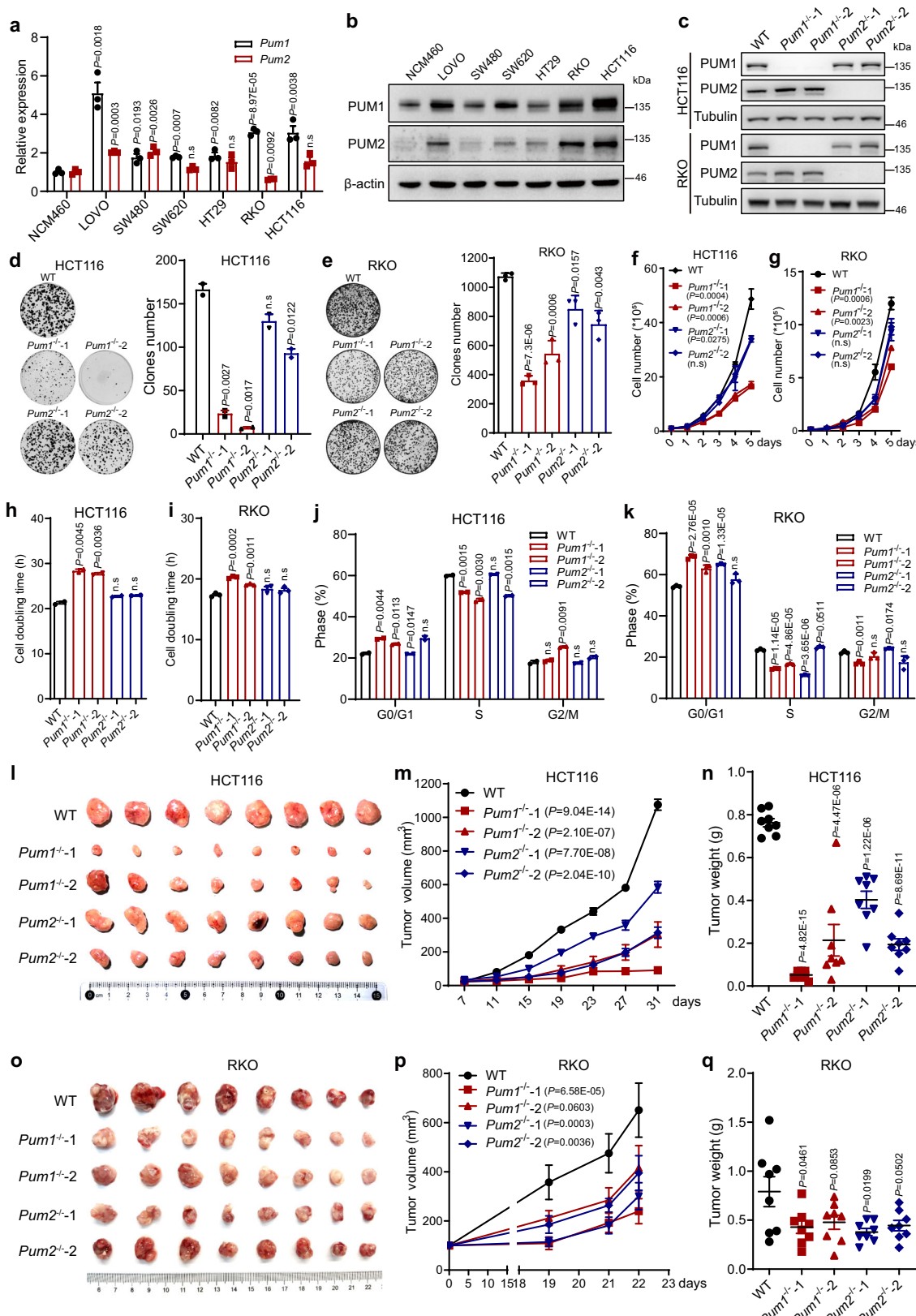

Data 3 and 4). The PCA of mass spectrometry based on normalized peptide abundance showed that biological replicates of each genotype could be clustered together (Supplementary Fig. 8b). Consistently, the mRNA and protein level of cell cycle inhibitor gene *Cdkn1a/p21* was upregulated in *Pum1*$^{-/-}$ HCT116 cells. This suggested that PUM1 may repress p21 translation in CRC. The protein levels of Wnt signaling inhibitor *Dkk1* and tumor suppressor genes *Susd2* and *Pdlim5* were also upregulated in *Pum2*$^{-/-}$ HCT116 cells, indicating the involvement of the Wnt pathway.

To analyze how much of the changes at the protein level is due to the changes at the mRNA level, because the nonlinear nature of

**Fig. 2 PUM1 and PUM2 drive colorectal cancer cell growth in vitro and tumorigenicity in vivo. a** The relative levels of *Pum1* and *Pum2* mRNA in six human colorectal cancer cell lines normalized over their levels in a human colon epithelial cell line, NCM460. Actin was used as an internal control to normalize the level of *Pum1* or *Pum2* among the duplicate samples ($n = 3$) of the same cell line. Error bars represent SEM. n.s not significant, *$P < 0.05$, **$P < 0.01$, ***$P < 0.001$ by Student's *t*-test. **b** Levels of PUM1 or PUM2 proteins in a normal colon epithelial cell line NCM460 and six human CRC cell lines. β-actin was used as a loading control. **c** Knockout efficiency of *Pum1* and *Pum2* in HCT116 and RKO cells, in which HCT116 *Pum1*$^{-/-}$-1 represents #12, HCT116 *Pum1*$^{-/-}$-2 represents #23, HCT116 *Pum2*$^{-/-}$-1 represents #10-4, HCT116 *Pum2*$^{-/-}$-2 represents #8-4, RKO *Pum1*$^{-/-}$-1 represents #3-4, RKO *Pum1*$^{-/-}$-2 represents #3-18, RKO *Pum2*$^{-/-}$-1 represents #2-19 and RKO *Pum2*$^{-/-}$-2 represents #2-27 according to numbers in Supplementary Fig. 5d, f. Tubulin was used as a loading control. **d**, **e** Representative images of colony formation assay in WT, *Pum1*$^{-/-}$ and *Pum2*$^{-/-}$ HCT116 (**d**, $n = 2$) and RKO (**e**, $n = 3$) cells. Bar graphs show the colony numbers per dish. Error bars represent SD. **f**, **g** Growth curves of WT, *Pum1*$^{-/-}$ or *Pum2*$^{-/-}$ HCT116 (**f**, $n = 4$) and RKO (**g**, $n = 3$) cells. Error bars represent SD. **h**, **i** Cell doubling time of WT, *Pum1*$^{-/-}$ or *Pum2*$^{-/-}$ HCT116 (**h**, $n = 2$) and RKO (**I**, $n = 3$) cells. Error bars represent SD. **j**, **k** Cell cycle analysis of WT, *Pum1*$^{-/-}$ or *Pum2*$^{-/-}$ HCT116 (**j**, $n = 2$) and RKO (**k**, $n = 3$) cells. Error bars represent SD. **l–n** Tumor growth (**l**), tumor volume (**m**), tumor weight (**n**) in mice injected with WT, *Pum1*$^{-/-}$ or *Pum2*$^{-/-}$ HCT116 cells ($n = 8$). Error bars represent SD. **o–q** Tumor growth (**o**), tumor volume (**p**), tumor weight (**q**) in mice injected with WT, *Pum1*$^{-/-}$ or *Pum2*$^{-/-}$ RKO cells ($n = 8$). Error bars represent SD. Source data are provided as a Source Data file.

the mass spectrometry data preludes direct comparison, we plotted the mRNA and protein log2FoldChange of *Pum1*$^{-/-}$ and *Pum2*$^{-/-}$ cells, respectively. The correlation of gene expression changes at the mRNA and protein levels for *Pum1* knockout is 0.695 (Fig. 3e) and for *Pum2* knockout 0.764 (Fig. 3f). This indicates that changes at the protein level to a large extent resulted from changes at the mRNA levels. In addition, *Pum1*$^{-/-}$ and *Pum2*$^{-/-}$ cells showed quite similar changes in gene expression, with a 0.782 correlation on the mRNA level (Fig. 3g) and 0.708 correlation on the protein level (Fig. 3h). In general, *Pum1* knockout resulted in slightly greater changes in mRNA abundance than *Pum2* knockout. On average, the fold change in mRNA abundance for the upregulated genes is 3.3 in *Pum1*$^{-/-}$ cells and 2.3 in *Pum2*$^{-/-}$ cells, and for the downregulated genes is 4.1 in *Pum1*$^{-/-}$ cells and 2.3 in *Pum2*$^{-/-}$ cells. With respect to individual genes, *Pum1*$^{-/-}$ cells also displayed a significantly greater extent of fold change than *Pum2*$^{-/-}$ cells at both the mRNA and protein levels (Supplementary Fig. 9; Wilcoxon rank sum test $P = 1.943e{-}06$ and $1.658e{-}11$ for mRNAs and proteins, respectively). KEGG pathway analyses of the changed mRNAs and proteins in *Pum1*$^{-/-}$ and *Pum2*$^{-/-}$ HCT116 cells revealed that cancer-related genes were enriched (Fig. 3i–l and Supplementary Data 5). Consistent with tumorigenic defects in *Pum1*$^{-/-}$ and *Pum2*$^{-/-}$ HCT116 cells, PUM proteins appear to regulate the expression of cancer-related genes to promote CRC growth.

To identify the direct targets of PUM1, we performed PUM1 PAR-CLIP experiments in WT HCT116 cells with *Pum1*$^{-/-}$ HCT116 cells as negative controls, which would map binding sites of PUM1 on its target mRNAs at the single-nucleotide resolution (Supplementary Fig. 10a, b). Two biological replicates were carried out for each condition, 2228 and 1961 clusters were grouped as PUM1 binding sites in two WT replicates by PARalyzer[56], respectively. The clusters showed large overlaps for PUM1 binding transcripts (Fig. 4a). We calculated the Spearman correlation of transcripts for two biological replicates of PUM1 PAR-CLIP, which showed high reproducibility (Supplementary Fig. 10c). We found that the binding sites showed the conserved TGTANATA binding motif (N represents A/C/U) (Fig. 4b). This motif was mostly present in the 3′UTRs of the mRNAs, with the second abundant region being the CDS (Fig. 4c). In addition to mRNAs, we identified the long noncoding RNA–NORAD (Supplementary Fig. 10d and Supplementary Data 6), a known target of PUM1 containing 17 binding sites[27]. Targets identified by PAR-CLIP were validated by qPCR (Fig. 4d, e). Given the observed G1/S transition delay in the *Pum1*-deficient cells, we mainly focused on the genes that are responsible for G1 progression and found that an upregulated gene in *Pum1*$^{-/-}$ HCT116 cells, p21/*Cdkn1a*, was enriched in PUM1 PAR-CLIP and contained a canonical PRE site

(TGTAnATA) presented in 3′UTR (Fig. 4f). Moreover, cell cycle and cancer pathways such as ErbB signaling pathway, p53 signaling pathway and CRC pathway are also enriched among the PUM1-direct targets (Fig. 4g and Supplementary Data 7).

To explore how PUM1 regulates its targets, we further investigated 1132 mRNAs whose expression were changed in *Pum1*$^{-/-}$ HCT116 cells and contained PUM1 binding sites as indicated by the PAR-CLIP analysis. We also analyzed 1968 proteins whose expression were changed in *Pum1*$^{-/-}$ HCT116 cells. We found that 15 changed mRNAs and 48 changed proteins were identified as targets of PUM1 (Fig. 4h–k). KEGG analysis on these 15 mRNAs and 48 proteins as well as on the target genes of PUM1 PAR-CLIP revealed that they are enriched in cell cycle and cancer pathways, etc. (Fig. 4l, m and Supplementary Data 7). The changes of 17 mRNAs in *Pum1*$^{-/-}$ HCT116 cells were all validated by RT-qPCR (Fig. 4n). These results indicate that PUM1 is required for the growth of CRC cells mainly by directly regulating cancer-related and cell cycle-related genes.

**PUM1 directly represses p21 to regulate CRC growth.** Among the PUM1 direct targets as revealed by PAR-CLIP-seq experiments, *p21* plays an important role in G1/S phase transition. In addition, *p21* expression is negatively correlated with the overall survival rate of CRC and, to a less extent, disease-free survival (Supplementary Fig. 11a). We therefore focused on the regulation of PUM proteins towards *p21*. RNA deep sequencing and protein mass spectrometry revealed that *p21* was repressed by PUM1 at both RNA and protein levels. We further confirmed these results by RT-qPCR and western blotting, which indicated that both RNA and protein levels of *p21* were increased in *Pum1*$^{-/-}$ cells (Fig. 5a, b). However, the increase of p21 RNA and protein level in *Pum2*$^{-/-}$ was not as significant as that in *Pum1*$^{-/-}$ cells (a similar phenomenon was observed in HCT116 cells transfected with siPum1 and/or siPum2, data not shown), consistent with this, our RNA co-immunoprecipitation (RIP) experiments indicated that *p21* mRNA was bound strongly (7.4-fold enrichment) by PUM1, while weaker (1.9-fold enrichment) by PUM2 (Fig. 5c, d). Therefore, we focus our effort on PUM1 regulation of *p21*.

To test whether PUM1 negatively regulates the stability of *p21* mRNA, we used actinomycin to inhibit transcription and then examined the mRNA level in HCT116 cells (Fig. 5e). The stability of *p21* mRNA was increased in *Pum1*$^{-/-}$ HCT116 cells, but not significantly in *Pum2*$^{-/-}$ HCT116 cells. These results demonstrated that PUM1 repressed *p21* by negatively regulating its mRNA stability, but PUM2 had little effect on *p21* mRNA stability. We next performed dual luciferase reporter assay to evaluate the PRE contribution in PUM1-mediated *p21* regulation. 3′UTR with mutated PRE showed a significant increase of

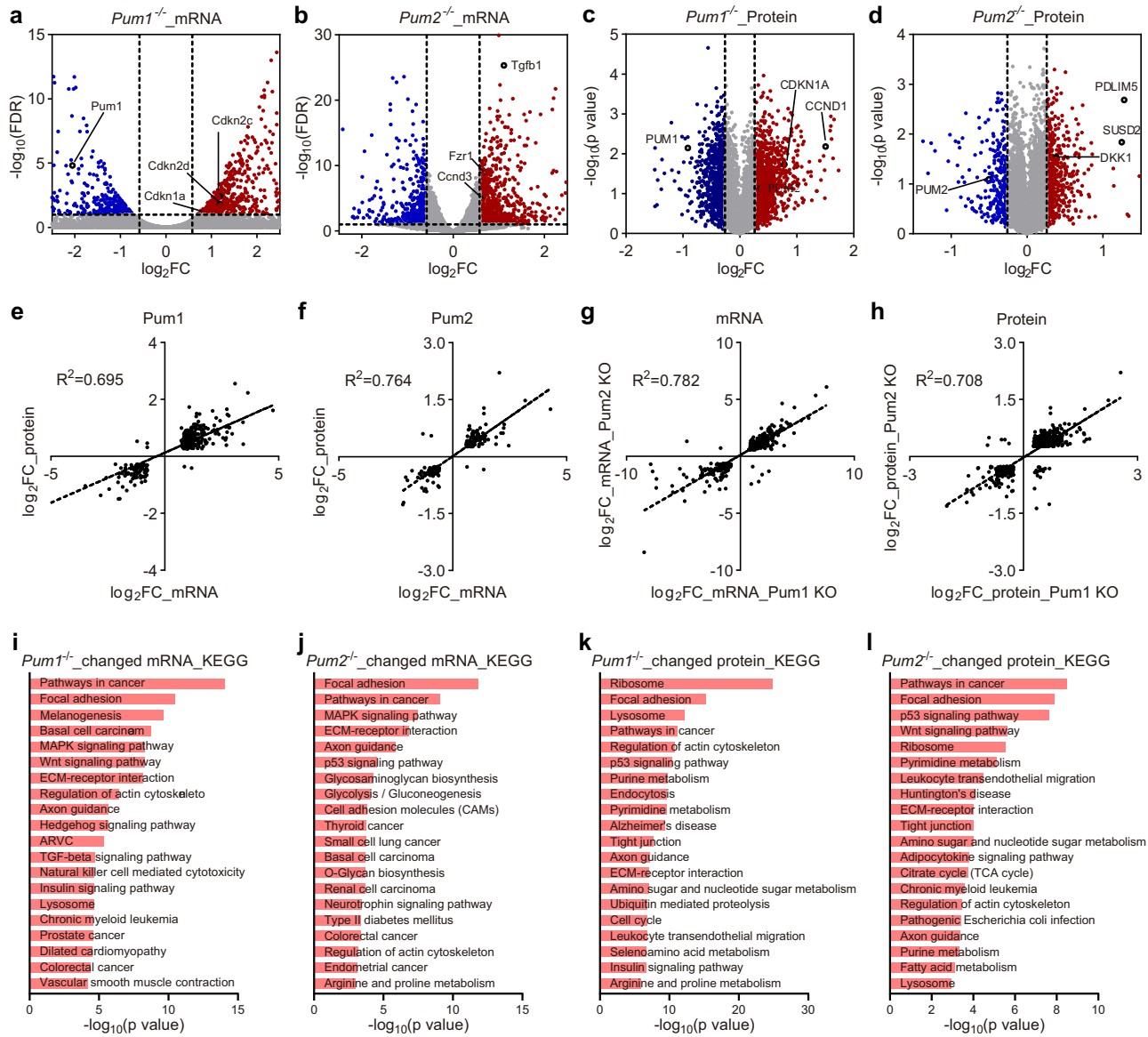

**Fig. 3 Cancer-related genes are enriched among genes affected in *Pum1*⁻/⁻ HCT116 cells. a–d** Volcano plot showing the significantly upregulated (red dots) and downregulated (blue dots) mRNA (**a**, **b**) or protein (**c**, **d**) in *Pum1*⁻/⁻ and *Pum2*⁻/⁻ cells, including those in cell cycle, *Pum1*, and *Pum2*, as labeled. For mRNA, cutoff criteria are false discovery rate (FDR) ≤ 0.1, fold change (FC) ≥ 1.5 or FC ≤ 1.5. For protein, cutoff criteria are FC ≥ 1.2 or FC ≤ 1.2. **e**, **f** Scatterplot of log2FoldChange at mRNA and protein levels upon *Pum1* (**e**) or *Pum2* (**f**) knockout. **g**, **h** Expression changes in *Pum1*⁻/⁻ cells correlate well with that in *Pum2*⁻/⁻ cells at both mRNA (**g**) and protein (**h**) levels. **i**, **j** KEGG analysis of differentially expressed mRNAs in *Pum1*⁻/⁻ (**i**) and *Pum2*⁻/⁻ (**j**) HCT116 cells. **k**, **l** KEGG analysis of differentially expressed proteins in *Pum1*⁻/⁻ (**k**) and *Pum2*⁻/⁻ (**l**) HCT116 cells. Source data are provided as a Source Data file.

luciferase signal over wild-type 3′UTR in the presence of Pum1 expression construct, to a level similar to the control construct without *p21* 3′UTR (Supplementary Fig. 11b). These results indicates that PUM1 inhibits p21 expression via the PRE in the 3′ UTR.

Given the repression of PUM1 towards *p21* and the possible involvement of *p21* in CRC, we investigated whether the deletion of PUM1 would reduce cell proliferation and delay G1-S transition by directly upregulating *p21* expression. We mutated the PRE in the *p21* gene by knocking in a mutated PRE donor sequence in the genome of HCT116 cells (Fig. 5f). DNA sequencing showed that we generated a mutant *p21* PRE (*p21* PRE^mut/mut) by converting TGTA to ACAT (Fig. 5g). As expected, mutating the *p21* PRE reduced the binding of PUM1 to *p21* mRNA (Fig. 5h), and led to increased expression of *p21*

mRNA and protein (Fig. 5i, j). These results indicated that *p21* PRE^mut/mut relieved *p21* mRNA from repression by PUM1.

We next analyzed cell growth and cell cycle of *p21* PRE^mut/mut cells. Again, as expected, *p21* PRE^mut/mut cells showed decreased colony formation, reduced cell proliferation (Fig. 5k–n), and increased proportion of cells in G0/G1 phase (Fig. 5o). Together, these results demonstrated that *p21* mRNA is a direct and main target of PUM1 in facilitating CRC cell growth.

**Nanoparticle-encapsulated PUM siRNA partially inhibits the development of CRC.** To explore the possibility of using PUM1 and PUM2 as targets for anti-cancer treatment, we further investigated the in vivo antitumor activity of siRNA-loaded nanoparticles using the orthotopic colon cancer models (Fig. 6a)[57]. Two colorectal tumor models, HCT116-luc and

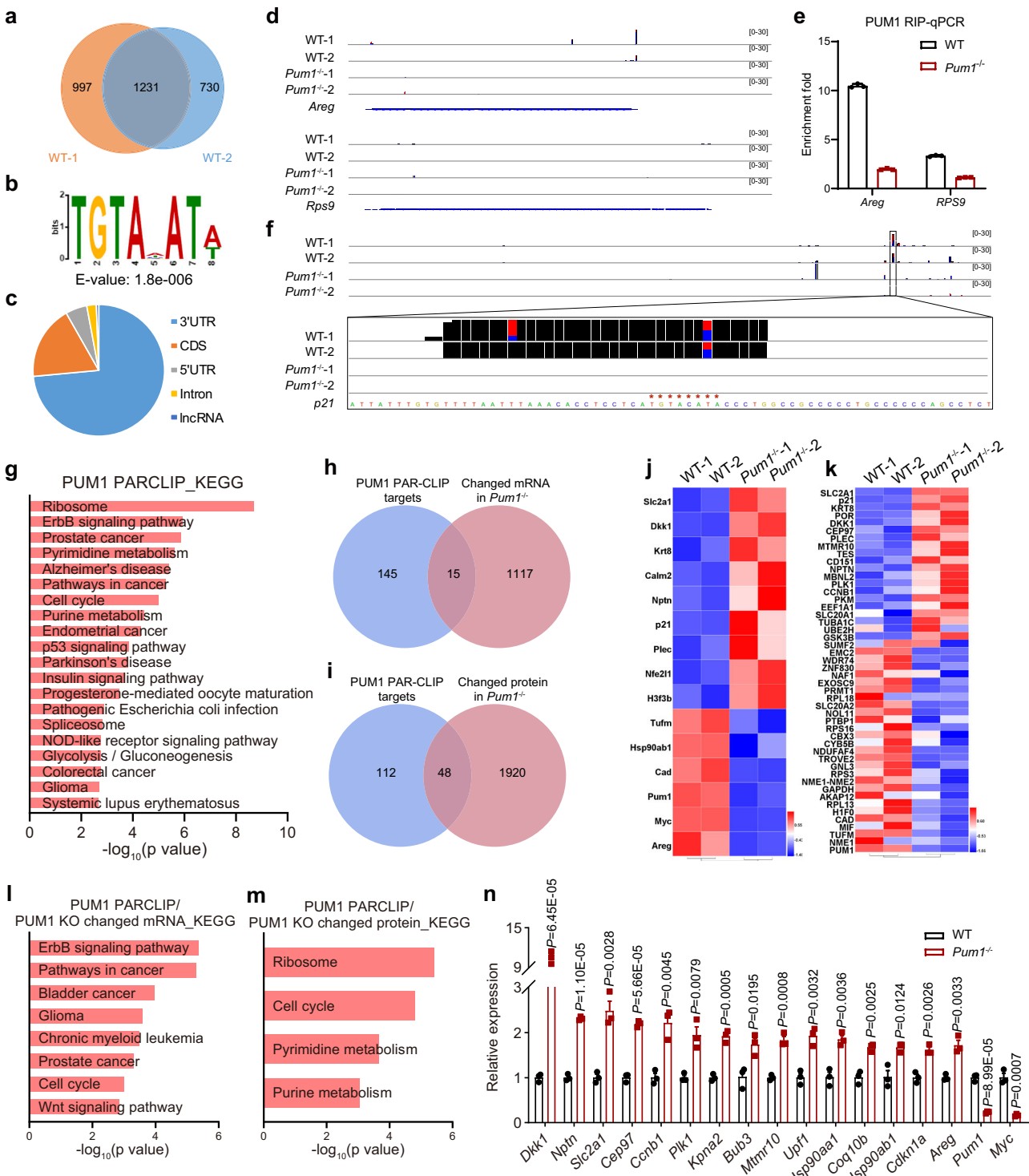

**Fig. 4 PAR-CLIP-seq identification of PUM1 targets. a** Reproducibility of two biological replicates for PUM1 photoactivatable ribonucleoside-enhanced crosslinking and immunoprecipitation (PAR-CLIP). **b** de novo discovery of the binding motif for PUM1 by MEME analysis. **c** Distribution of the PUM1-binding sites in different gene regions. **d** PUM1 PAR-CLIP peaks in the *Areg* and *Rps9* transcripts in WT and *Pum1*−/− HCT116 cells. *Areg* is a target of PUM1 but *Rps9* is not a target. **e** PUM1 RNA immunoprecipitation (RIP)-qPCR validation of the target *Areg* and nontarget *Rps9* (*n* = 3). Error bars represent SEM. **f** PUM1 PAR-CLIP peaks in the *p21* transcript in WT and *Pum1*−/− HCT116 cells. TGTACATA is the binding site of PUM1. **g** KEGG analysis of PUM1 PAR-CLIP targets in HCT116 cells. **h, i** A Venn diagram showing the overlap of PUM1 PAR-CLIP targets with differentially expressed mRNAs (**h**) and proteins (**i**) in *Pum1*−/− HCT116 cells. **j, k** Heatmap showing the expression of mRNAs (**j**) and proteins (**k**) of PUM1 PAR-CLIP targets in *Pum1*−/− HCT116 cells. **l, m** KEGG analysis of PUM1 PAR-CLIP target mRNAs (**l**) or proteins (**m**) that are changed in *Pum1*−/− HCT116 cells, demonstrating enrichment of genes involved in cell cycle. **n** RT-qPCR validation of overlap genes of PUM1 PAR-CLIP targets with changed mRNAs in *Pum1*−/− HCT116 cells (*n* = 3). Error bars represent SEM. Source data are provided as a Source Data file.

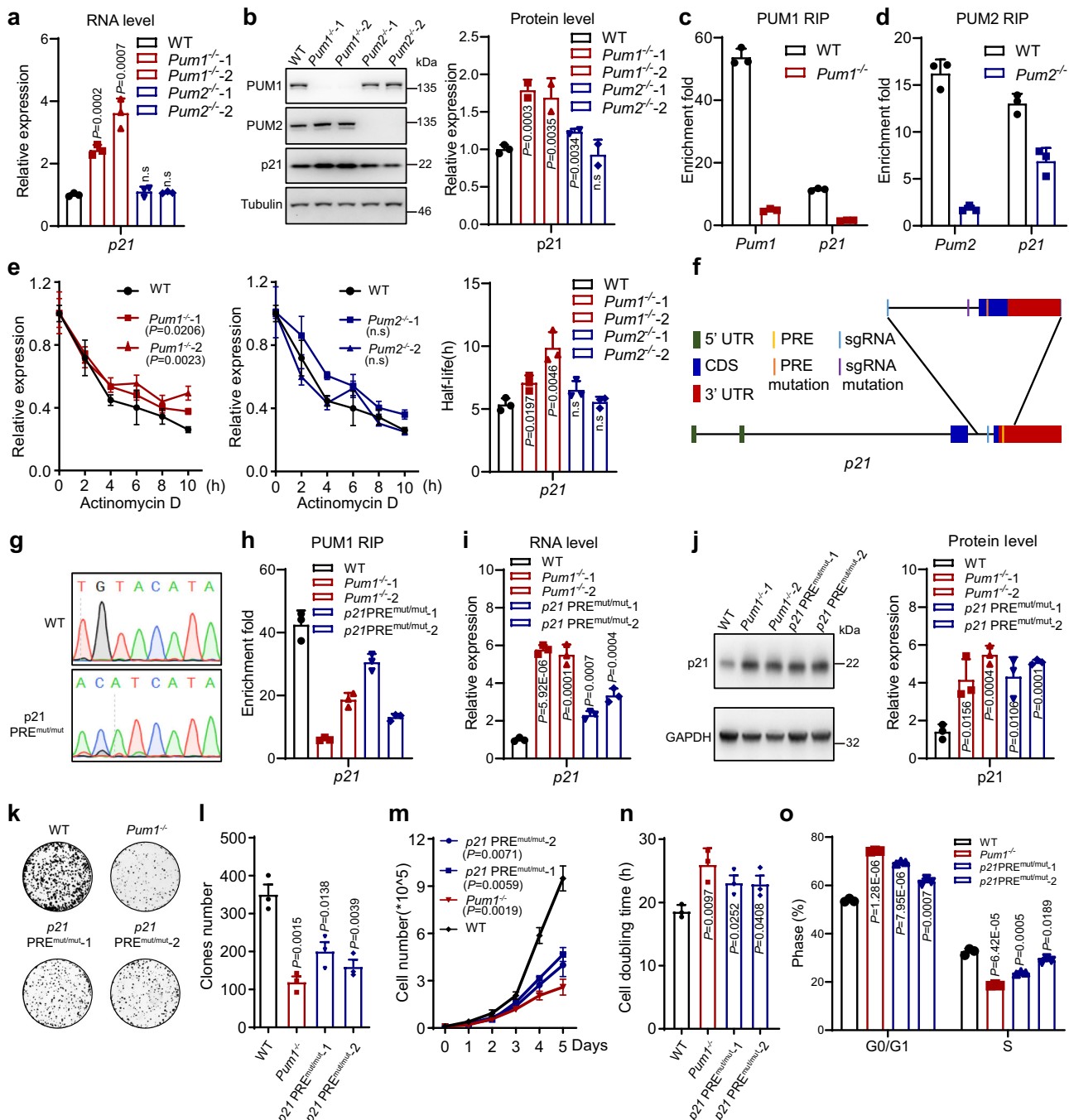

**Fig. 5 PUM1 regulates CRC growth by directly repressing *p21*. a** The mRNA levels of *p21* in WT, *Pum1⁻/⁻* and *Pum2⁻/⁻* HCT116 cells (*n* = 3) as assayed by RT-qPCR. Actin was used as an internal control. Error bars represent SEM. **b** Western blot showing the p21 protein levels in WT, *Pum1⁻/⁻* and *Pum2⁻/⁻* HCT116 cells. Tubulin was used as a loading control. The bar graph shows the quantification of the western blot (*n* = 3). Error bars represent SD. **c, d** Enrichment of *p21* mRNA by PUM1 RIP (**c**) or PUM2 RIP (**d**) in WT, *Pum1⁻/⁻* or *Pum2⁻/⁻* HCT116 cells (*n* = 3). Error bars represent SEM. **e** The *p21* mRNA levels in WT, *Pum1⁻/⁻* or *Pum2⁻/⁻* HCT116 cells treated with actinomycin D (10 mg/ml) for the indicated period (*n* = 3). Error bars represent SEM. The bar graph shows the calculated *p21* mRNA half-life (*n* = 3). Error bars represent SEM. **f** Schematic diagram of generating *p21* PRE mutations in HCT116 cells. **g** DNA sequencing results of PRE in WT and *p21* PRE^mut/mut^ cells. **h** Enrichment of *p21* mRNA by PUM1 RIP in WT, *Pum1⁻/⁻*, and *p21* PRE^mut/mut^ HCT116 cells (*n* = 3). Error bars represent SEM. **i** The mRNA of *p21* in WT, *Pum1⁻/⁻* and *p21* PRE^mut/mut^ HCT116 cells (*n* = 3). Actin mRNA was used as an internal control. Error bars represent SEM. **j** Western blot showing the p21 protein levels in WT, *Pum1⁻/⁻* and *p21* PRE^mut/mut^ HCT116 cells. GAPDH was used as a loading control. The bar graph shows the quantification of the western blot (*n* = 3). Error bars represent SD. **k** Images of colony formation assay in WT, *Pum1⁻/⁻* and *p21* PRE^mut/mut^ HCT116 cells. **l** Bar graph showing the colony numbers in each view in **k** (*n* = 3). Error bars represent SD. **m** Growth curve of WT, *Pum1⁻/⁻*, and *p21* PRE^mut/mut^ HCT116 cells (*n* = 3). Error bars represent SD. **n** Cell doubling time of WT, *Pum1⁻/⁻* and *p21* PRE^mut/mut^ HCT116 cells (*n* = 3). Error bars represent SD. **o** Cell cycle analysis of WT, *Pum1⁻/⁻*, and *p21* PRE^mut/mut^ HCT116 cells (*n* = 3). Error bars represent SD. Source data are provided as a Source Data file.

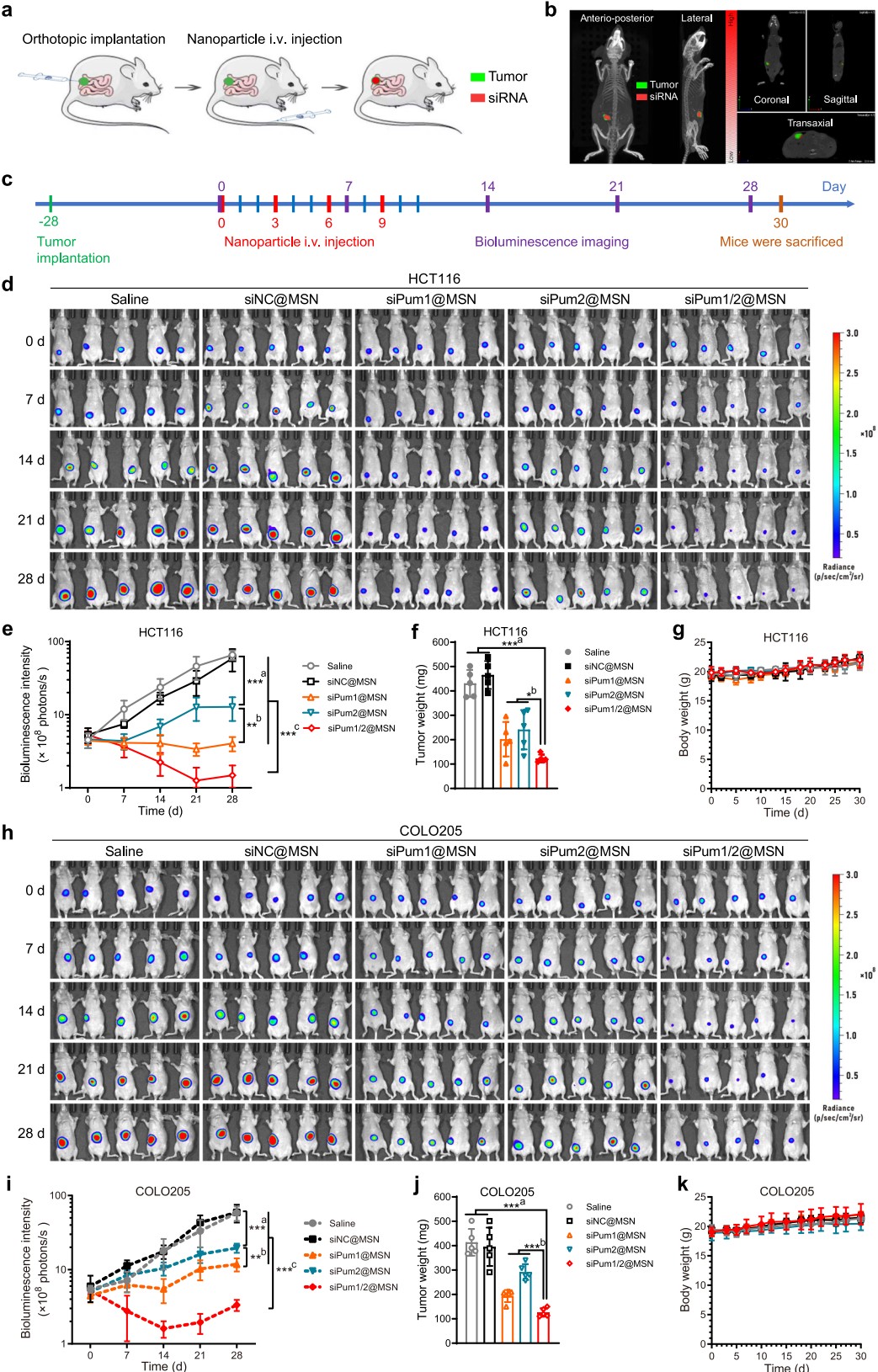

COLO205-luc, were used in this study. The established poly-ethylenimine (PEI)-coated, glutathione (GSH)-responsive, mesoporous silica nanoparticles (MSN) were adopted for anti-*Pum1* and *Pum2* siRNA delivery[57]. Empty MSN without PEI coating were spherical with a diameter of ~90 nm as shown in transmission electron microscopy (TEM; Supplementary

Fig. 12a). The hydrodynamic diameter of empty MSN without PEI determined by dynamic light scattering (DLS) were 100 nm (Supplementary Fig. 12b), a little larger than the dimension acquired by TEM. PEI coating and siRNA loading slightly increased the nanoparticle size. The zeta potential reversed from negative (−33 mV) for empty MSN to highly positive (34 mV) for

**Fig. 6 In vivo therapy using siRNA-loaded MSN in mice bearing orthotopic HCT116 or COLO205 colorectal tumor. a** Schematic diagram of orthotopic implantation and nanoparticle therapy. The mouse icon was from https://smart.servier.com/. **b** Tumor targeting of the siRNA-loaded nanoparticles was observed using IVIS Spectrum CT multimodal imaging system. Luciferase labeled tumor cells and the Cy5 labeled siRNA@MSN after 24 h i.v. injection. The green signals indicated the orthotopic colorectal tumor and the red fluorescence signals denoted the Cy5-labeled nanoparticles. The nanoparticle distribution in the tumors observed from the perspective, coronal, sagittal, and transaxial directions were separately shown. **c** Treatment regimen. **d**, **h** In vivo bioluminescence imaging of HCT116-luc (**d**) and COLO205-luc (**h**) tumor at the indicated time. **e**, **i** Tumor growth profiles were obtained through quantifying the bioluminescence in panel **d** and **h** ($n = 5$). Panel **e**: **a** siPum2@MSN versus saline ($P = 4.22E{-}06$) or siNC@MSN ($P = 0.0001$). **b** siPum1@MSN versus siPum2@MSN ($P = 0.0030$). **c** siPum1/2@MSN versus all other groups: saline ($P = 4.52E{-}07$), siNC@MSN ($P = 0.0002$), siPum1@MSN ($P = 0.0006$), siPum2@MSN ($P = 0.0006$). Panel **i**: **a** siPum2@MSN versus saline ($P = 0.0002$) or siNC@MSN ($P = 0.0006$). **b** siPum1@MSN versus siPum2@MSN ($P = 0.0013$). **c** siPum1/2@MSN versus all other groups: saline ($P = 1.14E{-}05$), siNC@MSN ($P = 5.2E{-}05$), siPum1@MSN ($P = 5.2E{-}05$), siPum2@MSN ($P = 1.22E{-}06$). Error bars represent SD. **f**, **j** Tumor weight on the last day (Day 30) of the test ($n = 5$). Panel **f**: **a** siPum1/2@MSN versus saline ($P = 2.21E{-}06$) or siNC@MSN ($P = 1.25E{-}06$). **b** siPum1/2@MSN versus siPum1@MSN ($P = 0.0403$) or siPum2@MSN ($P = 0.0122$). Panel **j**: **a** siPum1/2@MSN versus saline ($P = 3.82E{-}06$) or siNC@MSN ($P = 7.24E{-}05$). **b** siPum1/2@MSN versus siPum1@MSN ($P = 0.0011$) or siPum2@MSN ($P = 8.25E{-}06$). Error bars represent SD. **g**, **k** Mice body weight ($n = 5$). Error bars represent SD. Source data are provided as a Source Data file.

PEI-coated siRNA-loaded MSN (Supplementary Fig. 12c). siRNA molecules were efficiently released in the presence of 5 mM GSH-contained PBS, which well mimicked the environment of cancer cell cytoplasm (Supplementary Fig. 12d)[58,59]. Tumor targeting of the siRNA-loaded nanoparticles via the enhanced permeability and retention (EPR) effect[60] was observed using IVIS Spectrum CT multimodal imaging system (Fig. 6b). The Cy5 labeled siRNA@MSN specifically targeted luciferase-labeled tumor cells 24 h after tail vein injection, demonstrating the feasibility of our approach.

We implanted HCT116 and COLO205 tumor cells subcutaneously into 4-week-old male BALB/c nude mice. Twenty-eight days after the implantation, mice showed comparable bioluminescence intensity of the tumors and were divided into five groups (five mice/group). These mice were intravenously injected with nanoparticle formulations every three days for four times (day 0, 3, 6, 9; Fig. 6c). Bioluminescence imaging was used to monitor tumor growth every week (day 7, 14, 21, 28; Fig. 6d, h). For the HCT116 tumor model, non-specific siRNA control (siNC@MSN) did not delay tumor growth. In contrast, injection of nanoparticles loaded with anti-*Pum1* siRNA (siPum1@MSN) or anti-*Pum2* siRNA (siPum2@MSN) dramatically inhibited tumor growth. Particularly, mice treated with both anti-*Pum*1 and anti-*Pum*2 siRNAs (siPum1/2@MSN) showed the strongest antitumor effect (Fig. 6e). At the end of the experiment (day 30), the mice were sacrificed and tumors were excised and weighed. The tumors of mice treated with siPum1/2@MSN were 3.7–4.0-fold smaller than those treated with controls (Fig. 6f). This reduction in tumor weight was not due to overt toxicity, because the body weights of all groups had a similar increase during the treatment (Fig. 6g).

Histological examination and immunochemistry staining in siNC@MSN and siPum1/2@MSN groups were performed at day 11. At this time point, the expression of PUM1 and PUM2 was decreased in siPum1/2@MSN samples (Supplementary Fig. 13), validating the effect of the nanoparticle-mediated knock down of PUM1 and PUM2. The PUM1 and PUM2 knockdown did not cause any detectable tissue damage, as examined by H&E staining (Supplementary Fig. 14), indicating the siPum1/2@MSN treatment did not cause non-specific tissue toxicity. Expectedly, the expression of the p21 protein was upregulated in tumors treated with siPum1/2 (Supplementary Fig. 13). H&E staining further revealed a significant decrease of adenomas when *Pum1* and *Pum2* were knocked down (Supplementary Fig. 15). In addition, Ki-67 staining indicated that the proliferation of the tumor cells in the siPum1/2@MSN group was inhibited compared with siNC@MSN group (Supplementary Fig. 13). These data indicate the important function of PUM1 and PUM2 in tumor progression.

Because metastasis is the main cause of death in patients with CRC[61], with nearly 60% of CRC patients having metastasis in the liver[62], we examined the anti-metastatic effect of the treatment. On day 30, major organs (heart, liver, spleen, lung, and kidney) from all the mice were excised for anti-metastasis evaluation using bioluminescence imaging (Supplementary Fig. 16). In the saline control group, all mice had metastases in the liver. Two of the five mice also had spleen and lung metastases. No metastases were observed in kidney. The mice in the siNC@MSN control group had similar metastasis pattern to that of the saline group. In contrast, the treatment of siPum1@MSN or siPum2@MSN significantly inhibited metastases in liver, spleen, and lung. Specifically, we observed no metastases in spleen for the siPum1@MSN group and in lung for the siPum2@MSN group. Furthermore, the siPum1,2@MSN group showed no metastases in any of the examined organs, representing the best anti-metastasis effect. Similar antitumor effect was also observed in the COLO205 tumor-bearing mice model (Fig. 6i–k and Supplementary Fig. 17). Taken together, these data indicate that *Pum1* or/and *Pum2* siRNA delivery by nanoparticles prevented further growth of CRC in vivo, and that *Pum1/2* are potential targets for the treatment of CRC.

## Discussion

In this paper, we reported the requirement of PUM1 and PUM2, two members of the human PUF protein family, for the initiation and progression of CRC. Furthermore, we discovered that such a cancer cell growth-promoting function was in part achieved by inhibiting the expression of *p21*, a negative regulator of cell cycle. Finally, we showed that treating CRC models by intravenous injection of nanoparticle-packaged anti-PUM siRNAs effectively reduced tumor growth. These findings provide insights into a new molecular mechanism important for CRC tumorigenesis as well as a potentially new approach for treating CRC.

In recent years, several studies have reported that human PUM proteins play an important oncogenic role. However, the function of PUM proteins in different tumors is different, and there is a debate on whether PUM proteins are oncogenes or tumor suppressors. Kedde et al. reported that PUM1 and classical oncogenic miRNAs miR-221 and miR-222 co-repress the expression of the cell cycle inhibitory protein p27 in human breast cancer cells and HEK293 cells[63]. Because the knockdown of PUM1 reduced the number of cells in S phase and inhibit cell proliferation, the authors suggested that PUM1 may promote tumorigenesis in breast cancer. In non-small cell lung cancer, tumor suppressor miR-340 directly binds to the 3′UTR of *Pum1* and *Pum2* mRNA to inhibit their expression, thus reducing PUM1 and PUM2 that are required for the miR-221/222 interaction with the *p27* 3′

UTR[64]. In addition, recent studies in leukemia cells have shown that PUM proteins can directly bind to the mRNA of the transcription factor FOXP1 to increase the expression level of FOXP1 protein, thereby increasing the proliferation of hematopoietic stem cells and myeloid leukemia cells[24]. Moreover, PUM1 and PUM2 inhibit kinase activator GC-32 by deadenylation to promote the growth of EB virus immortalized B cells[65]. Finally, PUM1 can promote the development of ovarian cancer[23]. All these studies are similar to our findings that knockout or knockdown of PUM1 and PUM2 block cell G1/S phase transition by upregulating p21 expression (Figs. 2 and 5), thus significantly inhibits the proliferation of CRC cells. All of the above studies have revealed the requirement of PUM proteins for oncogenesis.

However, Miles et al. found that PUM1 and PUM2 co-inhibit the expression of oncogene E2F3 with miRNA in bladder cancer[28]. In this type of cancer, even though the level of PUM1 or PUM2 protein itself did not change significantly, the level of the miRNAs (miR-503, miR125b) synergizing with PUM proteins was decreased significantly, which allowed oncogenesis. In triple-negative breast cancer cells, PREs in the 3′UTR of oncogene c-jun were deleted due to alternative polyadenylation, which promotes cancer. These studies indicate an onco-suppressor role of PUM proteins[66]. Thus, PUM proteins have opposite functions in different cancers. This difference may be due to their targeting of different genes or partnering with different proteins in different types of cancer.

An important finding of our study is that p21 mRNA is a direct and main target of PUM1. In order to obtain more credible PUM1 target mRNAs, we used relatively stringent criteria in our CLIP data analysis, including parameter adjustments for PARalyzer, binding sites intersection from two biological replicates, and presumptive background signal removal. This results in 160 PUM1-bound target mRNAs. When we reduced stringency, more direct target genes were recovered from the CLIP, including an increased number of targets that overlapped with genes showing differentially expressed mRNAs or proteins in the Pum1[−/−] cells. We limit ourselves to the 160 targets to avoid false positive targets. It was surprising to us as well that only 15 and 48 of the >1000 genes whose expression was changed by Pum1[−/−] cells were bound by PUM1. This indicates that most of the differentially expressed genes are indirect targets of PUM1. Meanwhile, PUM1 may binds to some mRNAs without any significant regulatory function. In addition, it is possible that both PUM1 and PUM2 binding are needed to generate an easily detectable regulatory effort on some other of the 160 PUM1-target mRNAs, since PUM1 and PUM2 bind to overlapping set of targets[17]. Finally, it is possible that PUM1 might bind to some mRNAs without any significant regulatory function.

Among the 160 targets, p21 is the only one associated with the cell cycle among the direct PUM1-target genes that are regulated by PUM1 at both mRNA and protein levels (Fig. 5). A previously study reported that PUM1 and PUM2 positively regulated FOXP1, and FOXP1 mediates the growth-promoting activity of the PUM proteins by repressing the expression of p21 in human hematopoietic stem/progenitor cells and leukemic cells[24]. Their results suggested that PUM1 negatively regulates p21, which was consistent with our finding. However, we provide strong evidence indicating that PUM1 directly binds to and represses p21 via the 3′UTR PRE of p21 mRNA instead of through FOXP1 or other factors. Our results showed that PUM1 significantly downregulates both mRNA and protein levels of p21 by promoting p21 mRNA turnover, possibly through the well-known pathway in which PUM proteins interact with the CCR4-POP2-NOT deadenylase complex to inhibit mRNA stability[67]. This eventually results in CRC proliferation and shortened G1/S transition.

Based on our results, we propose the following model to illustrate the function of PUM proteins in human CRC: PUM1 is highly expressed in CRC and bind to p21 mRNA to reduce its expression. This relieves p21 suppression of cell cycle and cell growth mechanisms, allowing CRC progression. Either Pum1 deletion or p21 RPE mutation abolishes PUM repression of p21, which results in CRC growth defects and delayed G1/S transition.

PUM proteins are widely expressed in many tissues[20,68,69], so in order to evaluate whether a PUM protein can be used as a target for cancer treatment, one needs to determine whether such treatment will cause broad side effects. Remarkably, Pum1;Pum2 double knockout in small intestinal epithelial cells did not cause any detectable effect on intestinal homeostasis and function in our mouse CRC model (Supplementary Fig. 2). Instead, we found that conditional knockout of Pum1 and Pum2 effectively blocked the occurrence and development of colorectal tumors (Fig. 1e–i). Perhaps most importantly, when we treated colon orthotopic implant tumors by tail vein injection of nanoparticles encapsulated with anti-Pum1/2 siRNAs, these siRNAs prevented the further growth of colorectal tumors without obvious effects on other organs (Figs. 6g and S14). This might be because PUM proteins are expressed in CRC at higher levels than in normal tissues, and are therefore more sensitive to Pum knockdown and/or because nanoparticles are highly accumulated in these tumor cells. Our results point to the feasibility of using PUM proteins as targets for CRC treatment. A systematic examination of the biodistribution of Pum1/2 siRNA loaded nanoparticles, and the levels of Pum1 and Pum2 in other tissues in the future will further test this feasibility.

## Methods

**Mice**. Male or female BALB/c nude mice were purchased from Shanghai Lingchang Biotechnology Co., Ltd (Shanghai, China). All animals were housed and maintained in pathogen-free conditions and allowed free access to food and autoclaved water ad libitum in a 12 h light/dark cycle, with room temperature at $21 \pm 2\,°C$ and humidity between 45 and 65%. All animal experiments were performed in compliance with the Guide for the Care and Use of Laboratory Animals and approved by the Institutional Biomedical Research Ethics Committee of the ShanghaiTech University or Shanghai Jiao Tong University School of Medicine (SJTU-SM).

**Cell culture**. HCT116, RKO, COLO205, LOVO, SW480, SW620, HT29 cells were purchased from the American Type Culture Collection (ATCC) and cultured according to the culture methods of ATCC. Colon normal immortalized epithelial cell line NCM460 was obtained from In Cell (San Antonio, TX) and cultured according to the method of manufacturer.

**Cell cycle assay**. For each sample, $2–5 \times 10^6$ cells were collected by centrifugation at $400 \times g$ for 5 min at $4\,°C$ and washed twice with 1 ml ice cold PBS. Supernatant was discarded and cell pellet was resuspended with 250 μl ice cold PBS. Seven hundred and fifty microliter absolute ethyl alcohol (pre-cooled at $−20\,°C$) was slowly added into the suspension drop by drop to 75% final concentration on low speed vortex. Cell suspension was fixed at $4\,°C$ for overnight or for 4 h at $−20\,°C$. Cells were washed once with 1 ml ice cold PBS, collected by centrifugation at $1000 \times g$ for 5 min at $4\,°C$, stained by 400 μl PI Staining Solution/RNase solution for 30–60 min in the dark at $4\,°C$, and then analyzed by flow cytometry. The maximum excitation wavelength is 488 nm. Data analyses were performed using Modifit LT 5.0.

**Apoptosis assay**. HCT116 or RKO cells were stained with the FITC Annexin V apoptosis detection kit (556547, BD Biosciences) or PE Annexin V apoptosis detection Kit (559763, BD Biosciences) according to the manufacturer protocol and analyzed early-stage and late-stage apoptosis by FACS (FACS AriaTM IIII, BD Biosciences). Data analyses were performed using CytoExpert (version 2.0).

**Cell growth curve and colony formation**. For growth curve assay, a total of $1 \times 10^4$ HCT116 or RKO cells per well were seeded in 12-well plate, triplicate wells were seeded. Cells were counted every 24 or 48 h for a total of 5–8 days. For colony formation assay, a total of 1000 HCT116 or RKO cells per well were seeded in 6-well plate and triplicate wells were seeded. After 10 days, cells were stained with Crystal Violet Staining Solution (C0121, Beyotime).

**RNA extraction and real-time PCR**. Total RNA from cultured cells of desired genotypes was extracted with TRIzol reagent (Thermo, 15596026) and genomic

DNA was removed using DNA-free™ Kit DNase Treatment and Removal Reagents (Thermo, AM1906) according to the manufacturer's protocol. Total RNA was then converted to cDNA using the High-Capacity cDNA Reverse Transcription Kit (Thermo, 4368814). Quantitative real-time PCR reactions were performed according to the protocol of the iTaq Univer SYBR Green Supermix (Bio-Rad, 1725125). The same PCR program was used for all quantifications: 95 °C for 3 min, 95 °C for 10 s, 55 °C for 10 s, and 72 °C for 30 s. This cycle was repeated 40 times, followed by melting curve measurement. The sequence of primers used for mRNA are list in Supplementary Table 1. Primers of the actin gene were designed as a real-time PCR control. ΔΔCt method was used for normalized expression. Data analysis was performed in accordance with MIQE guidelines[70].

**Western blot**. Western blot analysis was performed according to our previous method[71]. The following antibodies were used: Recombinant Anti-Pumilio 1 antibody [EPR3795] (1:1000, Abcam, Cat#: ab92545), Recombinant Anti-Pumilio 2 antibody [EPR3813] (1:1000, Abcam, Cat#: ab92390), Recombinant Anti-Lamin A + Lamin C antibody [EPR4100]—Nuclear Envelope Marker (1:1000, Abcam, Cat#: ab108595), p21 Waf1/Cip1 (12D1) Rabbit mAb (1:1000, cell signaling technology, Cat#: 2947S), α-Tubulin (11H10) Rabbit mAb (1:1000, cell signaling technology, Cat#: 2125S), GAPDH (14C10) Rabbit mAb (1:1000, cell signaling technology, Cat#: 2118S), β-Actin (13E5) Rabbit mAb (1:1000, cell signaling technology, Cat#: 4970S) and HRP-conjugated Affinipure Goat Anti-Rabbit IgG(H + L) (1:5000, Proteintech, Cat#: SA00001-2).

**RNA immunoprecipitation (RIP) and statistical analysis**. For each condition in each biological repeat, cells in a 10 cm dish reaching about 90% density were homogenized in RIP lysis buffer (20 nM Tris-HCl, 150 mM NaCl, 1% IGEPAL 630 (SIGMA, I8896), 1 mM EDTA, 0.5 Mm DTT, cOmplete™ EDTA-free Protease Inhibitor Cocktail Tablets (MERCK, 4693132001), PhosStop Tablets (MERCK, 4906837001) and SUPER In RNase Inhibitor (Thermo, AM2694), spun at 13,000×g for 15 min to remove the debris. Ten microliter of empty Dynabeads Protein A (Thermo, 10002D) was added to the lysate and incubated for 30 min for pre-clearing. Recombinant Anti-Pumilio 1 antibody [EPR3795] (0.25 mg/ml, Abcam, ab92545) or Recombinant Anti-Pumilio 2 antibody [EPR3813] (0.25 mg/ml, Abcam, ab92390) was incubated with the pre-cleared lysate overnight at 4 °C, and 50 μl of Dynabeads were added. The incubation of lysates-antibody-Dynabeads was done with rotation at 4 °C for 4 h. Five percent of the lysate-antibody-Dynabeads was saved as input sample. Next, the lysates-antibody-Dynabeads were washed three times with the RIP wash buffer (20 mM Tris-HCl, 200 mM NaCl, 0.05% (v/v) IGEPAL 630 and 0.5 mM DTT), and the RNA was eluted using 1 ml of TRIzol (Thermo, 15596026) following the manufacturer's manual.

For quantification and statistical analysis, Percent Input Method was used for enrichment determination. In brief, signals obtained from the RIP were divided by signals obtained from an input sample. Before that, Input Ct value was corrected following the equation: $Ct_{corrected} = Ct_{input} - \log_2(20)$. Since three triplicates were analyzed for each sample, the average $Ct_{corrected}$ was calculated for target and non-target genes (primers of GAPDH were designed as a non-target control) in the input and the ΔCt in the RIP sample. Subsequently, the average ΔCt was calculated for each target RNA and samples (negative control: RIP in knockout cells, RIP sample: RIP in WT cells, Input). Finally, the enrichment fold was calculated as follows: enrichment fold = ($2^{-\Delta Ct} \times 100$).

**RNA deep sequencing**. Wild type, $Pum1^{-/-}$ and $Pum2^{-/-}$ HCT116 cells were cultured in McCoy's 5A (Gibco, 12330031) with 10% fetal bovine serum (Gibco, 10091148) according to the manufacturer's instructions and collected when grew to ~80% confluence. Total RNA was isolated using TRIzol (Invitrogen, 15596026) according to the protocol of the manufacturer. RNA quantification and quality were assayed using Nanodrop and Agilent RNA 6000 Nano RNA chip (Agilent Technologies, 5067-1511). rRNA was removed from 2 μg of each total RNA sample using NEBNext rRNA Depletion Kit (Human/Mouse/Rat) (NEB, E6310L). cDNA library was prepared using NEBNext Ultra RNA Library Prep Kit for Illumina (NEB, E7530S). The library was quantified by Nanodrop (Nanodrop Technologies) and assayed for size using Agilent High Sensitivity DNA Kit (Agilent, 5067-4626)

rRNA-depleted RNAs were sequenced using an Illumina HiSeq3000 platform (Lie Bing Co., Shanghai, China). The analyses of RNA-Seq data followed Shi et al.[72]. Briefly, paired-end reads were mapped onto the hg38 reference using STAR. HTSeq-count software was used to quantify the reads that were mapped to each gene. The R package DESeq2 was used to identify significantly differentially expressed genes in $Pum1^{-/-}/Pum2^{-/-}$ compared with wild type HCT116 cells (FDR ≤ 0.1 and fold change ≥ 1.5).

**Mass spectrometry and data analysis**. To profile the changes in protein expression in $Pum1^{-/-}$ and $Pum2^{-/-}$ cancer cells, mass spectrometry was performed using the TMT kit from Thermo Fisher (#90110) according to the product instructions. In brief, cells were suspended on ice in the lysis buffer (4% SDS, 100 mM Tris-HCl, pH = 7.6) with the protease inhibitor cocktail, following by the sonication of 10 s with an interval of 15 s for ten times, and the centrifugation for 10 min at 14,000 × g. The protein concentration was measured by the BCA kit. Proteins were fragmented by trypsin digestion for 16–18 h at 37 °C, following the

FASP method[73]. C18 Cartridge (3 M, 7 mm/3 ml) was used for the peptide desalting and the Thermo quantitative colorimetric peptides assay kit was used for peptide quantification. All fragmented peptides were separately labeled with TMT (ThermoFisher, 90110) for 1 h at room temperature and then combined prior to loading on the machine, Nexera X2 LC-30AD HPLC.

The raw mass spectrometry data were first searched against BSA sequence using Proteome discover 2.2 (Thermo Scientific) and the Sequest search engine. The precursor mass tolerance was set to 20 ppm and fragment mass tolerance was set to 1.2 da. The following variable modifications were taken into consideration: Carbamidomethyl (C), TMT6plex/+229.163 Da (Any N-Terminus), and TMT6plex/+229.163 Da (K). One percent FDR was used to filter away false positive hits. The selection criteria for upregulated proteins were that the number of unique peptides was more than 1 and the upregulated fold was greater than 1.2.

PUM1 and PUM2 peptides were detected in $Pum1^{-/-}$ and $Pum2^{-/-}$ cells in the above experiments because TMT uses an isobaric labeling system that inherently has the interference of contaminating ionized peptides that are from different proteins but share the same or almost the same retention time[74,75], and thus can not be discriminated by tandem mass spectrometry. To further validate the specificity of knockout cells, we prepared label-free samples from WT, $Pum1^{-/-}$ and $Pum2^{-/-}$ cells and measure them by mass spectrometry separately[73]. Different from TMT, the label-free quantification approach aims to correlate the mass spectrometric signal of intact peptides, or the number of peptide sequencing events with the relative protein quantity directly[76,77]. All MS/MS spectra were searched using SEQUEST against the human genome database. Each sample was separately run three times. Totally, 5503, 5496, and 5484 unique proteins were identified in WT, $Pum1^{-/-}$ and $Pum2^{-/-}$ samples, respectively, with 74.2, 72.1, and 73.8% overlaps with the triplicate runs of WT, $Pum1^{-/-}$ and $Pum2^{-/-}$ samples. This shows that label-free method provided deep coverage of the proteome (~80%). As summarized in Supplementary Table 2, 18 peptide-spectrum matches (PSMs) of nine PUM1-unique peptides and 8 PSMs of four PUM2-unique peptides were identified in WT cells. In contrast, no corresponding PUM1-unique or PUM2-unique peptide was detected in $Pum1^{-/-}$ or $Pum2^{-/-}$ cells, respectively. These results confirmed the absence of PUM1 and PUM2 in $Pum1^{-/-}$ or $Pum2^{-/-}$ cells, respectively.

**Photoactivatable ribonucleoside-enhanced crosslinking and immunoprecipitation (PAR-CLIP)**. PAR-CLIP was performed following a previous protocol[19]. For each condition (i.e., WT1-Pum1-PAR-CLIP, WT2-Pum1-PAR-CLIP, KO1-Pum1-PAR-CLIP, and KO2-Pum1-PAR-CLIP), 15 cm cell culture plates of HCT116 cells were grown in medium supplemented with 100 μM 4-thiouridine(4SU) for 14 h. Cells were washed once with cold PBS and cross-linked on ice using 0.15 J/cm² of 365 nm ultraviolet light in a Stratalinker. Cells were then scraped from culture dishes, transferred into a 15 ml Falcon tube, washed once with PBS, pelleted by centrifugation at 500 × g for 5 min and flash-frozen in liquid nitrogen for storage at −80 °C.

On the day of the experiment, cell pellets were thawed on ice, suspended in three volumes of lysis buffer [20 mM Tris-HCl pH = 7.4, 150 mM NaCl, 1 mM EDTA, 1% (v/v) IGEPAL™ CA-630, 0.5 mM DTT, cOmplete™ EDTA-free Protease Inhibitor Cocktail Tablets, and PhosStop Tablets] and incubated on ice for 30 min. (The volumes of cell pellets were ~2 ml, hence 6 ml of lysis buffer were added.) The cell lysate was collected by centrifugation at 13,000 × g at 4 °C for 30 min and further filtered by 0.45 μM pore size membrane filter (Millipore, Cat#: HAWP04700). Partial RNA digestion was performed before immunoprecipitation with 1 U/μl RNase T1 (Thermo Fisher Scientific, EN0541) at 22 °C for 15 min. Meanwhile, the Protein A bead-antibody complexes were prepared. For each sample, 0.25 mg/ml Pum1 antibody (Abcam, Cat#: ab92545) was conjugated to 100 μl protein A beads at 4 °C for 2 h, the unbound antibody was removed and the pre-cleared lysate was added to antibody-coupled beads and incubated overnight at 4 °C.

In the following day, the beads were washed five times with wash buffer [20 mM Tris-HCl pH = 7.4, 200 mM NaCl, 0.05% (v/v) IGEPAL® CA-630, 0.5 mM DTT, cOmplete™ EDTA-free Protease Inhibitor Cocktail Tablets, and PhosStop Tablets] and digested with 10U/μl RNase T1 at 22 °C for 15 min, followed by washing the beads three times in high salt wash buffer [20 mM Tris-HCl pH = 7.4, 300 mM NaCl, 0.05% (v/v) IGEPAL® CA-630, 0.5 mM DTT, cOmplete™ EDTA-free Protease Inhibitor Cocktail Tablets, and PhosStop Tablets]. Immunopurified protein–RNA complexes were dephosphorylated by resuspending beads in 200 μl dephosphorylation mix [1 × NEBuffer 3 (New England Biolabs, Cat#: B7003S), 0.5 U/μl calf intestinal alkaline phosphatase (Thermo Fisher Scientific, Catalog # M0290)] and incubated at 37 °C for 10 min. Then, the beads were washed twice with 1× NEBuffer 3 and twice in PNK buffer without DTT (50 mM Tris-HCl pH = 7.4, 50 mM NaCl, 10 mM MgCl₂·H₂O). Next, 3′ ligation was performed according to the NEBNext Multiplex Small RNA Library Prep Set for Illumina (Set 1; New England Biolabs, Cat#: E7300S).

On Day 3, the beads were washed twice with the high salt wash buffer followed by washing twice with PNK buffer without DTT. The protein–RNA complexes were then phosphorylated by resuspending beads in 100 μl phosphorylation mix [1×T4 DNA ligase buffer (Thermo Fisher Scientific, Cat#: B69), 1 U/μl of T4 Polynucleotide kinase (Thermo Fisher Scientific, Cat#: EK0031)] and incubated at 37 °C for 30 min. Then the beads were washed three times with the wash buffer.

Subsequently, NuPAGE LDS Sample Buffer (Thermo Fisher Scientific, Cat#: NP0007) was added to each sample and incubated at 95 °C for 5 min. Protein–RNA complexes were resolved using NuPAGE 4–12% Bis-Tris-HCl Gels (Thermo Fisher Scientific, Cat#: NP0335BOX) and desired complexes were excised from gel using a clean scalpel. The gel pieces were transferred to a D-tube Dialyzer Midi tube (Millipore, Cat#: 71506) and 800 µl of 1×SDS buffer (Thermo Fisher Scientific, Cat#: NP0001) was added. After electroeluted the cross-linked RNA-RBP complex, the solution was transferred to tubes, added 800 µl of proteinase K buffer [100 mM Tris-HCl pH = 7.4, 150 mM NaCl, 12.5 mM EDTA, 2% (v/v) SDS (Bio-Rad, Cat#:1610418)] with 1.2 mg/ml of proteinase K (New England Biolabs, Cat#P8107S), and incubated at 55 °C for 30 min. Then the RNA was recovered using Trizol reagent extraction according to the manufactory's protocol. The following steps of library preparation were performed as described according to the NEBNext Multiplex Small RNA Library Prep Set for Illumina (Set 1).

**Analysis of PAR-CLIP data**. The forward read1 was taken from the paired-end sequenced PAR-CLIP data of each sample. This is because the RBP (RNA-binding protein)-bound fragments were short (<70 bp), the 150 bp long paired-end read1 and read2 will share the same fragment, thus one read (either read1 or read2) is sufficient. TrimGalore (version 0.4.4_dev, http://www.bioinformatics.babraham.ac.uk/projects/trim_galore/) was used to trim off the adapter sequences. Only reads in the size range of 17–50 nucleotides were retained for downstream analyses. The size-selected reads were then aligned to the human hg19 genome using 0, 1, and 2 mismatches sequentially using bowtie (version 1.2.1.1)[78] and the following procedure: 1) map reads to the genome using 0 mismatch (-v 0), retain the mapped reads, and align the unmapped reads to the genome using 1 mismatch (-v 1); 2) retain the 1-mismatched reads and map the unmapped reads to the genome using 2 mismatches (-v 2)). A maximum of two mismatches were used because the RBP-binding sites contained T-to-C conversions during the PAR-CLIP treatment. To increase the signal-to-noise ratio, only reads resulting from up to 2 T-to-C conversions were selected, yet reads from other mismatches were discarded. Customized scripts and commands (available upon request) were developed for processing aligned SAM files to identify reads that perfectly aligned to the genome, and those aligned to the genome with 1 or 2 mismatches resulting from T-to-C conversion(s) only. Subsequently, the three SAM files (0 mismatch, 1 T-C converted SAM, 2 T-C converted SAM) were merged and sorted using samtools (version 1.4.1)[79], and acted as the input for PARalyzer (v1.5)[80]. The parameters in the ".ini" file used as the input for PARazyler are as follows:

BANDWIDTH = 3,
CONVERSION = T > C,
MINIMUM_READ_COUNT_PER_GROUP = 5,
MINIMUM_READ_COUNT_PER_CLUSTER = 2,
MINIMUM_READ_COUNT_FOR_KDE = 3,
MINIMUM_CLUSTER_SIZE = 11,
MINIMUM_CONVERSION_LOCATIONS_FOR_CLUSTER = 2,
MINIMUM_CONVERSION_COUNT_FOR_CLUSTER = 2,
MINIMUM_READ_COUNT_FOR_CLUSTER_INCLUSION = 1,
MINIMUM_READ_LENGTH = 20,
MAXIMUM_NUMBER_OF_NON_CONVERSION_MISMATCHES = 1,
GENOME_2BIT_FILE = Homo_sapiens.GRCh19.dna.primary_assembly.2 bit.

The bound regions identified by PARalyzer from the "resulting.cluster.file" were processed and intersected with the hg19 gene annotations using bedtools (version 2.22.0)[81] for detailed annotation (for instance, 3′UTR, or CDS, 5′UTR, or intron of particular genes of the hg19 genome). Two replicates for each genotype (wild type and knockout) were analyzed separately. To identify PUM1-bound genes, the bound genes from each replicate of the wild type were combined by taking the union not the intersect, and those genes identified to be bound by PUM1 in the knockout were excluded, as they presumably contain non-specific binding sites not resulting from PUM1 but other non-specific activities of the antibody. The bound regions were analyzed for enriched motif analysis and visualized in IGV[82] to infer the potential binding affinity expressed as the percentage of T-to-C conversions among all mapped reads covering the region of interest.

**Immunofluorescence microscopy**. Immunofluorescence microscopy assay was performed according to a previously described standard method[71]. The following antibodies were used: Recombinant Anti-Pumilio 1 antibody [EPR3795] (1:200, Abcam, Cat#: ab92545), Recombinant Anti-Pumilio 2 antibody [EPR3813] (1:200, Abcam, Cat#: ab92390) and Goat anti-Rabbit IgG (H + L) Highly Cross-Adsorbed Secondary Antibody, Alexa Fluor Plus 555 (1:1000, invitrogen, Cat#: A32732). Fluorescence microscopic images were acquired using Zeiss LSM 710 confocal microscope (Zeiss), analyses were performed using ZEN (version 2012).

**Immunohistochemistry**. Formalin-fixed, paraffin-embedded colorectal tumor tissue blocks from mice were used in our investigation. Immunohistochemical staining was performed using primary antibodies and HRP-conjugated secondary antibodies. The following antibodies were used: Recombinant Anti-Pumilio 1 antibody [EPR3795] (1:200, Abcam, Cat#: ab92545), Recombinant Anti-Pumilio 2 antibody [EPR3813] (1:200, Abcam, Cat#: ab92390), p21 Waf1/Cip1 (12D1) Rabbit mAb (1:200, cell signaling technology, Cat#: 2947S), Anti-Ki67 antibody (1:200, Santa Cruz Biotechnology Inc, Cat#: sc-7846) and HRP-conjugated Affinipure

Goat Anti-Rabbit IgG(H + L) (1:1000, Proteintech, Cat#: SA00001-2). TUNEL staining was performed using an in situ Cell Death Detection Kit (Roche) based on the manufacturer's instructions.

**Luciferase reporter assay**. The 3′UTR of p21 cDNA was amplified and cloned into psiCHECK2 vector (C8021, Promega) using primers specified in Supplementary Table 1. In the 3′UTR reporter assay, HCT116 cells in 24-well plates were transfected with 100 ng of the psiCHECK2-p21-3′UTR-WT or 3′UTR mutant plasmids with mutations in PUM1 binding sites and 500 ng pGL3-PUM1 plasmid along with 2 µl Lipofectamine 2000 (Invitrogen). Lysates were harvested after 48 h of transfection. The reporter activity was measured with the Dual Luciferase Assay (E1910, Promega).

**Tumorigenesis assay**. Four-week-old BALB/c nude mice (male) were purchased from Shanghai Lingchang Biotechnology Co., Ltd (Shanghai, China). All experiments were performed under the ShanghaiTech guidelines for the care and use of laboratory animals. BALB/c nude mice were randomly divided into two groups ($n = 8$ per group). Wild type, $Pum1^{-/-}$ and $Pum2^{-/-}$ HCT116 ($2 \times 10^6$) or RKO ($3 \times 10^6$) cells in 0.1 ml of medium mixture (medium: Matrigel = 1: 1) were subcutaneously injected into the nude mice. The tumor size was measured at time intervals as indicated in Fig. 2m, p. The mice were euthanized by cervical dislocation and the tumors were dissected. The tumor weight was measured, and the tumor volume was calculated using the following formula: $D \times d^2/2$, with "D" representing the longest diameter and "d" representing the shortest diameter. The maximum diameter of the tumor permitted by the animal ethics committee is 1.5 cm, and the size of all tumors in our experiment does not exceed 1.5 cm.

**AOM/DSS model**. There are two types of mice used in the AOM/DSS model: $Lgr5^{cre}::Pum1^{flox/flox}::Pum2^{flox/flox}$ and $Pum1^{flox/flox}::Pum2^{flox/flox}$. Six to eight-week-old C57BL/6J mice (male) were first induced by intraperitoneal injection with tamoxifen (120 mg/kg) every other day for a total of 7 days. Sixteen days later, mice were injected intraperitoneally with 12.5 mg/kg azoxymethane (Sigma, St.Louis, MO). Dextran sodium sulfate (DSS) 2.5% (wt/vol) (molecular mass, 36–50 kilodaltons; Meilun biotechnology) was given for 5 days. Mice were then given regular drinking water for 14 days, followed by one additional cycle with DSS. In the third cycle, the DSS was adjusted from 2.5 to 2%. On day 65, mice were sacrificed for testing.

**Preparation and characterization of siRNA-loaded MSN**. We previously developed a woven polyethylenimine (PEI) (1.8 kDa)-coated, glutathione (GSH)-responsive, mesoporous silica nanoparticle (MSN) for nucleic acid (miRNA-145) delivery[57]. The nanoparticles can quickly escape from the lysosome into the cytosol for miRNA-145 delivery, mediated by the proton sponge effect of PEI[78]. The reducing environment (0.5–10 mM GSH)[58,59] within tumor cells removed the PEI from the MSN surface through hydrolyzing the disulfide bond, and thus allow nucleic acid release in the cytoplasm[57]. Here, this smart nanocarrier was adopted for the delivery of siPum1 and 2, which were complexed with MSN to generate siPum1/2@MSN. Transmission electron microscopy (TEM) images were acquired on a FEI Talos F200X system. The hydrodynamic size and zeta potential of nanoparticles were determined through dynamic light scattering (DLS) and measured using a ZetaSizer Nano ZS instrument (Malvern, Worcestershire, UK).

siRNA (fluorescein (FAM) labeled) release test was performed in the presence of 5 mM GSH-contained PBS in the centrifuge tubes. The tubes were placed in a gas bath at 37 °C shaking at 100 rpm. At pre-determined time point (1, 2, 4, 8, 12, and 24 h), empty Dipalmitoylphosphatidylcholine (DPPC) liposomes (−30 mV, 120 nm, 25 mg/ml) were added to bind free PEI and PEI-siRNA complex. Then, the solution was transferred into Ultra Centrifugal Filters (molecular weight cut-off 50 kDa) and centrifuged under $2350 \times g$ at 4 °C for 30 min. FAM-labeled siRNA (Ex 488 nm, Em 520 nm) in the filtrate was collected for the quantification of nucleic acid release. The in vivo tumor targeting of the siRNA-loaded nanoparticles 24 h after i.v. injection was examined using the IVIS Spectrum/CT imaging system (PerkinElmer, USA), as previously described[57].

siRNA molecules specifically targeting the mRNA of Pum1 and Pum2 were purchased from RIBOBIO (China). siPum1: GGTCAGAGTTTCCATGTGA, siPum2: CTGAAGTAGTTGAGCGCTT[27]. siRNA (FAM labeled) release test was performed in the presence of 5 mM GSH-contained PBS in the centrifuge tubes. The tubes were placed in a gas bath at 37 °C shaking at 100 rpm. At pre-determined time point (1, 2, 4, 8, 12, and 24 h), empty DPPC liposomes (−30 nm, 120 nm, 25 mg/ml) were added to bind free PEI and PEI-siRNA complex. Then, the solution was transferred into Ultra Centrifugal Filters (molecular weight cut-off 50 kDa) and then centrifuged under $2350 \times g$ at 4 °C for 30 min. FAM-labeled siRNA (Ex 488 nm, Em 520 nm) in the filtrate was collected for the quantification of nucleic acid release.

**Antitumor therapy in orthotopic colorectal tumor model**. The orthotopic CRC model was developed in female BALB/c nude mice as described previously[57]. Twenty-eight days after orthotopic tumor cell (HCT116-luc, COLO205-luc) inoculation, the tumor-bearing mice were randomly allocated into five groups ($n = 5$) according to the tumor bioluminescent intensity and treated with saline (control), MSN loaded with negative control siRNA with a scrambled sequence (siNC@MSN), and MSN loaded with siPum1 (siPum1@MSN), siPum2

(siPum1@MSN) or both (siPum1/2@MSN). The siRNA dose was 1 mg/kg on days 0, 3, 6, and 9, respectively. Each group included ten mice. The tumor burden was monitored weekly using bioluminescence imaging (IVIS spectrum CT). The body weight was recorded throughout the study. At the end of the study (Day 30), the mice were sacrificed, the tumors were removed and weighted. Other major organs included heart, liver, spleen, lung, and kidney were harvested for ex vivo biolu-minescent imaging to examine the metastasis. In a separate study, on Day 11 (2 days after the last injection of the nanoparticles), three mice from each group were sacrificed. The major organs, including heart, liver, spleen, lung, and kidney were collected for H&E histological assay for toxicity evaluation.

**Statistics and reproducibility**. For comparisons of decay curves, significance between fitted curves is indicated. For comparisons between groups, two-tailed Stu-dents's $T$-tests were used. Exact $P$ values are shown in the figures. Data were presented as the mean ± SD or mean ± SEM and $P < 0.05$ was considered significant. Statistical analysis was performed using GraphPad Prism 8 software. Pearsons's correlation analyses were used to calculate the regression and correlation between two groups. As indicated in the figure legends, all assays were performed in three biological replicates unless stated otherwise. Representative micrographs and western blot shown in figures were repeated three times independently with similar results.

**Reporting summary**. Further information on research design is available in the Nature Research Reporting Summary linked to this article.

## Data availability
The accession number for the RNA-seq and PAR-CLIP data reported in this paper is GEO: PRJNA648706. The accession number for the Mass spectrum data reported in this paper is PXD: 027513. The mass spectrometry proteomics data have been deposited to the ProteomeXchange Consortium (http://proteomecentral.proteomexchange.org) via the iProX partner repository[83] with the dataset identifier PXD027513. Source data are provided with this paper.

## Code availability
We used the following publicly available software for analysis of the PAR-CLIP data: TrimGalore version 0.4.4_dev, available at http://www.bioinformatics.babraham.ac.uk/projects/trim_galore/, bowtie version 1.2.1.1, available at http://bowtie-bio.sourceforge.net/index.shtml, samtools version 1.4.1, available at https://github.com/samtools/samtools/releases/, PARalyzer version 1.5, available at https://ohlerlab.mdc-berlin.de/software/PARalyzer_85/,bedtools version 2.22.0, available at https://bedtools.readthedocs.io/en/latest/index.html. A custom script used to analyze the PAR-CLIP data is available in the Zenodo repository under https://doi.org/10.5281/zenodo.5981256.

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

## Acknowledgements

We thank Wenzhang Chen, Hongwei Zhang, and Wei Zhu in the ShanghaiTech University Proteomics platform for assistance with mass spectrometry; Fan Yang, Shuo Shi, Chen Wang, and Ting Lu in the SIAIS at ShanghaiTech University for technical assistance. We appreciate ShanghaiTech High Performance Computing Platform for providing the computing resources and technical support. This research was funded by the Shanghai Institute for Advanced Immunochemical Studies at ShanghaiTech University. S.L. was supported by National Natural Science Foundation of China (81772798). C.F. was supported by National Natural Science Foundation of China (81773274). Y.G. and Z.L. were supported by the School of Life Science and Technology at ShanghaiTech University.

## Author contributions

H.L. and S.L. conceived the project. Y.G., Z.L., Y.Y., C.F., S.L., and H.L. designed the experiments and interpreted the results. In details, Y.G. performed the AOM-DSS mouse experiments and S.L. did data analysis. Y.G. and Z.L. performed the experiments on Pum1/2 biological function in vitro and data analysis. Y.G. performed the experiments of PAR-CLIP, Z.Y. and Y.G. performed analyses of PAR-CLIP data. Z.L. performed RNA seq and mass spec experiments, Z.Y., Y.G., S.L., and W.W. did data analysis. Y.G. and J.Z. performed Pum1 target gene validation experiments and data analysis. Y.G. generated *p21* PRE mutations and studied their biological function. Y.Y. and Q.L. performed RNAi therapy experiments and data analysis. S.L. analyzed public cancer database. Y.G., S.L., C.F., and H.L. wrote the manuscript. Z.L. and Z.Y. commented on the manuscript. H.L. and S.L. supervised the overall work and C.F. was responsible for RNAi therapy.

## Competing interests

The authors declare no competing interests.
