## [Peer Review File · Nature Communications]

PUMILIO proteins promote colorectal cancer growth via suppressing p21REVIEWER COMMENTS

Reviewer #1 (Remarks to the Author):

This study investigates the relationship of two RNA binding proteins, PUM1 and PUM2, to colorectal cancer (CRC). Previous studies link these proteins to several cancers, including colorectal cancer, and showed that they bind and regulate cancer genes. Based on existing evidence, PUM1 and PUM2 do not appear to cause cancer, but may modify cancer phenotypes. PUMs are reported to promote proliferation in a number of contexts, including cancer (PMID: 28232582, PMID: 20818387; PMID: 30811992; PMID: 29428722; PMID: 27758885). The potential role of PUMs in CRC has not been extensively analyzed, though at least one publication provides evidence of their relevance (PMID: 33508364).

In this study, the authors first correlate over-expression of PUM1 and PUM2 to human CRC. While a few of the observed correlations are intriguing, overall the data and statistical analysis are not very convincing and the authors make several unsubstantiated conclusions (see below). They create the hypothesis that overexpression of PUMs promotes cancer; however, they do not actually test this hypothesis. Their cancer phenotype assays instead focus on loss of function approaches, primarily knockout. (Presently, there is no compelling evidence that loss of PUM1 or PUM2 function is related to cancer susceptibility in humans or mouse models.)

The authors use a conditional PUM1/2 knockout mouse model and human CRC cell lines to investigate the contribution of PUMs to CRC. Using a mouse model for Azoxymethane (AOM)/Dextran Sodium Sulfate (DSS) induced inflammatory CRC to investigate the effects of PUM1 and PUM2 on phenotypes, they found that, upon PUM1/2 knockout in the intestinal epithelia, the number of colon tumors and proliferation of tumor cells were reduced. The double knockout mice had somewhat longer colons and increased body weight. KO of PUM1 and PUM2 was confirmed by immunohistochemistry. Immunohistochemistry of a proliferation marker indicated that the double knockout reduced proliferation in the tumors. The authors state that apoptosis was increased; however, their data in Figure 1M shows that this difference was not statistically significant. The authors report that the morphology of the small intestine in the double knockout mice was not affected. This is surprising, since Uyhazi et al 2019 reported that PUM1 knockout mice had disrupted small intestine morphology. A pertinent control for these experiments would be to include analysis of the colon morphology, length, and body weight of WT and double PUM knockout mice in the absence of AOM and DSS.

The authors go on to assess PUM1 and PUM2 mRNA and protein expression in several human colon cancer cell lines compared to a single "normal" cell line. They attempt to correlate this to cancer but doing so is superficial and not convincing. In fact, a previous report and public data show that PUM1 and PUM2 are broadly expressed in normal and cancer cell lines and tissues (e.g. Spassov and Juresic 2002 Gene, Cancer Cell Line Encyclopedia, Genotype-Tissue Expression (GTEx) database).

The authors then studied the effect of PUM1 and PUM2 loss of function in HCT116 cells, a highly proliferative cancer cell line that is transformed by mutant RAS oncogene. In Fig S3, they report that RNAi depletion of either PUM1, PUM2 or both reduces proliferation. PUM1 has a small increase effect on cells in G1. In the results, the authors state that "we found that PUM1 and/or PUM2 knockdown displayed a significantly increased portion of cells in G0/G1 phase" However, this is not accurate because the effect of PUM2 knockdown was not significant (Fig. S3G).

The authors then knockout PUM1 or PUM2 in the HCT116 cell line and find that colony formation and proliferation are reduced by PUM1 knockout in two clonal lines. Similar observations are made in the RKO cell line. Strangely, the PUM2 knockout has variable effects in both cell lines. One clonal knockout line showed a significant effect on cell proliferation, colony formation, and cell cycle, whereas the other line did not. This is concerning and could be indicative of off target effects of the CRISPR knockout. The apparent discrepancy in the effects of RNAi and knockout of PUM2 on cell proliferation and colony formation is concerning. The authors report 5-6 clonal lines with PUM1 or PUM2 knockout in the supplement with corresponding genotypes. It's not clear which of these were used in Figure 2. It's unclear why only two of these lines were tested for changes in proliferation and colony formation. Doing so could help assess the variability of the PUM2 knockout

effects. Further, the authors should test the effect of knockout of both PUM1 and PUM2 simultaneously. The ability of PUM1 or PUM2 transgenes to rescue this proliferation defect would provide additional support.

The authors then attempt to identify the genes that are regulated by either PUM1 or PUM2. They performed RNA seq to measure differential levels of mRNAs in response to PUM1 or PUM2 knockout. Tandem mass tagging based proteomics was used to measure relative changes in protein levels. The authors also performed crosslinking and immunoprecipitation to identify mRNAs bound by PUM1 (PUM2 was not analyzed). Unfortunately, from this large amount of data, only a cursory analysis is provided. As a result, it is not clear what, if any, new insights these datasets provide. Instead, the presented results reaffirm what is already well-documented in the literature - that PUM proteins bind to a PRE site that is typically located in 3'UTR of mRNAs, and that PUMs regulate genes involved in cell cycle and proliferation.

The results of the global approaches are problematic for several reasons, including lacking rigor, evidence of reproducibility, and proper documentation:

1) For the RNA-seq analysis, the authors report thousands of genes with fold change and p-values. However, reviewing their data (Tables S2 and S3) shows that hundreds of these genes have values (RPKM? TPM?) that are 0 in one or more samples. Thus, the reported fold change for these genes are not valid. The reproducibility of the two replicate samples is not assessed. Concern about reproducibility is heightened by the heat maps in Fig. 4K, which indicate large differences between the replicates. Two replicates is the bare minimum for differential expression analysis. The authors should assess the Biological Coefficient of Variation (BCV) for replicates within each condition. Principal component analysis (PCA) or multidimensional scaling (MDS) analysis across conditions and replicates is necessary to determine if the results are reliable.

2) The RNA seq methodology is not adequately described including how RNAs were isolated, whether rRNA was depleted, or how libraries were generated. The authors should adhere to the standards set forth in the ENCODE Guidelines and Best Practices for RNA-Seq.

3) Documentation of the quantitative relative proteomics methodology and data analysis is completely inadequate. The methods section provides only two sentences. The Supplemental table reports values that are not defined. The primary data should be deposited to a publicly accessible database such as ProteomeXchange.

4) The results of the quantitative mass spectrometry data (Table S4 and S5) fail to provide any type of statistical analysis that is necessary to assess the reproducibility of the approach and statistical significance of the results. The values reported in the data table are not defined. Again, concern about reproducibility is heightened by Figure 4L, where the heat map shows large differences between the protein level measurements. Again, BCV, PCA or MDS analysis is needed to assess reproducibility.

5) Reproducibility of the PAR-CLIP data and statistical significance of the results are not addressed. In fact, from what is presented for PAR-CLIP in Figure 4A, there is substantial variation between the two replicates (only about 1/3 of the genes overlap). Proper statistical analysis is necessary (DOI: 10.1214/11-AOAS466; PMID: 27018577).

6) The data analysis is inadequate. The RNA Seq, proteomics, and PARCLIP datasets are given the most cursory of analysis. The effects of PUM1 and PUM2 knockout are not compared. The RNA and protein level effects are not directly compared. Illogically, different fold change cutoffs are used for protein and RNA level changes. The GO term analysis in Figure 3C-J and Figure 4 is illegible due to microscopic text. The reported GO term analysis should be reported in a supplementary table that includes the number and identity of the genes in each category along with the statistical significance.

7) The overlap of genes affected at either RNA or protein levels reported in Figure 4I and 4J show that the vast majority of genes in either category are likely indirect effects, or that there is a high incidence of false positives in the CLIP, RNA seq, and TMT datasets.

Overall, seemingly little new information is extracted from the RNA seq, proteomics, and PARCLIP

datasets. The authors arbitrarily selected CDKN1A/p21 mRNA from the list. In fact, when the differentially expressed mRNAs are ranked by fold change, CDKN1A is not even in the top 1000 of affected genes. This biased selection is not explained or justified, and could substantially miss the most important regulated genes. CDKN1A was already known to be a PUM target mRNA; previous research showed that PUMs bind and regulate CDKN1A mRNA (PMID: 18411299; PMID: 18776931; PMID: 30811992)..

Based on the RT and qPCR analysis in the PUM1 or PUM2 knockout cells, it appears that only PUM1 knockout leads to an increase in p21 mRNA (Fig. 5). Qualitative western blot data of p21 is also provided in Fig 5B; however, quantitative western blot analysis from multiple replicates would be more convincing, and could assess fold change increase in protein levels along with statistical significance.

While p21 is an attractive target because it has well documented tumor suppressor and cell cycle functions, several inconsistencies limit its importance to PUM regulation of proliferation. Both PUM1 and PUM2 affect proliferation and have increased expression in CRC. Given that both PUMs bind to PREs to inhibit genes, it seems that a shared set of mRNAs would be of vital importance. Yet the authors observe that only PUM1 inhibits p21. The authors wish to conclude the PUM1 inhibition of p21 is responsible for the control of cell proliferation, seemingly at odds with both PUM1 and PUM2 controlling proliferation. Additionally, it is well documented that PUM1 and PUM2 bind to the same PREs, and the authors confirm that both PUMs bind to p21 mRNA in Figure 5, Thus, the contention that only PUM1 inhibits p21 to control proliferation is paradoxical.

The authors attempt to analyze the effect of PUM1 and PUM2 on degradation of the p21 mRNA by measuring mRNA decay following transcription shutoff using actinomycin D. The analysis is flawed because they did not measure the half life of the mRNA in the different conditions. Instead, they appear to show only a significant difference between the final time points. This is not acceptable. Observed half lives should be calculated, reproducibility should be assessed, and statistical significance of the differential decay should be determined and reported. From the data presented, it is clear that the p21 mRNA is unstable in the absence of both PUM1 or PUM2, indicating that additional dominant mechanisms control its degradation.

The role of the PRE in the p21 mRNA was investigated. Its mutation appears to increase p21 protein, but again the lack of quantitative western blot data backed by replicates and statistics tempers the strength of the conclusions. At the mRNA level, the effect size of the PRE mutation does not phenocopy that of the PUM1 knockout (Fig. 5I) and PUM1 still can bind to the p21 PRE mt mRNA (Figure 5J). Therefore, the authors claim that "p21 PRE mut abolishes PUM1-mediated regulation of p21" (p14 line 299) is not supported by their data.

The RNA coimmunoprecipitation (RIP) data analysis method is not described in the manuscript and appears to be incorrect (Fig 4E, Fig 5C, D, J). The fold enrichment of the mRNA in each RIP sample should be calculated in each condition relative to the negative control RIP after those values are normalized to the amount of mRNA in the input of that specific sample. This is important to address changes in the input levels of p21 in the WT, PUM knockout, and p21 PRE mut conditions. Statistical analysis of the fold enrichment data must be included to assess reproducibility.

The RT-qPCR methods used throughout the study are not described. The methodology and data analysis should be reported in accordance with MIQE guidelines (PMID: 19246619). The authors indicate the RNA was purified by trizol, which notoriously leads to genomic DNA contamination. There is no mention of DNase treatment to remove contamination, nor was it assessed by the proper minus-RT controls.

Mutation of the PRE in the p21 gene is shown in Figure 5K-O to reduce proliferation, slightly increasing doubling time, and the portion of the cells in G1 phase. This is some of the most interesting data in the manuscript. Strengthening the preceding analysis of PUM binding and inhibition of p21 protein expression would bolster this important observation.

The authors attempted to classify the mRNAs based on microRNA sites, predicted RNA structure, and PUM binding. As presented on page 16, lines 319-334, the rationale and classification are difficult to understand. The approaches used to assign these categories are not explained at all.

How was RNA structure predicted and what types of parameters were employed. Why is analysis relevant? In silico RNA structure prediction of mRNAs is notoriously difficult, heterogeneous, and inaccurate. How the microRNA binding category was assigned is not described. How PUM1 and PUM2 binding was assigned is not described (the authors only did PAR-CLIP on PUM1). At the end of this confusing section, they state that significant changes are observed in mRNA, but then the corresponding Fig S10 does not include any type of statistical analysis to justify that statement. This section is confusing, inadequately documented, and doesn't provide meaningful insight.

The authors then analyze AGO2, one of several proteins that can form miRNA induced silencing complexes. Again, the rationale behind this is not entirely clear. While one study reported that PUM-AGO interaction could affect translation in vitro (PMID: 22231398), another found that AGOs are not required for PUM activity in cells (PMID: 24942623). The authors create a hypothetical model of PUM1 and AGO2 (Fig. 6A) that is speculative and is not tested experimentally. Indeed, their own data shows that PUM1 and AGO2 associate in a manner that is sensitive to RNase treatment, indicating that they co-occupy RNAs, instead of their presumed protein-protein interaction. The authors also test mutations of PUM1 and PUM2, and though they conclude that these mutations "weakened" the PUM1-AGO2 interaction, the co-IP data does not support that conclusion. A single lane of a western blot of a co-IP experiment does not provide a quantitative means of assessing binding affinity between two proteins. From Fig. 6B and 6C, it is apparent that AGO2 co-IPs with both WT and mt PUM1 and PUM2. In the end, this analysis of PUM and AGO doesn't provide new information that helps understand how PUM1 inhibits mRNAs.

Next, the authors focus on the microRNA miR130a and its effect on p21 expression. Another study (PMID: 25681685), not cited here, had previously reported that miR130a inhibits p21. In Fig. S11, the authors show that miR130a mimic increased proliferation of colon cancer cell lines, whereas an miR130a inhibitor decreased proliferation. They then analyze miR130a effect on p21 protein, mRNA, and in luciferase assays in Fig S12 and Fig 6. In these experiments, addition of miR130 reduced p21 mRNA and reporter activity. Oddly, the anti-miR130a was not effective - perhaps miR130a is not present/abundant in these cells? The ability of PUM1 or miR130a to inhibit the p21 reporter appears to be independent, not interdependent. The renilla protein expression of the p21 3'UTR reporters with mutant PRE and/or miR130a site, should have been directly compared to the wild type 3'UTR to measure the relative contributions of each binding site. At the level of p21 protein, the ability to interpret the effects of PUM1 and PUM2 and miR-130a on p21 expression are hindered by the lack of quantitation and assessment of reproducibility of western blots shown in Figures 6H, 6I. From those blots, loss of PUM1 does qualitatively support increased p21, but the effect of miR-130a or the anti-miR are hard to discern. That observation calls into question whether p21 is really the means by which miR130a affects cell proliferation.

The authors also investigate CDKN1B/p27 mRNA, which was already documented as inhibited by PUMs and microRNAs (PMID: 20818387; PMID: 29165587; PMID: 30811992). The effects of PUM1 and PUM2 on p27 expression are hard to discern in Figure 6K and 6L and are subject to the critiques raised above. The relevance of this analysis to the current study is not clear, and the results reaffirm what was already published. Overall, the section on PUMs and microRNAs does not provide new, convincing, and relevant insights.

Finally, one of the most interesting experiments in his manuscript is the assessment of feasibility targeting PUMs as a potential cancer therapy using nanoparticle-encapsulated siRNAs. Using a xenograft model, the authors report decreased colorectal tumor size and reduced metastasis caused by PUM1, PUM2 siRNAs. There are several reservations about the results. First, the authors do not confirm knockdown of PUM1 or PUM2 by the siRNAs in the xenografts (by western blot and/or by immunostaining). Single siRNAs are used for PUM1 and PUM2, raising the possibility that off target effects contribute to the observations. In their interpretation of these results, the authors wish to ascribe the effects on control of p21. However, they did not assess the effect of PUM1/2 siRNAs on p21 in the xenograft model. Also, both PUM1 and PUM2 siRNAs are effective in reducing growth of the xenograft, but the authors' earlier data showed that only PUM1 inhibits p21.

The authors wish to conclude that PUM1 and PUM2 are oncogenic, but their data does not prove this assertion. By definition, oncogenes have the capacity to transform normal cells to a cancerous state, an effect that is not tested for PUM1 and PUM2 in this manuscript. Instead, they provide

corroborating data that PUMs promote proliferation of previously transformed cancer cell lines or in an induced model of colon cancer.

Additional critiques:

The figures are extremely small, making it difficult or even impossible to read.

The figure legends are very superficial, making it difficult to interpret what is represented.

The authors make unsubstantiated claims based on data that are not statistically significant. In addition to the examples cited above, additional instances include:

The authors conclusion that "all these data strongly correlate PUM1 and PUM2 with progression of human CRC" is not supported by most of the data they represent in Figures 1 and Figure S1.

Figure S1A and S1B: In the results, the authors state that CRC patient samples exhibit the highest PUM1 and PUM2 protein expression. However, these figures do not actually show protein levels. Instead, they plot % patients relative to cancer samples.

Figure S1C and S1D: In the results, the authors claim that "PUM1 and PUM2 were higher in Hong colorectal and Skrzypaczak colorectal compared to normal colon cancer" However, only the PUM1 correlation is statistically significant, PUM2 is not.

Figure S1G, S1H: In the results, the authors claim that "the expression of PUM1 and PUM2 negatively correlated with the survival rate of CRC patients" The Kaplan-Meier graphs do not support this conclusion. First, the authors bin the data into high and low PUM1 and PUM2 but do not define these parameters. Second, the data included in the graphs show that these relationships are not statistically significant.

Figure 1A and 1C: PUM1 and PUM2 mRNA levels are compared between matched normal and CRC patient samples. Though 22 comparisons are plotted, only one p value is reported and its not clear how that was determined.

In the results, page 14, line 278, the authors state that p21 "expression is negatively correlated with survival rate in CRC" however the data presented in Figure S9A disease free survival indicate that the association is not statistically significant. The TPM categorization of high and low expression is not defined.

The methods, analysis, and results are inadequately documented, as described above.

Inadequate or missing data and statistical analysis (examples are noted above). For all experiments, the data and statistical analysis methods and results need to be reported, including number and type of replicates, error type and values, the type type of statistical tests.

Throughout the manuscript, citations of directly relevant literature are omitted.

An example is the statement: "Recent studies have shown that post-transcriptional regulation mediated by RNA binding proteins plays an important role in the initiation and development of CRC." yet they provide no citations for this statement.

Overall the manuscript is poorly written.

The manuscript and figures contain many grammatical and typographical errors.

The introduction is incomplete and confusing due to lack of logical progression and contradictory statements.

For example, statements in the first paragraph contradict each other. The authors state the "Recent studies have shown that post-transcriptional regulation mediated by RNA binding proteins plays an important role in the initiation and development of CRC." and then one sentence later state the opposite "However, the regulation at the regulation at the post-transcriptional levels mediated by RNA-binding proteins, emerging important, remains much less investigated."

The second paragraph of the introduction is an example of the confusing text - the authors attempt to relate some basic background information on PUM1 and PUM2 while arguing that they

may be either oncogenic or have roles in preventing cancer. The text is not logical, and I suspect that this is because the presentation of the previous data is incomplete and overly simplified. Another example is the third paragraph of the introduction, which starts with a confusing run-on-sentence.

Reviewer #2 (Remarks to the Author):

Gong et al. study is aimed to demonstrate the role of Pum1 and Pum2 in CRC, by using in vitro and in vivo models (CRC cell lines, transgenic mice and xenograft tumours). By combining clinical information, in vitro and in vivo data, supported by transcriptomic and proteomic analyses, the authors demonstrated how Pum1 and Pum2 are crucial oncogenic proteins for the development of CRC.

Overall, the study is well organized, supported by strong and valid data and it adds novel insights into the molecular regulation of CRC.

The following are the concerns of this reviewer:

- Authors should have characterized, both in vitro and in vivo, the cells that survived after Pum1 and Pum2 targeting. Indeed, the block of Pum1/2 reduces the proliferative potential of colorectal cancer cells, by enriching cells in the G0-G1 cell cycle phase. Given the importance of cancer stem cells (CSCs), enriched in slow cycling/quiescent cells, it is crucial to determine how the targeting of Pum1/2 could induce, or select, a CSC phenotype.

For this reason, treated cells should be also studied for the expression of CSC and/or EMT putative markers, and functionally characterized in terms of colony-forming and invasive potential.

- Accordingly, given the possibility for CSCs to exit quiescence and give rise to recurrence, the authors should investigate the possibility to transiently target Pum1/2 and give the chance to spared cells to regrowth, to assess their capacity to re-initiate tumour.

- The authors should include in the manuscript the characterization of subcutaneous xenografts generated by CRC cells following the treatment with nanoparticles-encapsulated PUM siRNA. In particular, they should perform histological and immunophenotypical analysis, in terms of proliferative potential, apoptosis, CSC and EMT markers, of mouse avatar xenografts generated by survived cells.

- The synergistic score of Pum1 and Pum2 targeting should be also validated in vitro.

Reviewer #3 (Remarks to the Author):

In the manuscript "PUMILIO proteins in Colorectal Cancer: Tumor Growth Promoting Function and Potential as Therapeutic Targets" Gong, Liu, Yuan, Yang, and colleagues present a comprehensive analysis of Pumilio protein posttranscriptional gene regulatory role in colorectal cancer cell lines, as well as in xenograft tumor models and Pum1/Pum2 knockout mice. The authors identify increased expression of PUM1 and/or PUM2 in colorectal cancers and find that in model cell lines their knockdown results in reduced cell cycle progression and tumorigenicity. They use RNA-seq and PAR-CLIP to identify regulated target RNAs on a transcriptome-wide scale and zero in and validate cell cycle related genes as top Pum targets responsible for the phenotype. Finally, they show that in vivo knockdown of Pum1 and/or Pum2 using nanoparticle-encapsulated siRNAs results reduced colorectal cancer in mouse models. This study is the most comprehensive analysis of Pumilio protein targets and function in mammals, spanning transcriptome-wide experiments in cell culture, coupled with tissue-specific mouse knockouts and murine tumor models. The combination of well-executed experiments allows a compelling picture of Pumilio proteins as potential oncogenes to emerge. There are a few points the authors may consider addressing before publication:

1. Figure 2: Considering that PUM1 and PUM2 are close paralogs and recognize the same sequence element it stands to reason to expect that they would compensate for each other and arguably, the subtle differences in the effect of Pum1 or Pum2 knockdown could be due to partial

compensation by different expression levels of the paralog. Thus, the posttranscriptional effect of Pumilio proteins could be exacerbated and more clearly revealed by Pum1/2 double knockout (DKO). Perhaps the authors could consider generating such DKO cell lines for transcriptomic and proteomic analyses.

2. Figure 3: Are the differences in proteome/transcriptome changes between Pum1/Pum2 KO qualitative or just due to differences in P-values from RNAseq analysis? Perhaps the authors could present heatmaps or scatterplots of gene expression values from Pum1 and Pum2 KO cells to allow insights into whether the proteins indeed regulate the same set of transcripts. This could also allow the authors to check whether the subtle differences in cell cycle gene expression in Figure 5E are indeed representative of a gene regulatory difference of the two Pum paralogs.

3. Figure 3A/B: It would be nice to see a scatterplot of log fold changes from proteomic and transcriptomic experiments after Pum1 or Pum2 KO to show whether protein changes are mainly due to transcript changes, which would confirm that Pum proteins most likely function by changing transcript levels (as the literature suggests).

4. Figure 4: The PAR-CLIP experiments and their integration with RNAseq could be presented more clearly. E.g. are the sites in Fig 4A target sites found by PARalyzer? How well-correlated are the two PAR-CLIP replicates (e.g. scatterplot of number of crosslinked reads per gene with corresponding correlation factor)? How many binding sites contain the MEME-derived motif? The authors could also take advantage of the semi-quantitative nature of PAR-CLIP and integrate with RNAseq in a manner that would reveal the regulatory effect of Pum; i.e. by binning PAR-CLIP targets based on number of crosslinked reads per gene and then showing the histogram of log-fold transcript changes after Pum KO (both Pum1 and Pum2) on the different bins (see e.g. Yamaji et al., Nature, 2017 for illustration).

5. Figure 6: Here the authors characterize the interaction of AGO proteins with PUM proteins. It is clear that they were inspired for this analysis by previous literature (e.g. Kedde, Nat Genet); that being said, the data don't conclusively show that these proteins interact, in fact, the data does show that the interaction is RNA bridged, which can be expected between almost any two 3'UTR binding RNA binding proteins. Why did the authors not consider examining the interaction between Pumilio and the CCR4-NOT deadenylase complex as the effector complex? Such an interaction is well-established in invertebrates and is clearly conserved in humans (e.g. Enwerem et al., RNA, 2021). The authors could check whether siRNA knockdown of CNOT1 and CNOT7 abrogates Pumilio-mediated target repression.

Minor:

1. I would encourage the authors to thoroughly proofread the text (as well as figure panels) again to eliminate the large number of typos or inadvertent grammar mistakes (e.g. l66 should read as "lead to genome instability"; l94: either the and or the but needs to be removed).

2. l49: reference missing that supports the idea that posttranscriptional gene regulation is recognized to play an important role in CRC.

3. l83: How does the Pumilio expression in CRC compare to expression in other human tissues and during development? Is it markedly higher than "normal" peak expression?

4. l95: it would be good if the authors could add a few words explaining the ROC analysis and what it shows. Not all readers will be familiar with it.

5. l102: it would be nice to explain how the tumor is induced in the AOM/DSS model.

6. l102-125: The section on Pum1/Pum2 mouse KO in the AOM/DSS could benefit from some mild rewriting. The authors first describe the effect of Pum KO on cancer progression and only towards the end of the section present the controls showing that Pum1/2 cKO does not impact intestinal

development. Consider switching the order to prevent any confusion on the reader's part.

7. l454: the authors could comment on why the nanoparticles would be expected to enrich at the tumors. Unless I missed something they don't appear to be modified in a way that would guide them there.

I. Reviewer 1's comments & Response

1. This study investigates the relationship of two RNA binding proteins, PUM1 and PUM2, to colorectal cancer (CRC). Previous studies link these proteins to several cancers, including colorectal cancer, and showed that they bind and regulate cancer genes. Based on existing evidence, PUM1 and PUM2 do not appear to cause cancer, but may modify cancer phenotypes. PUMs are reported to promote proliferation in a number of contexts, including cancer (PMID: 28232582, PMID: 20818387; PMID: 30811992; PMID: 29428722; PMID: 27758885). The potential role of PUMs in CRC has not been extensively analyzed, though at least one publication provides evidence of their relevance (PMID: 33508364). In this study, the authors first correlate over-expression of PUM1 and PUM2 to human CRC. While a few of the observed correlations are intriguing, overall the data and statistical analysis are not very convincing and the authors make several unsubstantiated conclusions (see below). They create the hypothesis that overexpression of PUMs promotes cancer; however, they do not actually test this hypothesis. Their cancer phenotype assays instead focus on loss of function approaches, primarily knockout. (Presently, there is no compelling evidence that loss of PUM1 or PUM2 function is related to cancer susceptibility in humans or mouse models.)

Response: We agree with these incisive comments and have strengthened our data, improved statistical analysis, and corrected unsubstantiated conclusions, including changing “promotes cancer” to “important for cancer progression” or similar statements in the title, abstract, and throughout the text. More specific information is provided below. In addition, we have cited and discussed these literatures mentioned by the reviewer in our revised manuscript ^[1-6] (Please refer to references 6, 21, 23, 24, 40, 53).

[1] Naudin C et al. PUMILIO/FOXP1 signaling drives expansion of hematopoietic stem/progenitor and leukemia cells. *Blood*. 2017, 129(18):2493-2506.

[2] Kedde M et al. A Pumilio-induced RNA structure switch in p27-3' UTR controls miR-221 and miR-222 accessibility. *Nat Cell Biol*. 2010, 12(10):1014-1020.

[3] Lin K et al. Mammalian Pum1 and Pum2 Control Body Size via Translational Regulation of the Cell Cycle Inhibitor Cdkn1b.

Cell Rep. 2019, 26(9):2434-2450.e6.

[4] Guan X et al. PUM1 promotes ovarian cancer proliferation, migration and invasion. *Biochem Biophys Res Commun.* 2018, 497(1):313-318.

[5] Miles WO et al. Alternative Polyadenylation in Triple-Negative Breast Tumors Allows NRAS and c-JUN to Bypass PUMILIO Posttranscriptional Regulation. *Cancer Res.* 2016, 76(24):7231-7241.

[6] Gor R et al. RNA binding protein PUM1 promotes colon cancer cell proliferation and migration. *Int J Biol Macromol.* 2021, 174:549-561.

2. The authors use a conditional PUM1/2 knockout mouse model and human CRC cell lines to investigate the contribution of PUMs to CRC. Using a mouse model for Azoxymethane (AOM)/Dextran Sodium Sulfate (DSS) induced inflammatory CRC to investigate the effects of PUM1 and PUM2 on phenotypes, they found that, upon PUM1/2 knockout in the intestinal epithelia, the number of colon tumors and proliferation of tumor cells were reduced. The double knockout mice had somewhat longer colons and increased body weight. KO of PUM1 and PUM2 was confirmed by immunohistochemistry. Immunohistochemistry of a proliferation marker indicated that the double knockout reduced proliferation in the tumors. The authors state that apoptosis was increased; however, their data in Figure 1M shows that this difference was not statistically significant. The authors report that the morphology of the small intestine in the double knockout mice was not affected. This is surprising, since Uyhazi et al 2019 reported that PUM1 knockout mice had disrupted small intestine morphology. A pertinent control for these experiments would be to include analysis of the colon morphology, length, and body weight of WT and double PUM knockout mice in the absence of AOM and DSS.

Response: There seems to be a misunderstanding about our statement here. Although the TUNEL signal improved slightly, conditional PUM1/2 knockout mice did not have a significant impact on apoptosis. To avoid misunderstanding, we have changed the description from “Furthermore, Ki-67 and TUNEL staining revealed that *Pum1/2^{CKO}* dramatically reduced tumor cell proliferation and slightly increased apoptosis” to “Furthermore, Ki-67 and TUNEL staining revealed that *Pum1/2^{CKO}* dramatically reduced tumor cell proliferation but had no significant effect on apoptosis (Please refer to Page 5)”.

To further examine the morphology of the small intestine in the double knockout mice, we compared the colon morphology, length, and body weight of WT and double PUM knockout mice in the absence of AOM and DSS as reviewer suggested. Mice of control and *Pum1/2^{CKO}* group with same age and gender were generated as described in the manuscript (Page 4). Consistently, there were no significant changes between the control and *Pum1/2^{CKO}* group in colon morphology (Response Figure 1A, B). In addition, there is no significant change in either colon length (Response Figure 1C) or body weight (Response Figure 1D) of these two groups in the absence of AOM/DSS. The difference between our *Pum1/2^{CKO}* phenotype and the *Pum1^{KO}* phenotype by Uyhazi *et al* (2019) could be due to that the global *Pum1^{KO}* (generated by breeding *Pum1^{Flox/+}* mice with *EIIa-Cre* mice as described in Uyhazi *et al.* [1]) may be caused by a role of PUM1 in early embryonic gut development. Alternatively, it is possible that PUM1 and PUM2 affect complementary (or opposing aspects) of gut development, just like in ESCs, so that the double mutant phenotype could be milder than *Pum1^{KO}*.

[1] Uyhazi KE et al. Pumilio proteins utilize distinct regulatory mechanisms to achieve complementary functions required for pluripotency and embryogenesis. *Proc Natl Acad Sci U S A.* 2020;117(14):7851-7862.

Response Figure 1. (also added to Revised Fig. S2) Re-examination the morphology of *Pum1/2^{CKO}* intestine. **(A)** The colon and rectum of control (*Pum1^{fllox/fllox}::Pum2^{fllox/fllox}*) and *Pum1/2^{CKO}* (*Lgr5^{cre}::Pum1^{fllox/fllox}::Pum2^{fllox/fllox}*) mice in the absence of AOM/DSS. **(B)** Representative micrographs of the colon in control and *Pum1/2^{CKO}* mice in the absence of AOM/DSS, as revealed by H&E staining. **(C)** The colon length of control and *Pum1/2^{CKO}* mice in the absence of AOM/DSS. Error bars represent SD. ns: not significant, Student's t-test. **(D)** The weight of control and *Pum1/2^{CKO}* mice in the absence of AOM/DSS. Error bars represent SD. ns: not significant, Student's t-test.

3. The authors go on to assess PUM1 and PUM2 mRNA and protein expression in several human colon cancer cell lines compared to a single “normal” cell line. They attempt to correlate this to cancer but doing so is superficial and not convincing. In fact, a previous report and public data show that PUM1 and PUM2 are broadly expressed in normal and cancer cell lines and tissues (e.g., Spassov and Juresic 2002 Gene, Cancer Cell Line Encyclopedia, Genotype-Tissue Expression (GTEx) database).

Response: We agree with the reviewer that PUM1 and PUM2 are broadly expressed in cancer cell lines (Cancer Cell Line Encyclopedia) and tissues (GTEx database), as reported by Spassov and Juresic (2002) by dot-blot assay of cancer cells without comparison to corresponding tissues^[1]. We apologize for omitting this citation, and have added it to our revised manuscript (Page 3). There are two novel aspects of our findings: First, we showed that PUM1 and PUM2 are expressed at higher-than-normal levels in human colorectal cancer samples from 22 patients compared with their adjacent tissues. Second, our analysis of cancer genome atlas (TCGA) datasets revealed PUM1 but not PUM2 mRNA is expressed at higher-than-normal levels in two types of colorectal cancer cell lines (Hong colorectal and Skrzypczak colorectal), as compared with normal colon tissues. We wish we have more normal cell lines to compare but this was limited by resource. Comparing to one is still more informative than no comparison. Most recently, Gor et al (2021) detected a higher level of PUM1 mRNA in colorectal cancer cell lines as compared to another normal cell line, FHC^[2], which supports our observations. We have now edited our statements throughout the text to make our finding clearer for readers.

[1] Spassov DS et al. Cloning and comparative sequence analysis of PUM1 and PUM2 genes, human members of the Pumilio family of RNA-binding proteins. *Gene*. 2002, 299(1-2):195-204.

[2] Gor R et al. RNA binding protein PUM1 promotes colon cancer cell proliferation and migration. *Int J Biol Macromol*. 2021,

4. The authors then studied the effect of PUM1 and PUM2 loss of function in HCT116 cells, a highly proliferative cancer cell line that is transformed by mutant RAS oncogene. In Fig S3, they report that RNAi depletion of either PUM1, PUM2 or both reduces proliferation. PUM1 has a small increase effect on cells in G1. In the results, the authors state that “we found that PUM1 and/or PUM2 knockdown displayed a significantly increased portion of cells in G0/G1 phase” However, this is not accurate because the effect of PUM2 knockdown was not significant (Fig. S3G).

Response: We thank the reviewer for catching this inaccurate statement and have now changed the text as recommended, to be more precise on this point (Please refer to Page 7).

5. The authors then knockout PUM1 or PUM2 in the HCT116 cell line and find that colony formation and proliferation are reduced by PUM1 knockout in two clonal lines. Similar observations are made in the RKO cell line. Strangely, the PUM2 knockout has variable effects in both cell lines. One clonal knockout line showed a significant effect on cell proliferation, colony formation, and cell cycle, whereas the other line did not. This is concerning and could be indicative of off target effects of the CRISPR knockout. The apparent discrepancy in the effects of RNAi and knockout of PUM2 on cell proliferation and colony formation is concerning. The authors report 5-6 clonal lines with PUM1 or PUM2 knockout in the supplement with corresponding genotypes. It's not clear which of these were used in Figure 2. It's unclear why only two of these lines were tested for changes in proliferation and colony formation. Doing so could help assess the variability of the PUM2 knockout effects.

Response: We are grateful for this suggestion and are also worried about the variable effects of PUM2 knockout has in the two cell lines. In order to address whether the variability is caused by the off-target effect, we introduced WT *PUM2* into *Pum2*^{-/-} cells by transfection and found that cell proliferation and colony formation defects are rescued (Response Figure 2), these lines of evidence indicate that the defects of *Pum2*^{-/-} is dependent on PUM2 loss, but not off-target effect.

Response Figure 2. Testing the off-target effect of PUM2 knockout. **(A)** Growth curve of WT+Con (pcDNA3.1 empty vector), *Pum2*^{-/-}+Con and *Pum2*^{-/-}+Pum2 (pcDNA3.1 with *Pum2* gene vector) in HCT116 cells (n=3). Error bars represent SD. ns: not significant, **P* < 0.05, Student's t-test. **(B)** Colony formation assay of WT+Con (pcDNA3.1 empty vector), *Pum2*^{-/-}+Con and *Pum2*^{-/-}+Pum2 (pcDNA3.1 with *Pum2* gene vector) in HCT116 cells (n=3). Error bars represent SD. ns: not significant, **P* < 0.05, Student's t-test. **(C)** Cell cycle analysis of WT+Con (pcDNA3.1 empty vector), *Pum2*^{-/-}+Con and *Pum2*^{-/-}+Pum2 (pcDNA3.1 with *Pum2* gene vector) in HCT116 cells (n=3). Error bars represent SD. ns: not significant, **P* < 0.05, Student's t-test.

To indicate more clearly which clones were used in Figure 2, we have now added the sample numbers to the legends of Figure 2 (Please refer to Page 10). In the submitted version, we only tested two of PUM1/2 knockout clones from HCT116 and RKO because defects in all of the eight knockout cell lines (2 PUM1 knockout clones from HCT116, 2 PUM2 knockout clones from HCT116, 2 PUM1 knockout clones from RKO and 2 PUM2 knockout clones from RKO) indicated that PUM proteins contribute to colorectal cancer cell growth *in vitro*. However, we agree that if more clones are tested for proliferation and colony formation, the data will be more reliable. Therefore, we have now added one more cell line for each knockout condition. Consistently, HCT116 *Pum1*^{-/-}-3, HCT116 *Pum2*^{-/-}-3 (Response Figure 3A-B), RKO *Pum1*^{-/-}-3 and RKO *Pum2*^{-/-}-3 (Response Figure 3C-D) caused decreased cell proliferation and colony formation.

Response Figure 3. Testing new clones of PUM1/2 knockout HCT116 and RKO cell lines. **(A)** Growth curve of WT, *Pum1*^{-/-}-3 (#3) and *Pum2*^{-/-}-3 (#5-2) in HCT116 cells (n=3). Error bars represent SD. **P* < 0.05, Student's t-test. **(B)** Colony formation assay of WT, *Pum1*^{-/-}-3 (#3) and *Pum2*^{-/-}-3 (#5-2) in HCT116 cells (n=2). Error bars represent SD. ns: not significant, **P* < 0.05, Student's t-test. **(C)** Growth curve of WT, *Pum1*^{-/-}-3 (#3-11) and *Pum2*^{-/-}-3 (#2-9) in RKO cells (n=3). Error bars represent SD. ****P* < 0.001, Student's t-test. **(D)** Colony formation assay of WT, *Pum1*^{-/-}-3 (#3-11) and *Pum2*^{-/-}-3 (#2-9) in RKO cells (n=3). Error bars represent SD. ns: not significant, **P* < 0.05, Student's t-test.

6. Further, the authors should test the effect of knockout of both PUM1 and PUM2 simultaneously. The ability of PUM1 or PUM2 transgenes to rescue this proliferation defect would provide additional support.

Response: We thank the reviewer for this suggestion. To generate *Pum1*^{-/-}; *Pum2*^{-/-} double knock out cells, pGL3-U6-sgRNA-PGK-puromycin (Addgene plasmid, #51133) expressing a pair of sgRNAs used for single *Pum2* knockout was transfected into *Pum1*^{-/-} cells followed by puromycin selection. The protein level of PUM2 was reduced about 65% in *Pum2* sgRNAs transfection pool compared with *Pum1*^{-/-} (Response Figure 4A). First, 176 single cell clones were screened by genome PCR (Response Figure 4B). Then, 38 single clones with PCR bands shift were screened by western blot and none of them showed PUM2 protein depletion (Response Figure 4C), even though these clones caused frameshift according to DNA sequencing results. It's possible that these clones re-initiate translation of *Pum2* from another ATG nearby the original ATG. Notably, about 20 clones were dead during passage, thus we couldn't identify their genotypes. We have tried twice to generate *Pum1/2* double knockout HCT116 cells (more than 400 single cell clones), but still could not recover *Pum1/2* double knockout HCT116 cells. Together, these results suggest that PUM1 and PUM2 double knockout cause severe cell lethality thus it is not feasible to analyze double-knockout cell lines.

Response Figure 4. Attempts to knockout both PUM1 and PUM2 simultaneously. **(A)** Western blot showing the PUM2 protein levels in Pum1 KO and Pum1 KO cells transfected with Cas9/sgRNAs that targeted all isoforms of *Pum2*. Actin was used as a loading control. **(B)** DNA agarose gel showing the representative clones genotyped by genome PCR. Red line indicates *Pum2* WT alleles, the bands above the red line indicates mutant alleles. **(C)** Western blot showing the PUM2 protein levels in representative potential *Pum1* and *Pum2* double knockout clones. GAPDH was used as a loading control.

7. The authors then attempt to identify the genes that are regulated by either PUM1 or PUM2. They performed RNA seq to measure differential levels of mRNAs in response to PUM1 or PUM2 knockout. Tandem mass tagging based proteomics was used to measure relative changes in protein levels. The authors also performed crosslinking and immunoprecipitation to identify mRNAs bound by PUM1 (PUM2 was not analyzed). Unfortunately, from this large amount of data, only a cursory analysis is provided. As a result, it is not clear what, if any, new insights these datasets provide. Instead, the presented results reaffirm what is already well-documented in the literature - that PUM proteins bind to a PRE site that is typically located in 3'UTR of mRNAs, and that PUMs regulate genes involved in cell cycle and proliferation.

Response: We have actually done quite extensive bioinformatic analysis to characterize the sequence and functional features of PUM-target mRNAs, trying to discover some novel regulation. However, as the reviewer pointed out, our data indeed indicate that PUM proteins in colorectal cancer cells regulate mRNAs involved in cell cycle and proliferation via PRE sites. These data further support the function of PUM proteins in cell proliferation, colony formation and cell cycle. Since we did not find novel regulation mechanism, we did not put all less important analysis to the paper, so not to unnecessarily burden the readers we did not put all analysis.

8. The results of the global approaches are problematic for several reasons, including lacking rigor, evidence of reproducibility, and proper documentation. For the RNA-seq analysis, the authors report thousands of genes with fold change and p-values. However, reviewing their data (Tables S2 and S3) shows that hundreds of these genes have values (RPKM? TPM?) that are 0 in one or more samples. Thus, the

reported fold change for these genes are not valid. The reproducibility of the two replicate samples is not assessed. Concern about reproducibility is heightened by the heat maps in Fig. 4K, which indicate large differences between the replicates. Two replicates are the bare minimum for differential expression analysis. The authors should assess the Biological Coefficient of Variation (BCV) for replicates within each condition. Principal component analysis (PCA) or multidimensional scaling (MDS) analysis across conditions and replicates is necessary to determine if the results are reliable.

Response: We thank this reviewer for his/her careful review of our data. The values we showed in Tables S2 and S3 were raw counts. Raw counts for a gene can be zero under some conditions (and not zero in others), We used pseudocounts for transformations of the form: $y = \log_2(n+n_0)$, where n represents the count values and n_0 is a positive constant.

To assess the reproducibility of the replicates, principal component analysis (PCA) was used to visualize the variation among different genotypes based on normalized read counts (Response Figure 5). Based on the random clustering of replicates in the PCA plot, we assume a high degree of uniformity among the replicates. The PCA results showed that the first principal component explained 76% of the total variation in our experimental data and clearly separated the *Pum1/2*-mutant datasets from the WT control group.

We understand that it's definitely better to have three or more replicates because two replicates capture fewer biological variations than three replicates so there may be more false positives. However, the main purpose of this paper is the analysis of downstream targets, our finding is not omics driven, we hope the reviewer will agree that real-time PCR and western blot verification following with the multi-omics data can better ensure that the problem of false positives in our downstream targets has been adequately solved. We realized that we really should involve more replicates and will include 3 or more replicates in our future analyses.

Response Figure 5. Principal component analysis of RNAseq data base on normalized read counts. Sample-to-sample distances (within and between genotypes) are illustrated for each dataset on the first two principal components comprising approximately 90% of the variation. Samples are plotted according to genotype.

9. The RNA seq methodology is not adequately described including how RNAs were isolated, whether rRNA was depleted, or how libraries were generated. The authors should adhere to the standards set forth in the ENCODE Guidelines and Best Practices for RNA-Seq.

Response: We thank the reviewer for this suggestion, and have made it clear in the Materials and Methods section (Please refer to Page 28-29).

10. Documentation of the quantitative relative proteomics methodology and data analysis is completely

inadequate. The methods section provides only two sentences. The Supplemental table reports values that are not defined. The primary data should be deposited to a publicly accessible database such as ProteomeXchange.

Response: We thank the reviewer for spotting this deficiency and have now added a detailed description of the quantitative proteomic method in Materials and Methods (Please refer to Page 29). We have also made it clear in Table S4 and S5 that the values in these two tables represent relative abundance of ions. The mass spectrometry proteomics data have been deposited to the ProteomeXchange Consortium via the iProX partner repository ^[1] with the dataset identifier (PXD027513/IPX0003309000). The share URL: <https://www.iprox.cn/page/PSV023.html?url=1627023666801n6XD>, Password: UZ2W.

[1] Ma J, Chen T, Wu S, Yang C, Bai M, Shu K, Li K, Zhang G, Jin Z, He F, Hermjakob H, Zhu Y. iProX: an integrated proteome resource. *Nucleic Acids Res.* 2019, 47:1211-1217.

11. The results of the quantitative mass spectrometry data (Table S4 and S5) fail to provide any type of statistical analysis that is necessary to assess the reproducibility of the approach and statistical significance of the results. The values reported in the data table are not defined. Again, concern about reproducibility is heightened by Figure 4L, where the heat map shows large differences between the protein level measurements. Again, BCV, PCA or MDS analysis is needed to assess reproducibility.

Response: We thank the reviewer for this suggestion and have added statistical analysis in Revised Tables S4 and S5. To assess the reproducibility of replicates across conditions and replicates, principal component analysis (PCA) was used to visualize the variation among different genotypes based on peptides abundance (Response Figure 6). Based on the random clustering of replicates in the PCA plot, we assume a high degree of uniformity among the replicates. The PCA results showed that the first principal component explained 53% of the total variation in our experimental data and biological replicates of each condition could be clustered together.

Response Figure 6. Principal component analysis of Mass spectrum data base on normalized peptide abundance. Sample-to-sample distances (within- and between-genotypes) are illustrated for each dataset on the first two principal components comprising approximately 70% of the variation. Samples are plotted according to genotype.

12. Reproducibility of the PAR-CLIP data and statistical significance of the results are not addressed. In fact, from what is presented for PAR-CLIP in Figure 4A, there is substantial variation between the two replicates (only about 1/3 of the genes overlap). Proper statistical analysis is necessary (DOI: 10.1214/11-AOAS466; PMID: 27018577).

Response: We are grateful to the reviewer for his/her suggestion on assessment reproducibility and for the recommending powerful statistical methods. Unlike clusters identified by peak calling ^[1] (PMID: 27018577), the statistical significance of PAR-CLIP was not shown in our manuscript because the Paralyzer employs a non-parametric kernel-density estimate classifier ^[2]. Since T to C conversion occurs at the binding site of PUM1 protein and RNA, Paralyzer generates two smoothed kernel density estimates, including T to C transitions and non-transitions. Interaction sites were identified when they meet the criterion (for details, see Materials and Methods section, please refer to Page 30).

In order to further prove the reproducibility of our PAR-CLIP data, we assessed reproducibility by calculating the Spearman correlation of crosslinked reads per gene with corresponding correlation factor (Revised Fig. S8 in the Supporting Information) instead of IDR analysis ^[1]. Overall, a good correlation between the two replicates is shown (Response Figure 7). Due to variation can arise from unnational differences in implementation, such as RNase fragmentation conditions, binding sites in lowly expressed RNAs are hard to reproduce, background reads also contain T to C conversions ^[3], greater variation could be introduced after cluster identification in Figure 4A. To improve the reliability of targets identified by PAR-CLIP, overlapped genes were used in the following analysis.

Response Figure 7 (also as Revised Fig. S8C). Scatterplot of the number of crosslinked reads between the two replicates. X-axis and Y-axis show the log2ConversionCount in replicate 1 and replicate 2, respectively.

[1] Van Nostrand EL et al. Robust transcriptome-wide discovery of RNA-binding protein binding sites with enhanced CLIP (eCLIP). *Nat Methods*. 2016, 13(6):508-14.

[2] Corcoran DL et al. PARalyzer: definition of RNA binding sites from PAR-CLIP short-read sequence data. *Genome Biol*. 2011, 12(8):R79.

[3] Friedersdorf MB et al. Advancing the functional utility of PAR-CLIP by quantifying background binding to mRNAs and lncRNAs. *Genome Biol*. 2014, 15(1):R2.

13. The data analysis is inadequate. The RNA Seq, proteomics, and PAR-CLIP datasets are given the most cursory of analysis. The effects of PUM1 and PUM2 knockout are not compared. The RNA and protein level effects are not directly compared. Illogically, different fold change cutoffs are used for protein and RNA level changes. The GO term analysis in Figure 3C-J and Figure 4 is illegible due to microscopic text. The reported GO term analysis should be reported in a supplementary table that includes the number and identity of the genes in each category along with the statistical significance.

Response: We thank the reviewer for these constructive questions and address these questions in the following four sections.

(1) The reviewer makes a good point about our set of different fold change cutoffs for mRNA and protein level changes. Our cutoff criteria are as follows: For mRNA: $FDR < 0.1$, $FC \geq 1.5$ or $FC \leq 1.5$ ($\log_2 FC \geq 0.585$ or $\log_2 FC \leq -0.585$). For protein: $FC \geq 1.2$ or ≤ 1.2 ($\log_2 FC \geq 0.263$ or $\log_2 FC \leq -0.263$), which is a generally accepted cutoff of fold change for proteins by the Tandem Mass Tags (TMT) method [1,2]. This is because the TMT method yields non-linearly compressed values in indicating the protein level difference. In addition, there is no algorithm to linearize the data, so it cannot be directly compared to certain RNA fold change. This is unfortunately the current state of the mass spec field. Finally, we did not consider the *P* value of the protein mass spectrum data because we had only two replicates (We list several proteins under this condition as an illustration, see the table below). Omics methods are high-throughput. Regardless of the defined multiple standards, the important issue is whether the results can be validated by independent methods. For example, p21's up-regulation in PUM1 KO cells revealed by mass spec is validated by western blot validation, whether the *P* value is considered or not.

Gene	WT-1	WT-2	Pum1 ^{-/-} -1	Pum1 ^{-/-} -2	Fold change	P value
SNTB1	0.98	0.84	2.11	2.77	2.7	0.08
NEBL	0.93	0.85	1.84	2.71	2.6	0.11
SLC23A2	1.15	1.69	4.24	2.59	2.4	0.16
CDC42BPA	0.98	0.82	1.97	2.22	2.3	0.06
NQO1	0.99	1.00	1.92	2.53	2.2	0.08
LIMD2	1.04	1.23	1.91	3.05	2.2	0.11
LAMP2	0.97	0.96	2.50	1.62	2.1	0.12
CRIP1	0.98	1.33	1.75	2.96	2.0	0.11
PLEKHA2	0.85	0.91	1.58	1.96	2.0	0.06
SERPINE1	1.02	1.77	2.85	2.79	2.0	0.09
PDE5A	1.67	1.85	3.94	3.12	2.0	0.09
ACBD7	1.27	0.97	2.01	2.46	2.0	0.10
BCAM	1.08	0.97	1.93	2.19	2.0	0.06
PDLIM2	1.05	0.84	1.78	1.93	2.0	0.06
HOXB7	0.92	0.91	1.61	1.97	2.0	0.06

[1] Chai YN et al. TMT proteomics analysis of intestinal tissue from patients of irritable bowel syndrome with diarrhea: Implications for multiple nutrient ingestion abnormality. *J Proteomics*. 2021, 231:103995.

[2] Zhang C et al. TMT-Based Quantitative Proteomic Analysis Reveals the Effect of Bone Marrow Derived Mesenchymal Stem Cell on Hair Follicle Regeneration. *Front Pharmacol*. 2021, 12:658040.

(2) With the above defined criteria, we identified 1,132 and 1,226 significantly differentially expressed genes at the mRNA level upon *PUM1* and *PUM2* knockout, respectively (Revised Table S2 and S3). Changes in gene expression are quite similar between *PUM1* knockout and *PUM2* knockout, with a 0.78 correlation at the mRNA level and 0.70 correlation on the protein level, respectively (Response Figure 8). *PUM1* knockout resulted in slightly greater transcriptional changes in comparison to *PUM2* knockout. On average, the transcriptional fold change for the upregulated genes is 3.3 (following *PUM1* knockout) and 2.3 (following *PUM2* knockout), and for the downregulated genes 4.1 (*PUM1* knockout) and 2.3 (*PUM2* knockout). With respect to individual genes, fold change is also significantly greater following *PUM1* knockout in comparison to *PUM2* knockout both on the mRNA level (Wilcoxon rank sum test *P*-value=1.943e-06) and protein level (Wilcoxon rank sum test *P*-value=1.658e-11) (Response Figure 9).

Response Figure 8. (also added to Revised Fig. 3) Changes in the mRNA level and the protein level upon PUM1 knockout correlate well with those upon PUM2 knockout. **(A)** and **(B)** show the scatterplots comparing the changes (using log2FoldChange in knockout versus in wild type) between PUM1 knockout (X-axis) and PUM2 knockout (Y-axis) at the mRNA level (A) and protein level (B).

Response Figure 9. (Revised Fig. S7 in the manuscript) Boxplots showing greater transcriptional and translational changes upon *PUM1* knockout compared to *PUM2* knockout. Left and right panels show boxplots comparing the log2FoldChange (knockout genotype versus the wild type) between *PUM1* knockout (red boxplot) and *PUM2* knockout (blue boxplot).

(3) We plotted the mRNA and protein log2FoldChange (Response Figure 10) following *PUM1* knockout and *PUM2* knockout, respectively. The correlation of gene expression changes on the mRNA level and RNA level for PUM1 knockout is 0.7 (Response Figure 10A) and for PUM2 knockout 0.76 (Response Figure 10B). This suggests that protein level changes, to a large extent, result from transcriptional changes.

Response Figure 10. (also added to Revised Fig. 3) Scatterplot of log₂FoldChange on mRNA and protein level upon PUM1 (A) or PUM2 (B) knockout. X-axis and Y-axis show the log₂FoldChange on the mRNA and protein level (X- and Y-axis respectively).

(4) When integrating PAR-CLIP data and mRNA-Seq data, we looked into the relationship between the number of crosslinked reads by gene and fold change of gene expression resulting from *Pum1* knockout. The analysis of PAR-CLIP data identified 302 genes as PUM1's direct targets. Among them, only 28 genes have statistically significant changes in expression (FDR<0.1) at the mRNA level upon *Pum1* knockout. We binned the 28 targets into two groups, one group with weaker PUM1 binding (10 genes with ConversionCount≤8) and the other with stronger binding (18 genes with ConversionCount>8). Then within each group, we plotted their fold change according to their altered expression pattern (Upregulated or Downregulated) upon *Pum1* knockout. The upregulated genes did not show statistically significant correlation between fold change and PUM1 binding strength by Wilcoxin rank sum test (Response Figure 11). This is likely due to an extremely small number of genes that went into this analysis. However, among the downregulated genes, a greater fold change is correlated with stronger binding (Wilcoxin rank sum test P-value=0.022) (Response Figure 11).

Response Figure 11. Relationship between binding affinity by PUM1 (PAR-CLIP) and its altered level of expression upon *Pum1* knockout (RNA-Seq). X-axis shows the log₂ConversionCount, an indicator of the

binding strength by PUM1, Y-axis shows the altered level of expression upon *Pum1* knockout.

For the KEGG enrichment analysis in Figures 3 and 4, we have increased the font size and placed the details in revised Table S6 and S8.

14. The overlap of genes affected at either RNA or protein levels reported in Figure 4I and 4J show that the vast majority of genes in either category are likely indirect effects, or that there is a high incidence of false positives in the CLIP, RNA seq, and TMT datasets.

Response: Although it is generally believed that PUM proteins regulate all of their target genes, when Lapointe *et al* used RNA tagging approach to identify RNAs bound by a PUM homolog in *Saccharomyces cerevisiae*, Puf3p, they found Puf3p binding to many RNAs without detectable regulatory effect^[1]. Similarly, a recent study in mouse embryonic stem cells from our lab found less than 5% of PUM1 and PUM2 targets exert translational up-regulation^[2], which means not all target mRNAs are regulated by PUM proteins, in contrast to common belief.

[1] Lapointe CP et al. Protein-RNA networks revealed through covalent RNA marks. Nat Methods. 2015, 12(12):1163-70.

[2] Uyhazi KE et al. Pumilio proteins utilize distinct regulatory mechanisms to achieve complementary functions required for pluripotency and embryogenesis. Proc Natl Acad Sci U S A. 2020, 117(14):7851-7862.

15. Overall, seemingly little new information is extracted from the RNA seq, proteomics, and PARCLIP datasets. The authors arbitrarily selected CDKN1A/p21 mRNA from the list. In fact, when the differentially expressed mRNAs are ranked by fold change, CDKN1A is not even in the top 1000 of affected genes. This biased selection is not explained or justified, and could substantially miss the most important regulated genes. CDKN1A was already known to be a PUM target mRNA; previous research showed that PUMs bind and regulate CDKN1A mRNA (PMID: 18411299; PMID: 18776931; PMID: 30811992).

Response: Thanks for the comment. If selected on the fold change of mRNA or protein, it is true that CDKN1A/p21 does not rank highly, but it was selected to study for the following three reasons:

First, when the results of RNA and protein profiling are combined with the PUM1 PAR-CLIP data, p21 is the only one associated with the cell cycle among the direct PUM1-target genes that are regulated by PUM1 at both mRNA and protein levels (Revised Fig. 4J-4M). We therefore chose p21 for further study.

Second, p21 has been consistently identified by three other groups as a direct PUM1 target as well. Morris et al.^[1] identified p21 as one of PUM1-associated mRNAs in human HeLa cells by PUM1 RIP-chip, even though they did not study the regulatory effect of the binding. Galgano et al.^[2] reported genome-wide identification of mRNAs bound by both PUM1 and PUM2 and also found p21 mRNA associates with both proteins. Moreover, Lin et al.^[3] showed that protein level of p21 is upregulated in *Pum1*^{-/-} mouse embryonic fibroblasts, even though they did not show that PUM1 proteins are associated with p21 mRNA in their system.

Third, our data reveal that *Pum1* directly binds and represses p21 via 3' UTR in human colorectal cancer cells. Furthermore, our functional assays performed in p21 PRE-mutated cells indicate p21 is a key downstream gene of PUM1. Therefore, we selected p21 as a PUM-target gene to focus.

[1] Morris AR et al. Ribonomic analysis of human Pum1 reveals cis-trans conservation across species despite evolution of diverse mRNA target sets. Mol Cell Biol. 2008, 28(12):4093-103.

[2] Galgano A et al. Comparative analysis of mRNA targets for human PUF-family proteins suggests extensive interaction with the miRNA regulatory system. PLoS One. 2008, 3(9):e3164.

[3] Lin K et al. Mammalian Pum1 and Pum2 Control Body Size via Translational Regulation of the Cell Cycle Inhibitor Cdkn1b. Cell Rep. 2019, 26(9):2434-2450.e6.

16. Based on the RT and qPCR analysis in the PUM1 or PUM2 knockout cells, it appears that only PUM1 knockout leads to an increase in p21 mRNA (Fig. 5). Qualitative western blot data of p21 is also provided in Fig 5B; however, quantitative western blot analysis from multiple replicates would be more convincing, and could assess fold change increase in protein levels along with statistical significance.

Response: We agree with this comment, and have now added quantitative western blot data from three biological replicates to the revised version (Response Figure 12, please also refer to revised Fig. 5B).

Response Figure 12 (Revised Fig. 5B in the manuscript). Quantification of p21 protein levels in Pum mutant cancer cells.

17. While p21 is an attractive target because it has well documented tumor suppressor and cell cycle functions, several inconsistencies limit its importance to PUM regulation of proliferation. Both PUM1 and PUM2 affect proliferation and have increased expression in CRC. Given that both PUMs bind to PREs to inhibit genes, it seems that a shared set of mRNAs would be of vital importance. Yet the authors observe that only PUM1 inhibits p21. The authors wish to conclude the PUM1 inhibition of p21 is responsible for the control of cell proliferation, seemingly at odds with both PUM1 and PUM2 controlling proliferation. Additionally, it is well documented that PUM1 and PUM2 bind to the same PREs, and the authors confirm that both PUMs bind to p21 mRNA in Figure 5, Thus, the contention that only PUM1 inhibits p21 to control proliferation is paradoxical.

Response: As described above, not all target mRNAs are regulated by PUM proteins, in contrast to common belief. Our data indicate that PUM2 binds to *p21* mRNA but does not regulate it. Although both PUM1 and PUM2 affect proliferation, it is possible that they affect proliferation by regulating different genes. Our data indicate that p21 is likely a key gene downstream of PUM1 but not PUM2.

18. The authors attempt to analyze the effect of PUM1 and PUM2 on degradation of the p21 mRNA by measuring mRNA decay following transcription shutoff using actinomycin D. The analysis is flawed because they did not measure the half life of the mRNA in the different conditions. Instead, they appear to show only a significant difference between the final time points. This is not acceptable. Observed half lives should be calculated, reproducibility should be assessed, and statistical significance of the differential decay should be determined and reported. From the data presented, it is clear that the p21 mRNA is unstable in the absence of both PUM1 or PUM2, indicating that additional dominant mechanisms control its degradation.

Response: We thank the reviewer for the suggestion, and have now added a time course measurement of p21 mRNA decay to determine the half-life under different conditions in our revised version (revised Fig 5E).

Decay curves shown in the different conditions (Fig. 5E) are normalized from three biological replicates, each line is shown in Response Figure 13, which showed excellent reproducibility. we calculated mRNA half-life following equation: $t_{1/2} = \ln(2)/k_{decay}$, where $t_{1/2}$ is the half-life of an mRNA, k_{decay} is the rate constant [1]. Half-lives of p21 mRNA in two *Pum1* KO cells are significantly increased compared with WT, while none of *Pum2* KO cells showed significant change, indicating p21 mRNA is more stable in the absence of PUM1 instead of PUM2. The significant difference in the revised Figure 5E does not represent the final time point, but the difference between the fitting curves. We have made it clear in the legend of Figure 5E.

[1] Chen CY et al. Messenger RNA half-life measurements in mammalian cells, *Methods Enzymol*, 2008, 448: 335-57.

Response Figure 13. The p21 mRNA levels in WT, *Pum1*^{-/-} or *Pum2*^{-/-} HCT116 cells treated with actinomycin

D (10 mg/mL) for the indicated period. Plot represents relative p21 mRNA expression in the indicated time, the fitted curves follow first order kinetics.

19. The role of the PRE in the p21 mRNA was investigated. Its mutation appears to increase p21 protein, but again the lack of quantitative western blot data backed by replicates and statistics tempers the strength of the conclusions. At the mRNA level, the effect size of the PRE mutation does not phenocopy that of the PUM1 knockout (Fig. 5I) and PUM1 still can bind to the p21 PRE mt mRNA (Figure 5J). Therefore, the authors claim that “p21 PRE mut abolishes PUM1-mediated regulation of p21” (p14 line 299) is not supported by their data.

Response: We have added quantitative western blot data to our revised version (Response Figure 14, revised Fig. 5J). At the mRNA level, PRE mutation indeed significantly upregulated p21 expression though not much high as PUM1 knockout. To avoid misunderstanding, we have changed our statement from “p21 PRE mut abolishes PUM1-mediated regulation of p21” to “p21 PRE mut suppresses PUM1-mediated regulation of p21”.

Response Figure 14 (Revised Fig. 5J in the manuscript) Quantification of p21-*PRE*^{mut} expression.

20. The RNA coimmunoprecipitation (RIP) data analysis method is not described in the manuscript and appears to be incorrect (Fig 4E, Fig 5C, D, J). The fold enrichment of the mRNA in each RIP sample should be calculated in each condition relative to the negative control RIP after those values are normalized to the amount of mRNA in the input of that specific sample. This is important to address changes in the input levels of p21 in the WT, PUM knockout, and p21 PRE mut conditions. Statistical analysis of the fold enrichment data must be included to assess reproducibility.

Response: Thanks for the comments. There seems to be a misunderstanding here. For fold enrichment calculation of the mRNA in each RIP sample, we used PUM1/2 IP in knockout (*Pum1/Pum2*^{-/-}-1) cells as the negative control. In addition, we included the input sample to normalize background abundance in order to increase the confidence (see revised Materials and Methods section, Page 28). The reason we showed relative expression of input in each condition is to illustrate that even though p21 expression was higher in *Pum1* knockout cells than in p21 *Pum2* knockout cells (Fig. 4E, revised Fig 5C, D, J), it was still drastically reduced in *Pum1* knockout cells, just like in *Pum2* knockout cells. To avoid misunderstanding, we have now excluded the relatively input expression, since it has been indicated in Figure 5A.

21. The RT-qPCR methods used throughout the study are not described. The methodology and data analysis should be reported in accordance with MIQE guidelines (PMID: 19246619). The authors indicate the RNA was purified by trizol, which notoriously leads to genomic DNA contamination. There is no mention of DNase treatment to remove contamination, nor was it assessed by the proper minus-RT controls.

Response: We highly appreciate the reviewer’s suggestions. Actually, genomic DNA was treated using

DNA-free™ Kit Dnase Treatment and Removal Reagents (#AM1906) according to the protocol of the manufacturer [1]. As suggested, we have made this procedure clear in our revised Materials and Methods section (Please refer to Page 27).

[1] Bustin SA et al. The MIQE guidelines: minimum information for publication of quantitative real-time PCR experiments. Clin Chem. 2009, 55(4):611-622.

22. Mutation of the PRE in the p21 gene is shown in Figure 5K-O to reduce proliferation, slightly increasing doubling time, and the portion of the cells in G1 phase. This is some of the most interesting data in the manuscript. Strengthening the preceding analysis of PUM binding and inhibition of p21 protein expression would bolster this important observation.

Response: We really appreciate this reviewer for recognizing our contribution, and have revised the relevant contents on the analysis of PUM binding and inhibition of p21 protein expression as detailed above.

23. The authors attempted to classify the mRNAs based on microRNA sites, predicted RNA structure, and PUM binding. As presented on page 16, lines 319-334, the rationale and classification are difficult to understand. The approaches used to assign these categories are not explained at all. How was RNA structure predicted and what types of parameters were employed. Why is analysis relevant? In silico RNA structure prediction of mRNAs is notoriously difficult, heterogenous, and inaccurate. How the microRNA binding category was assigned is not described. How PUM1 and PUM2 binding was assigned is not described (the authors only did PAR-CLIP on PUM1). At the end of this confusing section, they state that significant changes are observed in mRNA, but then the corresponding Fig S10 does not include any type of statistical analysis to justify that statement. This section is confusing, inadequately documented, and doesn't provide meaningful insight.

Response: Thanks for the comment. We analyzed the results of this part according to a previously published paper [1]. We defined the interacting miRNAs as those that have a significant q-value within 200 nt (upstream or downstream) of the PUM1- or PUM2-binding site. We then assessed the effect of co-occurrence of these miRNA binding sites with PUM1/2 binding sites on the mRNA stability when PUM1 and PUM2 are depleted. This allowed us to classify the transcripts into five groups: (i) no binding to either PUM proteins or miRNA (None); (ii) Binding only to PUM1 and/or PUM2 (PUM-only); (iii) Binding only to miRNA (miRNA-only); (iv) Binding to both a PUM protein(s) and a miRNA(s) but does not form stem-loop that span the PUM and miRNA binding sites (Both (no stem-loop)); (v) Binding to both a PUM protein(s) and a miRNA(s) and potentially form stem-loop that span the PUM and miRNA binding sites (Both (stem-loop)). More detailed analysis can be found in this reference [1]. The goal is to assess how many PUM1/2 target mRNAs are also subject to the regulation of miRNAs, either antagonistic to or independent of PUM1/2 binding.

[1] HafezQorani S et al. Modeling the combined effect of RNA-binding proteins and microRNAs in post-transcriptional regulation. Nucleic Acids Res. 2016, 44(9):e83.

24. The authors then analyze AGO2, one of several proteins that can form miRNA induced silencing complexes. Again, the rationale behind this is not entirely clear. While one study reported that PUM-AGO interaction could affect translation in vitro (PMID: 22231398), another found that AGOs are not required for PUM activity in cells (PMID: 24942623). The authors create a hypothetical model of PUM1 and AGO2 (Fig. 6A) that is speculative and is not-tested experimentally. Indeed, their own data shows that PUM1 and AGO2 associate in a manner that is sensitive to RNase treatment, indicating that they co-occupy RNAs, instead of their presumed protein-protein interaction. The authors also test mutations of PUM1 and PUM2, and though

they conclude that these mutations “weakened” the PUM1-AGO2 interaction, the co-IP data does not support that conclusion. A single lane of a western blot of a co-IP experiment does not provide a quantitative means of assessing binding affinity between two proteins. From Fig. 6B and 6C, it is apparent that AGO2 co-IPs with both WT and mt PUM1 and PUM2. In the end, this analysis of PUM and AGO doesn’t provide new information that helps understand how PUM1 inhibits mRNAs.

Response: We agree that the interaction of PUM1 and AGO2 is RNA-dependent, at least for their endogenous interaction, which makes it difficult to validate any of these two models. Thus, we removed Fig. 6A. For Fig. 6B and C, though the blotting band of mutant PUM 1 and 2 is weaker than the one of WT PUMs, it is clear that the mutant PUM proteins retain the ability to bind with AGOs to some extent. In line with it, the domain mapping and RNase A treatment experiments clearly showed that the binding between PUMs and AGOs is RNA-dependent instead of direct protein-protein interaction. Given the concern that the western blot is not good enough for the quantification, we also removed Fig. 6B and C.

Prior to our analysis, at least three models were proposed for the mechanism of how PUM regulates mRNAs: 1. PUM may directly recruit the CCR4-NOT deadenylase complex to shorten polyA tail and promote mRNA decay; 2. PUM directly recruits AGO-eEF1A complex to repress RNA translation; 3. PUM binds to mRNA and exposes its miRNA-binding site, which facilitates miRNA-mediated mRNA regulation. Our analysis eliminated the second possibility in colorectal cancer cell line HCT116. Furthermore, we showed that PUM’s regulation on p21 might be mediated by the first model (CCR4-NOT complex) in Figure 6, and its regulation on p27 is mediated by the miRNA-mediated model, which is consistent with a previous study ^[1]. Taken together, we revealed that the PUM may regulate mRNAs in different mechanisms on different targets. In light of the reviewer’s comment, we’ve extensively revised the text to make this clearer (Pages 15-18).

[1] Kedde M et al. A Pumilio-induced RNA structure switch in p27-3’ UTR controls miR-221 and miR-222 accessibility. *Nat Cell Biol.* 2010, 12(10):1014-20.

25. Next, the authors focus on the microRNA miR130a and its effect on p21 expression. Another study (PMID: 25681685), not cited here, had previously reported that miR130a inhibits p21. In Fig. S11, the authors show that miR130a mimic increased proliferation of colon cancer cell lines, whereas an miR130a inhibitor decreased proliferation. They then analyze miR130a effect on p21 protein, mRNA, and in luciferase assays in Fig S12 and Fig 6. In these experiments, addition of miR130 reduced p21 mRNA and reporter activity. Oddly, the anti-miR130a was not effective - perhaps miR130a is not present/abundant in these cells? The ability to interpret the effects of PUM1 and PUM2 and miR-130a on p21 expression are hindered by the lack of quantitation and assessment of reproducibility of western blots shown in Figures 6H, 6I. From those blots, loss of PUM1 does qualitatively support increased p21, but the effect of miR-130a or the anti-miR are hard to discern. That observation calls into question whether p21 is really the means by which miR130a affects cell proliferation. The authors also investigate CDKN1B/p27 mRNA, which was already documented as inhibited by PUMs and microRNAs (PMID: 20818387; PMID: 29165587; PMID: 30811992). The effects of PUM1 and PUM2 on p27 expression are hard to discern in Figure 6K and 6L and are subject to the critiques raised above. The relevance of this analysis to the current study is not clear, and the results reaffirm what was already published. Overall, the section on PUMs and microRNAs does not provide new, convincing, and relevant insights.

Response: Thank you for the careful comments. To address the reviewer’s comments, we conducted RT-PCR and detected the expression of miR-130a in HCT116 and RKO (Response Figure 15). p21 is not up-regulated after anti-miR-130a because PUM1 still exists in the cells, and PUM1 can inhibit the expression of p21. In PUM1 KO cells, anti-miR-130a could upregulate the expression of p21 (revised Fig. 6I). We

repeated these results at least three times, and all of the results were consistent, with a representative result shown in the figure.

Response Figure 15. miR-130a expression in HCT116 and RKO cells by RT-PCR

Although regulation of p27 by PUMILIO has been reported previously, these reports indicated that, out of two miR-221/222 binding sites in the 3'UTR of p27 mRNA, on the second site that could form a loop with PRE played a role of miR-221/222 in inhibiting p27. Our findings reveal that the first miR-221/222 binding site also plays a role in the inhibition of p27 (Revised Fig. 6M and 6O). These findings indicate that PUM1 regulates p21 and p27 in different ways: it regulates p21 in a miRNA-independent manner but regulates p27 in part via coordinated regulation by miR-221/222.

26. Finally, one of the most interesting experiments in his manuscript is the assessment of feasibility targeting PUMs as a potential cancer therapy using nanoparticle-encapsulated siRNAs. Using a xenograft model, the authors report decreased colorectal tumor size and reduced metastasis caused by PUM1, PUM2 siRNAs. There are several reservations about the results. First, the authors do not confirm knockdown of PUM1 or PUM2 by the siRNAs in the xenografts (by western blot and/or by immunostaining). Single siRNAs are used for PUM1 and PUM2, raising the possibility that off target effects contribute to the observations. In their interpretation of these results, the authors wish to ascribe the effects on control of p21. However, they did not assess the effect of PUM1/2 siRNAs on p21 in the xenograft model.

Response: Thanks for the comments. We conducted immunohistochemistry on tumor treatment models and detected the knockdown of siPum1 and siPum2 using and found that the expression of PUM1 and PUM2 was decreased whereas p21 expression was upregulated (Response Figure 16). We have now added these data as new Fig. S15.

Response Figure 16. (Revised Fig. S14 in the manuscript) Representative immunohistochemistry staining of PUM1, PUM2, and p21 in siNC@MSN and siPum1/2@MSN group.

27. Also, both PUM1 and PUM2 siRNAs are effective in reducing growth of the xenograft, but the authors' earlier data showed that only PUM1 inhibits p21.

Response: Thank you for the comment. As PUM1 and PUM2 have many targets, our data suggest that PUM2 affects tumor cell growth probably by regulating other genes, not p21.

28. The authors wish to conclude that PUM1 and PUM2 are oncogenic, but their data does not prove this assertion. By definition, oncogenes have the capacity to transform normal cells to a cancerous state, an effect that is not-tested for PUM1 and PUM2 in this manuscript. Instead, they provide corroborating data that PUMs promote proliferation of previously transformed cancer cell lines or in an induced model of colon cancer.

Response: Good comment. We have revised our statement from “promote” to “important for”.

Additional critiques:

29. The figures are extremely small, making it difficult or even impossible to read. The figure legends are very superficial, making it difficult to interpret what is represented.

Response: We have revised figures and figure legends as suggested.

30. The authors make unsubstantiated claims based on data that are not statistically significant. In addition to the examples cited above, additional instances include: The authors conclusion that “all these data strongly correlate PUM1 and PUM2 with progression of human CRC” is not supported by most of the data they represent in Figures 1 and Figure S1.

Response: We have eliminated all these claims. For example, we have revised our statement from “all these data strongly correlate PUM1 and PUM2 with the progression of human CRC” to “all these data indicated that PUM1 and PUM2 were involved in the progression of human CRC (Please refer to Page 4)”.

31. Figure S1A and S1B: In the results, the authors state that CRC patient samples exhibit the highest PUM1 and PUM2 protein expression. However, these figures do not actually show protein levels. Instead, they plot % patients relative to cancer samples.

Response: The PUM1 and PUM2 protein expression data were shown in Figure S1A and S1B and are from the public database Human Protein Atlas. Protein expression of PUM1 and PUM2 in 20 different cancers is summarized by a selection of four standard cancer tissue samples representative of the overall staining pattern. “Patients %” in Fig S1A and B indicates the percentage of patients with high and medium protein expression levels. We have now added this explanation in the figure legend.

32. Figure S1C and S1D: In the results, the authors claim that “PUM1 and PUM2 were higher in Hong colorectal and Skrzypczak colorectal compared to normal colon cancer” However, only the PUM1 correlation is statistically significant, PUM2 is not.

Response: We’ve now changed this sentence to “PUM1 but not PUM2 was higher in Hong colorectal and Skrzypczak colorectal compared to normal colon cancer (Please refer to Page 4)”.

33. Figure S1G, S1H: In the results, the authors claim that “the expression of PUM1 and PUM2 negatively correlated with the survival rate of CRC patients” The Kaplan-Meier graphs do not support this conclusion. First, the authors bin the data into high and low PUM1 and PUM2 but do not define these parameters. Second, the data included in the graphs show that these relationships are not statistically significant.

Response: Thanks for the comments. We used 50:50 to define the TPM categorization of high and low PUM1 and PUM2 and have now added this information in the figure legend. In addition, we have changed “the expression of PUM1 and PUM2 negatively correlated with the survival rate of CRC patients” to “there may be a negative correlation between PUM1 expression and the survival rate of CRC patients” (Please refer to Page 4).

34. Figure 1A and 1C: PUM1 and PUM2 mRNA levels are compared between matched normal and CRC patient samples. Though 22 comparisons are plotted, only one p value is reported and its not clear how that was determined.

Response: P value is calculated by paired t-test, we have added this to the Figure legend (Please refer to Page 6).

35. In the results, page 14, line 278, the authors state that p21 “expression is negatively correlated with survival rate in CRC” however the data presented in Figure S9A disease free survival indicate that the association is not statistically significant. The TPM categorization of high and low expression is not defined.

Response: Thanks for the comments. We have changed “p21 expression is negatively correlated with survival rate in CRC” to “Considering that p21 plays an important role in G1/S phase transition and its expression is negatively correlated with overall survival rate of CRC but not disease free survival (Please refer to Page 15)”. We used 50:50 to define the TPM categorization of high and low p21. We have now added this information in the figure legend.

36. The methods, analysis, and results are inadequately documented, as described above.

Response: Good comment. We have improved the documentation of methods, analysis, and results according to suggestions in our revised version.

37. Inadequate or missing data and statistical analysis (examples are noted above). For all experiments, the data and statistical analysis methods and results need to be reported, including number and type of replicates, error type and values, the type of statistical tests.

Response: Thank you for this suggestion. We have added the all descriptions mentioned above in our revised version.

38. Throughout the manuscript, citations of directly relevant literature are omitted. An example is the statement: “Recent studies have shown that post-transcriptional regulation mediated by RNA binding proteins plays an important role in the initiation and development of CRC.” yet they provide no citations for this statement.

Response: We thank the reviewer to point out this mistake. We have added the citations in our revised version (Reference 4-6 in the manuscript).

[1] Pereira B et al. RNA-Binding Proteins in Cancer: Old Players and New Actors. *Trends Cancer*. 2017, 3(7):506-528.

[2] Chatterji P et al. RNA Binding Proteins in Intestinal Epithelial Biology and Colorectal Cancer. *Trends Mol Med*. 2018, 24(5):490-506.

[3] Gor R, Sampath SS, Lazer LM, Ramalingam S. RNA binding protein PUM1 promotes colon cancer cell proliferation and migration. *International journal of biological macromolecules* **174**, 549-561 (2021).

39. Overall the manuscript is poorly written. The manuscript and figures contain many grammatical and typographical errors.

Response: Following the reviewer’s suggestion, we have now carefully gone through the manuscript and supplementary materials and have double-checked the revised manuscript to exclude grammatical errors.

40. The introduction is incomplete and confusing due to lack of logical progression and contradictory statements. For example, statements in the first paragraph contradict each other. The authors state the “Recent studies have shown that post-transcriptional regulation mediated by RNA binding proteins plays an important role in the initiation and development of CRC.” and then one sentence later state the opposite “However, the regulation at the regulation at the post-transcriptional levels mediated by RNA-binding proteins, emerging important, remains much less investigated.” The second paragraph of the introduction is an example of the confusing text - the authors attempt to relate some basic background information on PUM1 and PUM2 while arguing that they may be either oncogenic or have roles in preventing cancer. The text is not logical, and I suspect that this is because the presentation of the previous data is incomplete and overly simplified. Another example is the third paragraph of the introduction, which starts with a confusing run-on-sentence.

41. **Response:** We really appreciate the reviewer’s comments and apologize for a quite disorganized Introduction containing logical errors and grammatical mistakes. This was a result of a hasty compression of a longer but much better written Introduction without a careful review (an omission in assigning different author’s responsibilities). We are quite embarrassed by this and have now revised Introduction to incorporate the reviews comments and to fit other errors. For example, we have changed the relevant text in the first paragraph to more precisely describe the current status as “Recent studies have shown that RNA binding proteins appear to play an important role in the initiation and development of CRC (Pereira, Billaud, and Almeida 2017) (Chatterji and Rustgi 2018; Gor et al. 2021). However, how these proteins regulate their RNA targets to achieve their function in CRC remains largely unknown.” All changed are in red text. Thank you!

II. Reviewer 2's comments & Response

Gong et al. study is aimed to demonstrate the role of Pum1 and Pum2 in CRC, by using in vitro and in vivo models (CRC cell lines, transgenic mice and xenograft tumours). By combining clinical information, in vitro and in vivo data, supported by transcriptomic and proteomic analyses, the authors demonstrated how Pum1 and Pum2 are crucial oncogenic proteins for the development of CRC. Overall, the study is well organized, supported by strong and valid data and it adds novel insights into the molecular regulation of CRC.

Response: We appreciate the encouraging and positive comments of the reviewer.

1. Authors should have characterized, both in vitro and in vivo, the cells that survived after Pum1 and Pum2 targeting. Indeed, the block of Pum1/2 reduces the proliferative potential of colorectal cancer cells, by enriching cells in the G0-G1 cell cycle phase. Given the importance of cancer stem cells (CSCs), enriched in slow cycling/quiescent cells, it is crucial to determine how the targeting of Pum1/2 could induce, or select, a CSC phenotype. For this reason, treated cells should be also studied for the expression of CSC and/or EMT putative markers, and functionally characterized in terms of colony-forming and invasive potential.

Response: We thank the reviewer for this incisive comment. To address this comment, we conducted qRT-PCR analysis and detected the expression of putative CSC markers, such as CD133 [1-4], CD166 [3,5], CD26 [6], CD44 [3,5,7], EpCAM [5], GLI-1 [8], Msi1 [9,10], ALDH1A1 [11,12] and Lgr5 [3,13,14,15,16] in WT HCT116, *Pum1*^{-/-}, *Pum1*^{-/-}, *Pum2*^{-/-} and *Pum2*^{-/-} cells (Response Figure 17). Among these CSCs markers, ALDH1A1 and Lgr5 expression are too low to detect. Some of the putative CSCs markers in including CD133, CD166, CD44, GLI-1 and Msi1 showed a various degree of reduction in the Pum1/2-deficient cells, even though the difference between the two biological replicates were sometime quite large. For example, the expression of CD26 was significantly downregulated in both Pum1 and Pum2 knock-out cells, while EpCAM expression was decreased in Pum2 but not Pum1 knock out cells. One marker, CD133 is most drastically downregulated in *Pum1*^{-/-}, *Pum1*^{-/-} and *Pum2*^{-/-}, even though it was significantly upregulated in *Pum2*^{-/-}. Thus, Pum1/2 knockout may have an effect on the CSC population, at least with regard to CD133 expression.

[1] O'Brien et al. A human colon cancer cell capable of initiating tumor growth in immunodeficient mice. Nature. 2007, 445(7123):106-110.

[2] Ricci-Vitiani L et al. Identification and expansion of human colon-cancer-initiating cells. Nature. 2007, 445(7123):111-115.

[3] Vermeulen L et al. Single-cell cloning of colon cancer stem cells reveals a multi-lineage differentiation capacity. Proc Natl Acad Sci U S A. 2008, 105(36):13427-13432.

[4] Zhu L, Gibson P et al. Prominin 1 marks intestinal stem cells that are susceptible to neoplastic transformation. Nature. 2009, 457(7229):603-607.

[5] Dalerba P et al. Phenotypic characterization of human colorectal cancer stem cells. Proc Natl Acad Sci U S A. 2007, 104(24):10158-10163.

[6] Pang R et al. A subpopulation of CD26+ cancer stem cells with metastatic capacity in human colorectal cancer. Cell Stem Cell. 2010, 6(6):603-615.

[7] Yeung TM et al. Cancer stem cells from colorectal cancer-derived cell lines. Proc Natl Acad Sci U S A. 2010, 107(8):3722-3727.

[8] Varnat F et al. Human colon cancer epithelial cells harbour active HEDGEHOG-GLI signalling that is essential for tumour growth, recurrence, metastasis and stem cell survival and expansion. EMBO Mol Med. 2009, 1(6-7):338-351.

[9] Weichert W et al. ALCAM/CD166 is overexpressed in colorectal carcinoma and correlates with shortened patient survival. J Clin Pathol. 2004, 57(11):1160-1164.

[10] Todaro M et al. IL-4-mediated drug resistance in colon cancer stem cells. Cell Cycle. 2008, 7(3):309-313.

[11] Huang EH et al. Aldehyde dehydrogenase 1 is a marker for normal and malignant human colonic stem cells (SC) and tracks SC overpopulation during colon tumorigenesis. Cancer Res. 2009, 69(8):3382-3389.

[12] Carpentino JE et al. Aldehyde dehydrogenase-expressing colon stem cells contribute to tumorigenesis in the transition from colitis to cancer. Cancer Res. 2009, 69(20):8208-8215.

[13] Barker N et al. Identification of stem cells in small intestine and colon by marker gene Lgr5. Nature. 2007, 449(7165):1003-1007.

[14] Sato T et al. Single Lgr5 stem cells build crypt-villus structures in vitro without a mesenchymal niche. Nature. 2009, 459(7244):262-265.

[15] Barker N et al. Crypt stem cells as the cells-of-origin of intestinal cancer. Nature. 2009, 457(7229):608-611.

[16] Schepers AG et al. Lineage tracing reveals Lgr5+ stem cell activity in mouse intestinal adenomas. Science. 2012, 337(6095):730-735.

Response Figure 17. (Revised Fig. S6E in the manuscript) qRT-PCR of the expression of putative cancer stem cell markers mRNAs in WT (black), *Pum1*^{-/-} (red) and *Pum2*^{-/-} (blue) HCT116 cells. Error bars represent SEM. ns: not significant, ***P* < 0.01, ****P* < 0.001, Student's t-test.

To further characterize the effect of targeting *Pum1/2* on CSCs, we conducted flow cytometry using CD44 and CD133 as markers, since they are commonly used in colorectal CSC studies. The number of CD133⁺ cells was remarkably reduced in *Pum1*^{-/-} and *Pum2*^{-/-} cells, except for one sample (*Pum2*^{-/-}-2) but the number of CD44⁺ cells are not reduced (Response Figure 18A & B). Our results indicate that knocking out *Pum1* *Pum2* could affect the characters of CD44⁺CD133⁺ cells in colorectal cancer, transforming many of them to CD44⁺CD133⁻. We have now added this information into the revised manuscript. (Page 8-9 and Fig. S6F and H)

Response Figure 18. (Revised Fig. S6F and H in the manuscript) (A) CD133⁺CD44⁺ subpopulations were drastically reduced in *Pum1* or *Pum2*-depleted HCT116 cells by FACS analyses; **(B)** Bar graph showing the percentage of CD44 and/or CD133 positive cells in HCT116 wt, *Pum1*^{-/-1}, *Pum1*^{-/-2}, *Pum2*^{-/-1} and *Pum2*^{-/-2} cells (n=3). Error bars represent SD. ns: not significant, ***P* < 0.01, ****P* < 0.001, Student's t-test.

To further characterize the effect of PUM1 and PUM2 on CRC stem cells, we used flow cytometry and qRT-PCR to detect *Lgr5*⁺, an intestinal stem cell marker. As indicated by flow cytometry, the *Lgr5*⁺ subpopulation is extremely small (Response Figure 19A). Unexpectedly, the percentage of *Lgr5*⁺ cells was increased in both *Pum1* and *Pum2* knockout cells, as indicated by FACS (Response Figure 19A and B). Hence, even though knocking out *Pum1/2* reduced the number of CD133⁺ cells, it increased the number of *Lgr5*⁺ cells, and had no effect on CD44⁺ cells. These changes reflect that PUM1 and PUM2 do have a function in regulating stem cell fate, with their deficiency skew the CD133⁺*Lgr5*⁺ and CD133⁻*Lgr5*⁻ cells, likely by converting these two populations of cells to CD133⁻*Lgr5*⁺ cells. We have now added this information into the revised manuscript. (Pages 8-9 and Fig. S6G and I)

Response Figure 19. (Revised Fig. S6G and I in the manuscript) (A) Lgr5⁺ subpopulations were detected in Pum1 or Pum2-depleted HCT116 cells by FACS analyses. **(B)** Bar graph showing the percentage of Lgr5 positive cells in HCT116 WT, *Pum1*^{-/-1}, *Pum1*^{-/-2}, *Pum2*^{-/-1} and *Pum2*^{-/-2} cells (n=3). Error bars represent SD. ns: not significant, **P* < 0.05, ***P* < 0.01, ****P* < 0.001, Student's t-test.

2. Accordingly, given the possibility for CSCs to exit quiescence and give rise to recurrence, the authors should investigate the possibility to transiently target Pum1/2 and give the chance to spared cells to regrowth, to assess their capacity to re-initiate tumour.

Response: We appreciate this comment and are very tempted to do these experiments. Unfortunately, after obtaining our new results on CSC markers as indicated above, we realized that the impact of Pum1 and Pum2 on CSC fate is complicated and multifaceted. It will take a long-term effort to study this impact by investigating how Pum1 and Pum2 differentially regulate the expression of CD44, CD133 and Lgr5, and how these changes in gene expression reflect the changes in different aspects of the CSC fate and properties, which is beyond the scope of our paper. In light of the reviewer's comments, we added new results on CSC characterization to the manuscript (Pages 8-9) and discussed the implication of these new findings to the role of PUM1 and PUM2 in regulating CSC fate.

3. The authors should include in the manuscript the characterization of subcutaneous xenografts generated by CRC cells following the treatment with nanoparticles-encapsulated PUM siRNA. In particular, they should perform histological and immunophenotypical analysis, in terms of proliferative potential, apoptosis, CSC and EMT markers, of mouse avator xenografts generated by survived cells.

Response: We agree with this reviewer's comments. H&E staining and immunohistochemistry staining of Ki-67 in siNC@MSN and siPum1/2@MSN group were performed, as shown in Response Figures 20 & 21 and revised Figs. S14-15 in the manuscript. Representative H&E staining showing a significant decrease of adenomas when knocking down Pum1 and Pum2 (Response Figure 20 and revised manuscript Figure S14), indicating PUM1 and PUM2 are required for tumor progression. Ki-67 staining revealed that cell proliferation in siPum1/2@MSN group was inhibited as compared with siNC@MSN group (Figure 21 and revised manuscript Figure S14).

Response Figure 20. (Revised Fig. S16 in the manuscript) Representative H&E staining in siNC@MSN and siPum1/2@MSN group.

Response Figure 21. (Revised Fig. S14 in the manuscript) Representative immunocytochemistry staining of Ki-67 in siNC@MSN and siPum1/2@MSN group.

To understand the effects of Pum proteins on apoptosis, we performed immunocytochemistry staining on apoptotic marker TUNEL in both siNC@MSN and siPum1/2@MSN groups (Response Figure 22). Neither control nor experimental samples showed significant signals of TUNEL, indicates there was no significant apoptosis in both groups. To further illustrate Pum effects on CSC and EMT makers, we utilized CD44 as a CSC marker and N-cadherin and E-cadherin as EMT markers (Response Figure 22). In line with FACS analysis on HCT116 cells, there was no significant change between the two groups in the expression of CD44 or the two EMT makers. Together, these results indicate that Pum proteins have no significant effect on tumor apoptosis, EMT, or the CD44⁺ populations.

Response Figure 22. Representative immunochemistry staining of TUNEL, CD44, E-cadherin, and N-cadherin in siNC and siPum1&2 group.

5. The synergistic score of Pum1 and Pum2 targeting should be also validated in vitro.

Response: We thank the reviewer for this suggestion. To address this suggestion and a similar comment by Reviewer 1 (comment 6, page 5), we tried twice to generate Pum1/2 double knockout HCT116 cells, selecting from more than 400 single cell clones (for details, see our response to Reviewer 1's comment 6 in pages 5-6). Unfortunately, we did not get Pum1/2 double knockout HCT116 cells at last. These results suggest that PUM1 and PUM2 double knockout cause severe cell lethality thus it is not feasible to analyze double-knockout cell lines.

III. Reviewer 3's comments & Response

In the manuscript "PUMILIO proteins in Colorectal Cancer: Tumor Growth Promoting Function and Potential as Therapeutic Targets" Gong, Liu, Yuan, Yang, and colleagues present a comprehensive analysis of Pumilio protein posttranscriptional gene regulatory role in colorectal cancer cell lines, as well as in xenograft tumor models and Pum1/Pum2 knockout mice. The authors identify increased expression of PUM1 and/or PUM2 in colorectal cancers and find that in model cell lines their knockdown results in reduced cell cycle progression and tumorigenicity. They use RNA-seq and PAR-CLIP to identify regulated target RNAs on a transcriptome-wide scale and zero in and validate cell cycle related genes as top Pum targets responsible for the phenotype. Finally, they show that in vivo knockdown of Pum1 and/or Pum2 using nanoparticle-encapsulated siRNAs results reduced colorectal cancer in mouse models. This study is the most comprehensive analysis of Pumilio protein targets and function in mammals, spanning transcriptome-wide experiments in cell culture, coupled with tissue-specific mouse knockouts and murine tumor models. The combination of well-executed experiments allows a compelling picture of Pumilio proteins as potential oncogenes to emerge. There are a few points the authors may consider addressing before publication:

Response: We really appreciate the encouraging and positive comments of the reviewer.

1. Figure 2: Considering that PUM1 and PUM2 are close paralogs and recognize the same sequence element it stands to reason to expect that they would compensate for each other and arguably, the subtle differences in the effect of Pum1 or Pum2 knockdown could be due to partial compensation by different expression levels of the paralog. Thus, the posttranscriptional effect of Pumilio proteins could be exacerbated and more clearly revealed by Pum1/2 double knockout (DKO). Perhaps the authors could consider generating such DKO cell lines for transcriptomic and proteomic analyses.

Response: Thanks for this suggestion, which was also raised by Reviewers 1 (comment 6, page 5) and 2 (comment 5, page 28). To address this, we tried twice to generate Pum1/2 double knockout HCT116 cells, selecting from more than 400 single cell clones (for details, see our response to Reviewer 1's comment 6 in pages 5-6). Unfortunately, we did not get Pum1/2 double knockout HCT116 cells at last. These results suggest that PUM1 and PUM2 double knockout cause severe cell lethality thus it is not feasible to analyze double-knockout cell lines.

2. Figure 3: Are the differences in proteome/transcriptome changes between Pum1/Pum2 KO qualitative or just due to differences in P-values from RNAseq analysis? Perhaps the authors could present heatmaps or scatterplots of gene expression values from Pum1 and Pum2 KO cells to allow insights into whether the proteins indeed regulate the same set of transcripts. This could also allow the authors to check whether the subtle differences in cell cycle gene expression in Figure 5E are indeed representative of a gene regulatory difference of the two Pum paralogs.

3. Figure 3A/B: It would be nice to see a scatterplot of log fold changes from proteomic and transcriptomic experiments after Pum1 or Pum2 KO to show whether protein changes are mainly due to transcript changes, which would confirm that Pum proteins most likely function by changing transcript levels (as the literature suggests).

Response to comments 2 and 3: We thank the reviewer for the suggestion. A similar comment was raised by reviewer 1 (comment 13, page 9). To address this comment, we plotted the mRNA and protein \log_2 FoldChange (Response Figure 10) in Pum1 knockout and Pum2 knockout cells, respectively. The correlation of gene expression changes on the mRNA level and RNA level for PUM1 knockout is 0.7 (Figure 10A) and for PUM2 knockout is 0.76 (Figure 10B). This suggests that protein level changes, to a large extent, result from transcriptional changes. (For details, please refer to our response to reviewer #1's comment #13

on pages 10-11 and Response Figs. 8-10)

4. Figure 4: The PAR-CLIP experiments and their integration with RNAseq could be presented more clearly. E.g. are the sites in Fig 4A target sites found by PARalyzer? How well-correlated are the two PAR-CLIP replicates (e.g. scatterplot of number of crosslinked reads per gene with corresponding correlation factor)? How many binding sites contain the MEME-derived motif? The authors could also take advantage of the semi-quantitative nature of PAR-CLIP and integrate with RNAseq in a manner that would reveal the regulatory effect of Pum; i.e. by binning PAR-CLIP targets based on number of crosslinked reads per gene and then showing the histogram of log-fold transcript changes after Pum KO (both Pum1 and Pum2) on the different bins (see e.g. Yamaji et al., Nature, 2017 for illustration).

Response: We appreciate these excellent questions, which was similar to reviewer 1's comment 13. To integrate PAR-CLIP data and mRNA-Seq data, we examined the relationship between the number of crosslinked reads per gene and the fold change of gene expression in *Pum1* knockout cells. From the analysis of PAR-CLIP data, 302 genes are PUM1's direct targets. Among them, only 28 genes have statistically significant changes in expression (FDR<0.1) at the mRNA level in *Pum1* knockout cells. We binned the 28 targets into two groups, one group with weaker PUM1 binding (10 genes with ConversionCount<=8) and the other with stronger binding (18 genes with ConversionCount>8). Within each group, we plotted their fold change according to their altered expression pattern (Upregulated or Downregulated) in *Pum1* knockout cells. The upregulated genes did not show statistically significant correlation between fold change and PUM1 binding strength by Wilcoxin rank sum test (see page 12, Response Figure 11). This is likely due to an extremely small number of genes that went into this analysis. However, among the downregulated genes, a greater fold change is correlated with stronger binding (Wilcoxin rank sum test P-value=0.022) (Response Figure 11).

5. Figure 6: Here the authors characterize the interaction of AGO proteins with PUM proteins. It is clear that they were inspired for this analysis by previous literature (e.g. Kedde, Nat Genet); that being said, the data don't conclusively show that these proteins interact, in fact, the data does show that the interaction is RNA bridged, which can be expected between almost any two 3'UTR binding RNA binding proteins. Why did the authors not consider examining the interaction between Pumilio and the CCR4-NOT deadenylase complex as the effector complex? Such an interaction is well-established in invertebrates and is clearly conserved in humans (e.g. Enwerem et al., RNA, 2021). The authors could check whether siRNA knockdown of CNOT1 and CNOT7 abrogates Pumilio-mediated target repression.

Response: We fully agree with this comment. Our data support that PUM and AGO association is RNA-bridged. Indeed, the CCR4-NOT deadenylate complex is a well identified partner of PUM proteins in diverse species to induce deadenylation-mediated mRNA turnover and translational repression [1-5]. Following this reviewer's suggestion, we examined whether siRNA knockdown of CNOT1 and CNOT7 abrogates PUM-mediated target repression. We employed p21 as a reporter gene since PUM-mediated p21 repression has been well defined in our manuscript. siRNAs of CNOT1 and CNOT7 were transfected into HCT116 cells. The regulatory effect of CNOT1 and CNOT7 to p21 was evaluated by real-time PCR and western blot. Knocking down of CNOT1 and CNOT7 upregulated p21 RNA (Response Figure 23A) and protein level (Response Figure 23B and C). Based on the known CNOT-PUM interaction, these results suggest that CNOT Complex components may interact with PUM to repress p21 expression in colorectal cancer cells.

Response Figure 23. (A) The mRNA levels of p21 when knocking out CNOT1 or CNOT7 in HCT116 cells. Error bars represent SEM. Significance of difference between each siCNOT knock-down and siNC control is indicated. ns: not significant, ** $P < 0.01$, *** $P < 0.001$, Student's t-test. **(B)** The protein levels of p21 when knocking out CNOT1 in HCT116 cells. **(C)** The protein levels of p21 when knocking out CNOT7 in HCT116 cells.

[1] Goldstrohm AC et al. PUF proteins bind Pop2p to regulate messenger RNAs. *Nat Struct Mol Biol.* 2006, 13(6):533-9.

[2] Goldstrohm AC et al. PUF protein-mediated deadenylation is catalyzed by Ccr4p. *J Biol Chem.* 2007, 282(1):109-14.

[3] Van Etten J et al. Human Pumilio proteins recruit multiple deadenylases to efficiently repress messenger RNAs. *J Biol Chem.* 2012, 287(43):36370-83.

[4] Arvola RM et al. Unique repression domains of Pumilio utilize deadenylation and decapping factors to accelerate destruction of target mRNAs. *Nucleic Acids Res.* 2020, 48(4):1843-1871.

[5] Enwerem III et al. Human Pumilio proteins directly bind the CCR4-NOT deadenylase complex to regulate the transcriptome. *RNA.* 2021, 27(4):445-464.

6. I would encourage the authors to thoroughly proofread the text (as well as figure panels) again to eliminate the large number of typos or inadvertent grammar mistakes (e.g. 166 should read as "lead to genome instability"; 194: either the and or the but needs to be removed).

Response: Thank you for your kindly reminder. We apologize for typos and grammatical mistakes and have corrected them throughout the text.

7. Reference missing that supports the idea that posttranscriptional gene regulation is recognized to play an important role in CRC.

Response: Thanks for the reviewer's suggestion. We have now added the following references ^[1-2] (Please refer to reference 4, 5 in the manuscript).

[1] Pereira B et al. RNA-Binding Proteins in Cancer: Old Players and New Actors. *Trends Cancer.* 2017, 3(7):506-528.

[2] Chatterji P et al. RNA Binding Proteins in Intestinal Epithelial Biology and Colorectal Cancer. *Trends Mol Med.* 2018, 24(5):490-506.

8. How does the Pumilio expression in CRC compare to expression in other human tissues and during development? Is it markedly higher than "normal" peak expression?

Response: To compare Pumilio expression in CRC with other human tissues, we utilized RNA_Seq data from public TCGA and GTEx databases, using GEPIA for data processing (Response Figure 24A). *Pum1* is expressed at higher levels in 10 of out 24 different types of cancers as compared with paired normal tissues,

including colon adenocarcinoma (COAD) and rectum adenocarcinoma (READ)--the two cancer types most relevant to the RKO and HCT116 colon adenocarcinoma cell lines that we used in our study. *Pum2* is also expressed at higher levels in 11 out of 24 different types of cancers, including READ, as compared to normal tissues. While in the remaining 13-14 types of cancer cells, *Pum1* and *Pum2* are expressed at similar levels as in their corresponding normal tissues.

To examine Pumilio expression during development, we utilized another set of RNA_set data from the public database Expression Atlas (Response Figure 24B). Although Pumilio expression level was not detected in all developmental stage, our heatmap indicates that both PUM1 and PUM2 are expressed at higher levels in early developmental stages.

Response Figure 24. (A) *Pum1* and *Pum2* expression in different cancers compared with normal tissues. Original data from TCGA and GTEx database, GEPIA were used for data processing (Fold Change Cutoff = 1.2, q-value Cutoff = 0.01). Red dots represent samples from tumors and green dots represent samples from normal tissues. T: Tumor, N: Normal, COAD: Colon adenocarcinoma, READ: Rectum adenocarcinoma, ESCA: Esophageal carcinoma, GBM: Glioblastoma multiforme, HNSC: Head and Neck squamous cell carcinoma, STAD: Stomach adenocarcinoma, TGCT: Testicular Germ Cell Tumors, UCEC: Uterine Corpus Endometrial Carcinoma, UCS: Uterine Carcinosarcoma, SKCM: Skin Cutaneous Melanoma, BLCA: Bladder Urothelial

Carcinoma, BRCA: Breast invasive carcinoma, CESC: Cervical squamous cell carcinoma and endocervical adenocarcinoma, KICH: Kidney Chromophobe, KIRC: Kidney renal clear cell carcinoma, KIRP: Kidney renal papillary cell carcinoma, LIHC: Liver hepatocellular carcinoma, LUAD: Lung adenocarcinoma, LUSC: Lung squamous cell carcinoma, OV: Ovarian serous cystadenocarcinoma, PCPG: Pheochromocytoma and Paraganglioma, PRAD: Prostate adenocarcinoma, SARC: Sarcoma, THCA: Thyroid carcinoma. **(B)** *Pum1* and *Pum2* time-series expression of the development in major organs. A heatmap generated from the RNA_seq data from Expression Atlas reflecting TPM values is shown for major organs. Green represents lower gene expression and red represents higher gene expression.

9. It would be good if the authors could add a few words explaining the ROC analysis and what it shows. Not all readers will be familiar with it.

Response: Good suggestion. We have now explained ROC analysis more clearly in our revised manuscript. (Please refer to Page 4).

10. It would be nice to explain how the tumor is induced in the AOM/DSS model.

Response: Good suggestion. We have now added more description on how the tumor is induced in the AOM/DSS model to our revised manuscript (Please refer to Page 4-5).

11. The section on *Pum1/Pum2* mouse KO in the AOM/DSS could benefit from some mild rewriting. The authors first describe the effect of *Pum* KO on cancer progression and only towards the end of the section present the controls showing that *Pum1/2* cKO does not impact intestinal development. Consider switching the order to prevent any confusion on the reader's part.

Response: This is a good suggestion. We have now reorganized our description in our revised manuscript (Please refer to Page 4-5).

12. The authors could comment on why the nanoparticles would be expected to enrich at the tumors. Unless I missed something they don't appear to be modified in a way that would guide them there.

Response: Good comment. After i.v. injection, nanoparticles with proper size (100-200 nm) can be accumulated in solid tumor via the enhanced permeability and retention (EPR) effect, which was the mechanism responsible for the nanoparticle enrich in solid tumor^[1]. Related content has been added in the revised manuscript (Please refer to Page 21).

[1] Shi J et al. Cancer nanomedicine: progress, challenges and opportunities. Nat Rev Cancer. 2017, 17(1):20-37.

We hope that the above responses have satisfactorily addressed the reviewers' concerns. This review process significantly strengthened our manuscript. We thank you and the reviewers again for your time and valuable comments, and look forward to a favorable reply from you!

Sincerely,

Sanhong Liu, Ph.D.

Haifan Lin, Ph.D.

Chao Fang, Ph.D.

REVIEWER COMMENTS

Reviewer #1 (Remarks to the Author):

Overall the manuscript is substantially improved, and I commend the authors for their efforts; however, multiple remaining issues need to be addressed, including a potentially major problem with the knockout cells.

The authors' responded to the majority of my original critiques and performed additional analyses that are necessary for assessing their datasets. However, in a number of cases they failed to incorporate those analyses into the manuscript.

Cases in point:

Response Fig. 3: Testing new clones of PUM1/2 KO HCT116 and RKO cells

Response Figs. 5, and 6: The PCA analysis of proteomics and transcriptomics datasets should be included in the manuscript.

Though the authors state in the response that the observations in Figure 6E, F, H, and I were reproducible, they have not provided quantitation and statistical analysis to support the validity of these important observations.

Those analyses should be included in the manuscript.

A major concern pertaining to the CRISPR/Cas9 KO of PUM1 and PUM2 arises from the proteomics analysis. It's perplexing that PUM1 protein is detected in the PUM1 KO proteomics dataset in Table S4. In fact, PUM1 levels were only decreased by half in the KO. Likewise, PUM2 protein is detected in the PUM2 KO in Table S5, and its levels only decreased by ~1/3. These observations raise serious concerns that the knockout cell lines are not truly knockouts, and that the cells still express PUM proteins that were not detected by the individual PUM1 and PUM2 antibodies used in Figure S5. The authors must address these issues, otherwise it invalidates multiple conclusions based on the use of the knockout cells.

P15, lines 302-303: The gene "p27" appears to be erroneously indicated in this sentence. Instead, "p21" should be indicated, consistent with the remainder of the paragraph.

P18, lines 374-375: The stated conclusion is not supported by the evidence. The authors do not test p21 mRNA at all. In fact, PUM and AGO2 are likely to co-occupy many mRNAs.

Figure S10 lacks the crucial statistical analysis that is essential for assessing the validity of the authors' conclusions

Reviewer #2 (Remarks to the Author):

The authors have addressed all the concerns of this reviewer. Indeed, the revised version of the manuscript shows improved scientific quality and clarity of the results, thus making the research very solid.

Reviewer #3 (Remarks to the Author):

The authors exhaustively engaged with the reviewer's comments. Below are a few remaining minor concerns that could be addressed.

Minor concerns:

- typos on p4, ll 80-81.
- in the paragraph starting on l229, the authors repeatedly equate changes in mRNA abundance with transcriptional changes. Strictly speaking transcriptional changes would be due to changes in transcriptional activities, however, PUM1/2 are posttranscriptional regulators that influence mRNA half-life. Therefore it would be better if the authors would refer to "mRNA stability" or - even better, in the absence of half-life measurements - "mRNA abundance".
- it would be good if the authors could spell out in Fig. 4 and the accompanying text how many binding sites for PUM1/2 they identify in the experiments (so that the reader does not need to refer the supplementary table). Based on the fact that only 15 and 45 of the > 1000 genes changed by PUM KO are also among targets leads to the question whether the CLIP experiment was exhausting. It may be necessary to discuss this.

- The miRNA-PUM part of the story remains the weakest. In the end, the authors demonstrate only that PUM regulation is stronger than miRNA regulation on the individual targets, but it remains absolutely unclear how this ties into their story that is focused on the potentially oncogenic role of PUM. They don't demonstrate convincingly that the two pathways for miRNA regulation systematically cooperate and distract from their message. I would strongly encourage the authors to drop Fig. 6 from the manuscript and develop an independent story from it.
- In the rebuttal the authors mention that generating DKO cell lines proved impossible, which is interesting in its own right and could be mentioned. Nevertheless, the authors could have tried to use siRNAs to deplete the remaining PUM in a PUM single knockout cell.

I. Reviewer 1's comments & Response

Overall the manuscript is substantially improved, and I commend the authors for their efforts; however, multiple remaining issues need to be addressed, including a potentially major problem with the knockout cells. The authors' responded to the majority of my original critiques and performed additional analyses that are necessary for assessing their datasets. However, in a number of cases they failed to incorporate those analyses into the manuscript.

Cases in point:

1. Response Fig. 3: Testing new clones of PUM1/2 KO HCT116 and RKO cells. Response Figs. 5, and 6: The PCA analysis of proteomics and transcriptomics datasets should be included in the manuscript.

Response: We thank the reviewer for the suggestion. We have added these results to our revised manuscript, see new Figs. S7 and S8.

2. Though the authors state in the response that the observations in Figure 6E, F, H, and I were reproducible, they have not provided quantitation and statistical analysis to support the validity of these important observations. Those analyses should be included in the manuscript.

Response: We thank the reviewer for this suggestion and have done quantitation and statistical analysis. The results fully support our conclusion. However, reviewer 3 suggests us to remove Fig. 6 and its relevant text. We agree with reviewer 3's assessment of the overall structure of this manuscript and have deleted Fig. 6, the related text, and supplementary figures (Figs. S10-12) from the manuscript to make the manuscript less cumbersome and logically clearer.

3. A major concern pertaining to the CRISPR/Cas9 KO of PUM1 and PUM2 arises from the proteomics analysis. Its perplexing that PUM1 protein is detected in the PUM1 KO

proteomics dataset in Table S4. In fact, PUM1 levels were only decreased by half in the KO. Likewise, PUM2 protein is detected in the PUM2 KO in Table S5, and its levels only decreased by ~1/3. These observations raise serious concerns that the knockout cell lines are not truly knockouts, and that the cells still express PUM proteins that were not detected by the individual PUM1 and PUM2 antibodies used in Figure S5. The authors must address these issues, otherwise it invalidates multiple conclusions based on the use of the knockout cells.

Response: We thank the reviewer for these careful observations. The PUM peptides were detected in *Pum1*^{-/-} and *Pum2*^{-/-} cells because of the nature of the tandem mass tag (TMT) system that we used for quantitative mass spectrum, in which samples were labeled separately with different isobaric variants of TMT and then combined together for HPLC-MS/MS analysis (Response Figure 1). This isobaric labeling system inherently has the interference of contaminating ionized peptides that are from different proteins but share the same or almost the same charge-to-mass ratio (i.e., retention time)¹⁻². Hence, TMT is not suitable for determining the presence vs absence of a protein but is perhaps the best available method for comparing differential expression of the proteome.

Response Figure 1. The experimental designs of the TMT experiments before this revision and the label-free mass spec experiments added for this revision. WT, *Pum1*^{-/-} and *Pum2*^{-/-} were grown in parallel. Proteins were extracted from cell cultures and digested into peptides, which were measured using LC-MS/MS. In TMT, peptides from the different samples were mixed after isobaric chemical labeling. In the label-free method, the samples were prepared and measured separately.

To further validate the specificity of knockout cells, we collected WT, *Pum1*^{-/-} and *Pum2*^{-/-} cells, label-free sample were prepared and measured by mass spectrometry separately (Response Figure 1)³. Different from TMT, the label-free quantification approach aims to correlate the mass spectrometric signal of intact peptides or the number of peptide sequencing events with the relative protein quantity directly⁴⁻⁵. All MS/MS spectra were searched using SEQUEST against the human genome database. Each sample was separately run three times. Totally, 5503, 5496 and 5484 unique proteins were identified in WT, *Pum1*^{-/-} and *Pum2*^{-/-} samples, respectively, with 74.2%, 72.1% and 73.8% overlaps among the triplicate runs of WT, *Pum1*^{-/-} and *Pum2*^{-/-} samples. This shows that label-free method provided deep coverage of the proteome (~80%). As summarized in

Response Table 1 and Table 2, 18 peptide-spectrum matches (PSMs) of nine PUM1-unique peptides and 8 PSMs of four PUM2-unique peptides were identified in WT cells. In contrast, no corresponding PUM1- or PUM2-unique peptide was detected in *Pum1*^{-/-} or *Pum2*^{-/-} cells, respectively. These results clearly confirmed the absence of PUM1 and PUM2 in *Pum1* and *Pum2* knock-out cells, respectively. We have now added these new results to the manuscript text (see “Methods” section, “Mass Spectrometry and data analysis” subsection and new Supplementary Table 9).

Response Table 1. PUM1 protein identification by label-free MS, with the numbers of PUM1-unique peptides tabulated.

peptides	Run	WT			Pum1 ^{-/-}		
		run1	run2	run3	run1	run2	run3
DSAWGTS D H S V S Q P I M V Q R		1	1	0	0	0	0
WPTGDNIHAEHQVR		1	1	1	0	0	0
SASSASSL F S P S S T L F S S S R		1	1	0	0	0	0
SQDDAMVDYFFQR		1	1	1	0	0	0
GIFLGDQWR		1	0	0	0	0	0
AVLIDEVCTMNDGPHSALYTMMK		1	1	1	0	0	0
MIDVAEPGQR		0	1	0	0	0	0
SMDELNHDFQALALEGR		0	1	1	0	0	0
FWETDESSKDGPK		0	0	1	0	0	0

Response Table 2. PUM2 protein identification by label-free MS, with the number of PUM2-unique peptides tabulated.

peptides	Run	WT			Pum2 ^{-/-}		
		run1	run2	run3	run1	run2	run3
ETAWGASHHMSQPIMVQR		1	1	1	0	0	0
DAETDGPEKGDQK		1	0	0	0	0	0
ALLIDEVCCQNDGPHSALYTMMK		1	0	1	0	0	0
ASPFEEQNR		0	1	1	0	0	0

References:

- [1] Ting L, Rad R, Gygi SP, Haas W. MS3 eliminates ratio distortion in isobaric multiplexed quantitative proteomics. *Nat Methods* **8**, 937-940 (2011).
- [2] Martinez-Val A, et al. On the Statistical Significance of Compressed Ratios in Isobaric Labeling: A Cross-Platform Comparison. *J Proteome Res* **15**, 3029-3038 (2016).
- [3] Wiśniewski JR, Zougman A, Nagaraj N, Mann M. Universal sample preparation method for proteome analysis. *Nature methods* **6**, 359-362 (2009).
- [4] Bantscheff M, Schirle M, Sweetman G, Rick J, Kuster B. Quantitative mass spectrometry in proteomics: a critical review. *Anal Bioanal Chem* **389**, 1017-1031 (2007).

[5] Bantscheff M, Lemeer S, Savitski MM, Kuster B. Quantitative mass spectrometry in proteomics: critical review update from 2007 to the present. *Anal Bioanal Chem* **404**, 939-965 (2012).

4. P15, lines 302-303: The gene “p27” appears to be erroneously indicated in this sentence. Instead, “p21” should be indicated, consistent with the remained of the paragraph.

Response: We thank the reviewer to point out this mistake. We have changed p27 to p21 in our revised manuscript.

5. P18, lines 374-375: The stated conclusion is not supported by the evidence. The authors do not test p21 mRNA at all. In fact, PUM and AGO2 are likely to co-occupy many mRNAs.

Response: We thank the reviewer for the valuable comment. As reviewer said, PUM and AGO2 can co-occupy many mRNAs, and p21 is only one of them. Our description is inaccurate in the text. Following Reviewer 3’s suggestion, this part of the content has been removed.

6. Figure S10 lacks the crucial statistical analysis that is essential for assessing the validity of the authors’ conclusions

Response: To address this comment, we did statistical analysis, and the results are consistent with those described in our manuscript. Following Reviewer 3’s suggestion, this part of the content has been removed.

II. Reviewer 3’s comments & Response

The authors exhaustively engaged with the reviewer's comments. Below are a few remaining minor concerns that could be addressed.

Minor concerns:

1. Typos on p4, ll 80-81.

Response: We thank the reviewer to point out this mistake and have corrected these typos.

2. In the paragraph starting on 229, the authors repeatedly equate changes in mRNA abundance with transcriptional changes. Strictly speaking transcriptional changes would be due to changes in transcriptional activities, however, PUM1/2 are posttranscriptional regulators that influence mRNA half-life. Therefore, it would be better if the authors would refer to "mRNA stability" or - even better, in the absence of half-life measurements - "mRNA abundance".

Response: We agree with this comment, and two related descriptive errors have been corrected (Please refer to Page 12, Line 245-246).

3. It would be good if the authors could spell out in Fig. 4 and the accompanying text how many binding sites for PUM1/2 they identify in the experiments (so that the reader does not need to refer the supplementary table).

Response: Thanks for the suggestion. We have now provided a more detailed description for reader's convenience (please refer to Page 13, Line 267-268).

4. Based on the fact that only 15 and 45 of the > 1000 genes changed by PUM KO are also among targets leads to the question whether the CLIP experiment was exhausting. It may be necessary to discuss this.

Response: We appreciate this suggestion. In order to obtain more credible PUM1 targets, we used relatively stringent criteria in our CLIP data analysis, including parameter adjustments for PARalyzer, binding sites intersection from two biological replicates, and presumptive background signal removal. This results in 160 PUM1-bound target mRNAs. When we reduced stringency, more direct target genes were recovered from the CLIP, including an increased number of targets that overlapped with genes showing differentially expressed mRNAs or proteins in the *Pum1*^{-/-} cells. We limited ourselves to the 160 targets to avoid false positive targets. It was surprising to us as well that only 15 and 48 of the >1000 genes whose expression was changed by *Pum1*-KO were bound by PUM1. This indicates that most of the differentially expressed genes are indirect targets of PUM1. Meanwhile, PUM1 may binds to some mRNAs without any significant regulatory function. In addition, it is possible that both PUM1 and PUM2 binding are needed to generate an easily detectable regulatory effort on some other of the 160 PUM1-target mRNAs, since PUM1 and PUM2 bind to overlapping set of targets¹. Finally, it is possible that PUM1 might binds to some mRNAs without any significant regulatory function. We have now added this discussion into our manuscript. (Please refer to Page 23, Lines 467-479).

References:

[1] Zhang M, Chen D, Xia J, Han W, Cui X, Neuenkirchen N, Hermes G, Sestan N, & Lin H. Post-transcriptional regulation of mouse neurogenesis by Pumilio proteins. *Genes & Development*, **31(13)**, 1354–1369 (2017).

5. The miRNA-PUM part of the story remains the weakest. In the end, the authors demonstrate only that PUM regulation is stronger than miRNA regulation on the individual targets, but it remains absolutely unclear how this tie into their story that is focused on the potentially oncogenic role of PUM. They don't demonstrate convincingly that the two pathways for miRNA regulation systematically cooperate and distract from their message. I would strongly encourage the authors to drop Fig. 6 from the manuscript and develop an independent story from it.

Response: We agree with the reviewer for this suggestion. Independently, we had contemplated with and produced versions with or without the PUM1-miRNA interaction part. In light of the reviewer's comment, we have removed this part of the manuscript (including

Fig. 6 and Figs. S10-12). The revised manuscript indeed reads better.

6. In the rebuttal the authors mention that generating DKO cell lines proved impossible, which is interesting in its own right and could be mentioned. Nevertheless, the authors could have tried to use siRNAs to deplete the remaining PUM in a PUM single knockout cell.

Response: We thank the reviewer for the valuable suggestion and have added the DKO cell lines contents in our revised manuscript (Page 8, Lines 164-167). In addition, we will use this strategy mentioned by this reviewer in our future research.

This revision further strengthened our manuscript. We thank you and the reviewers again for your time and valuable comments and look forward to a favorable reply from you!

Sincerely,

Sanhong Liu, Ph.D.

Haifan Lin, Ph.D.

Chao Fang, Ph.D.